# Engineering tumor-specific gene nanomedicine to recruit and activate T cells for enhanced immunotherapy

Yue Wang[1,6], Shi-Kun Zhou[1,6], Yan Wang[2], Zi-Dong Lu[2], Yue Zhang[1,3], Cong-Fei Xu ®[1,3,4] ✉ & Jun Wang ®[1,3,5] ✉

PD-1/PD-L1 blockade therapy that eliminates T-cell inhibition signals is successful, but poor benefits are often observed. Increasing T-cell infiltration and quantity of PD-1/PD-L1 inhibitors in tumor can improve efficacy but remains challenging. Here, we devise tumor-specific gene nanomedicines to mobilize tumor cells to secrete CXCL9 (T-cell chemokine) and anti-PD-L1 scFv (αPD-L1, PD-L1 blocking agent) for enhanced immunotherapy. The tyrosinase promoter-driven NP$_{Tyr-C9AP}$ can specifically co-express CXCL9 and αPD-L1 in melanoma cells, thereby forming a CXCL9 gradient for T-cell recruitment and high intratumoral αPD-L1 concentration for enhancing T-cell activation. As a result, NP$_{Tyr-C9AP}$ shows strong antimelanoma effects. Moreover, specific co-expression of CXCL9 and αPD-L1 in various tumor cells is achieved by replacing the tyrosinase promoter of NP$_{Tyr-C9AP}$ with a survivin promoter, which increases T-cell infiltration and activation and therapeutic efficacy in multiple tumors in female mice. This study provides a strategy to maximize the immunotherapeutic outcome regardless of the heterogeneous tumor microenvironment.

The interaction of programmed death ligand 1 (PD-L1) with its receptor PD-1 in the tumor microenvironment is one major cause of T-cell dysfunction and exhaustion, resulting in tumor immune escape[1–3]. Immune checkpoint inhibitors (ICIs) targeting PD-1/PD-L1 have shown good efficacy in cancer treatment by enhancing T-cell activation[4,5]; nevertheless, only a minority of patients derive benefits due to the intratumoral insufficiency of both T cells and injected PD-1/PD-L1 inhibitors[6–8]. Increasing either the tumor-infiltrating T cells or the intratumoral amount of PD-1/PD-L1 inhibitors has proven to be effective in improving PD-1/PD-L1 blockade therapy[9–11]. For example, anti-angiogenic inhibitors and immunogenic drugs can increase the responses of PD-1/PD-L1 inhibitors by indirectly enhancing T-cell infiltration[12–14]. Conjugating PD-1/PD-L1 inhibitors on nanoparticles can enhance their tumor accumulation and efficacy via the enhanced permeability and retention (EPR) effect[15,16]. However, these strategies may nonspecifically increase T-cell infiltration or deliver PD-1/PD-L1 inhibitors into normal tissues, limiting their efficacy while increasing side effects[17,18].

The highly directional migration of T cells is mainly guided by CXC-chemokine receptor 3 (CXCR3) following gradients of CXC-chemokine ligand (CXCL) 9, 10 or 11[19,20]. Studies have shown that the intratumoral abundance of CXCL9, 10 or 11 determines the numbers of tumor-infiltrating T cells, whereas most tumors restrain the expression of CXCL9, 10 or 11 to reduce T-cell recruitment, leading to the frequent

[1]School of Biomedical Sciences and Engineering, South China University of Technology, Guangzhou International Campus, Guangzhou 511442, P.R. China. [2]School of Medicine, South China University of Technology, Guangzhou 510006, P.R. China. [3]National Engineering Research Center for Tissue Restoration and Reconstruction, South China University of Technology, Guangzhou 510006, P.R. China. [4]Guangdong Provincial Key Laboratory of Biomedical Engineering, South China University of Technology, Guangzhou 510006, P.R. China. [5]Key Laboratory of Biomedical Materials and Engineering of the Ministry of Education, South China University of Technology, Guangzhou 510006, P.R. China. [6]These authors contributed equally: Yue Wang, Shi-Kun Zhou. ✉e-mail: xucf@scut.edu.cn; mcjwang@scut.edu.cn

failure of PD-1/PD-L1 blockade therapy[21,22]. Several groups have tried to inject recombinant proteins or viral expression vectors intratumorally to build gradients of CXCL9, 10 or 11 between tumor and peripheral tissues[23–25]. However, nonpersistent recombinant proteins can barely recruit enough T cells, while viral vectors result in nonspecific expression. Furthermore, intratumoral injection is inapplicable for metastatic tumors. Similarly, although local delivery systems have been reported to release PD-1/PD-L1 inhibitors into the tumor microenvironment directly[26,27], they are mainly useful for primary tumors and cannot ensure that PD-1/PD-L1 inhibitors penetrate tumors[28]. Therefore, a novel strategy that can specifically and robustly increase the intratumoral levels of T-cell chemokines and PD-1/PD-L1 inhibitors is needed for enhanced recruitment and activation of T cells. Genetic vectors, such as plasmids and viruses, can express various functional proteins and have been widely used for gene therapy[29,30]. More importantly, the levels and cell specificity of the expression of target proteins can be regulated by using different promoters[31].

In this work, we propose a tumor-specific expression strategy utilizing tumor-specific gene nanomedicines to drive tumor cells to secrete CXCL9 and anti-PD-L1 scFv (αPD-L1), thereby forming high intratumoral concentrations of CXCL9 and αPD-L1 for T-cell recruitment and PD-L1 blockade, respectively (Fig. 1). For this aim, a melanoma-specific coexpression plasmid of CXCL9 and αPD-L1 is constructed using the tyrosinase (Tyr) promoter[32], denoted pTyr-C9AP. Then, pTyr-C9AP is encapsulated into a biodegradable nanoparticle (NP) to fabricate the melanoma-specific gene nanomedicine (NP$_{Tyr-C9AP}$). After intravenous (i.v.) injection, NP$_{Tyr-C9AP}$ can cross physiological barriers and co-express abundant CXCL9 and αPD-L1 specifically in melanoma cells but not in normal cells. The resulting gradient of CXCL9 between the melanoma and peripheral tissues efficiently recruits T cells into the melanoma; moreover, the secreted αPD-L1 directly blocks intratumoral PD-L1, further promoting the recruited T cells to induce enhanced antimelanoma effects while avoiding T-cell-mediated autoimmunity. Furthermore, the survivin (Sur) promoter[33] is used for engineering the universal gene

nanomedicine (NP$_{Sur-C9AP}$). NP$_{Sur-C9AP}$ is demonstrated to specifically co-express CXCL9 and αPD-L1 in different tumor cells, which proves the expandability of tumor-specific gene nanomedicine to increase T-cell infiltration and activation for enhanced immunotherapy.

## Results

### Engineering the melanoma-specific gene nanomedicine NP$_{Tyr-C9AP}$

Melanoma is one of the deadliest cancers and has a high metastatic potential[34]. Although several ICIs have been successively approved for the treatment of metastatic melanoma, their clinical responses are unsatisfactory[35,36]. To specifically express CXCL9 and αPD-L1 in melanoma cells for recruiting T cells and unleashing their antimelanoma reactivity, we used the Tyr promoter responsible for melanin synthesis[32] to construct the melanoma-specific coexpression plasmid of CXCL9 and αPD-L1 (pTyr-C9AP) (Supplementary Figure 1a). Then, to overcome the in vivo delivery barriers of plasmid[37], we encapsulated pTyr-C9AP into biodegradable NPs fabricated with PEG-PLGA and the cationic lipid DOTAP (Fig. 2a). As measured by dynamic light scattering (DLS), the resulting NP-encapsulating pTyr-C9AP (NP$_{Tyr-C9AP}$) showed a 107.2 nm hydrodynamic diameter (Z-average size) (Fig. 2b), which is within the optimal size for accumulating in tumors via the EPR effect[38]. The zeta potential of NP$_{Tyr-C9AP}$ was 15.3 mV, and the slightly positive surface charge could enhance tumor cell uptake. Transmission electron microscopy (TEM) images showed that NP$_{Tyr-C9AP}$ had a spherical vesicle structure with a PEG shell to prolong blood circulation and protect pTyr-C9AP from degradation (Fig. 2c)[39]. In the stability test (Supplementary Figure 2), the size and polydispersity index (PDI) of NP$_{Tyr-C9AP}$ remained at approximately 100 nm and 0.2 after five days of incubation in medium containing 10% fetal bovine serum (FBS), suggesting that NP$_{Tyr-C9AP}$ could remain stable in the physiological environment. These results indicated that NP$_{Tyr-C9AP}$ was suitable for delivering pTyr-C9AP into tumor cells in vivo.

After the construction and characterization of the gene nanomedicine NP$_{Tyr-C9AP}$, we investigated the melanoma specificity of

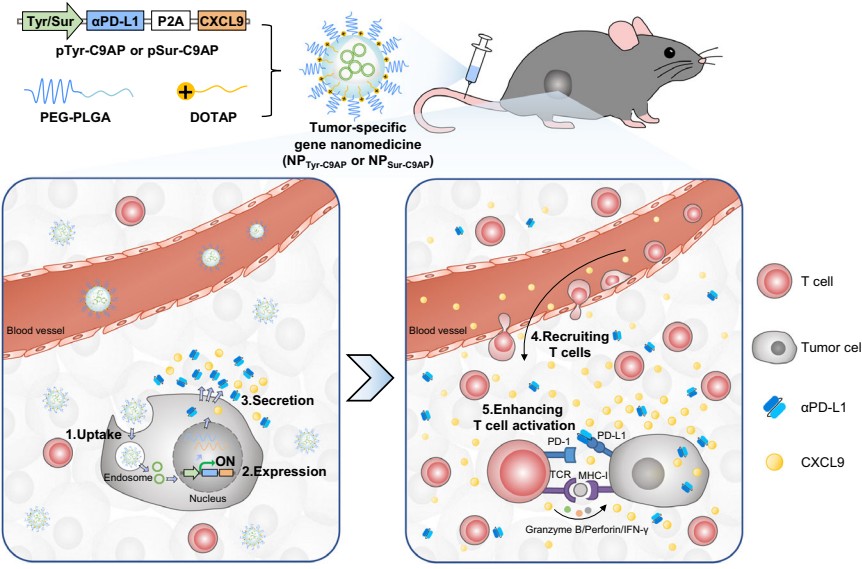

**Fig. 1 | Schematic illustration of devising tumor-specific gene nanomedicines to recruit and activate T cells for enhanced immunotherapy.** The Tyr or Sur promoter-driven CXCL9 and αPD-L1 coexpression plasmid (pTyr-C9AP or pSur-C9AP) was encapsulated into NPs fabricated with poly (ethylene glycol)-poly (lactic-co-glycolic acid) (PEG-PLGA) and cationic lipid 1,2-dioleoyl-3-trimethylammonium-propane (DOTAP) to prepare tumor-specific gene nanomedicines (NP$_{Tyr-C9AP}$ or NP$_{Sur-C9AP}$). P2A (a self-cleaving 2A peptide) was incorporated into the αPD-L1-P2A-CXCL9 cassette for the separate translation of CXCL9 and αPD-L1. After i.v.

injection, although NP$_{Tyr-C9AP}$ or NP$_{Sur-C9AP}$ was internalized by both tumor cells and normal cells, the Tyr or Sur promoter of pTyr-C9AP or pSur-C9AP drove the coexpression of CXCL9 and αPD-L1 specifically in melanoma cells or different tumor cells. As a result, CXCL9 secreted by the tumor cells established a chemokine gradient between the tumor and peripheral organs for recruiting T cells; αPD-L1 secreted by the tumor cells directly blocked PD-L1 in the tumor microenvironment to enhance the activation of the recruited T cells, thereby inducing enhanced antitumor T-cell responses.

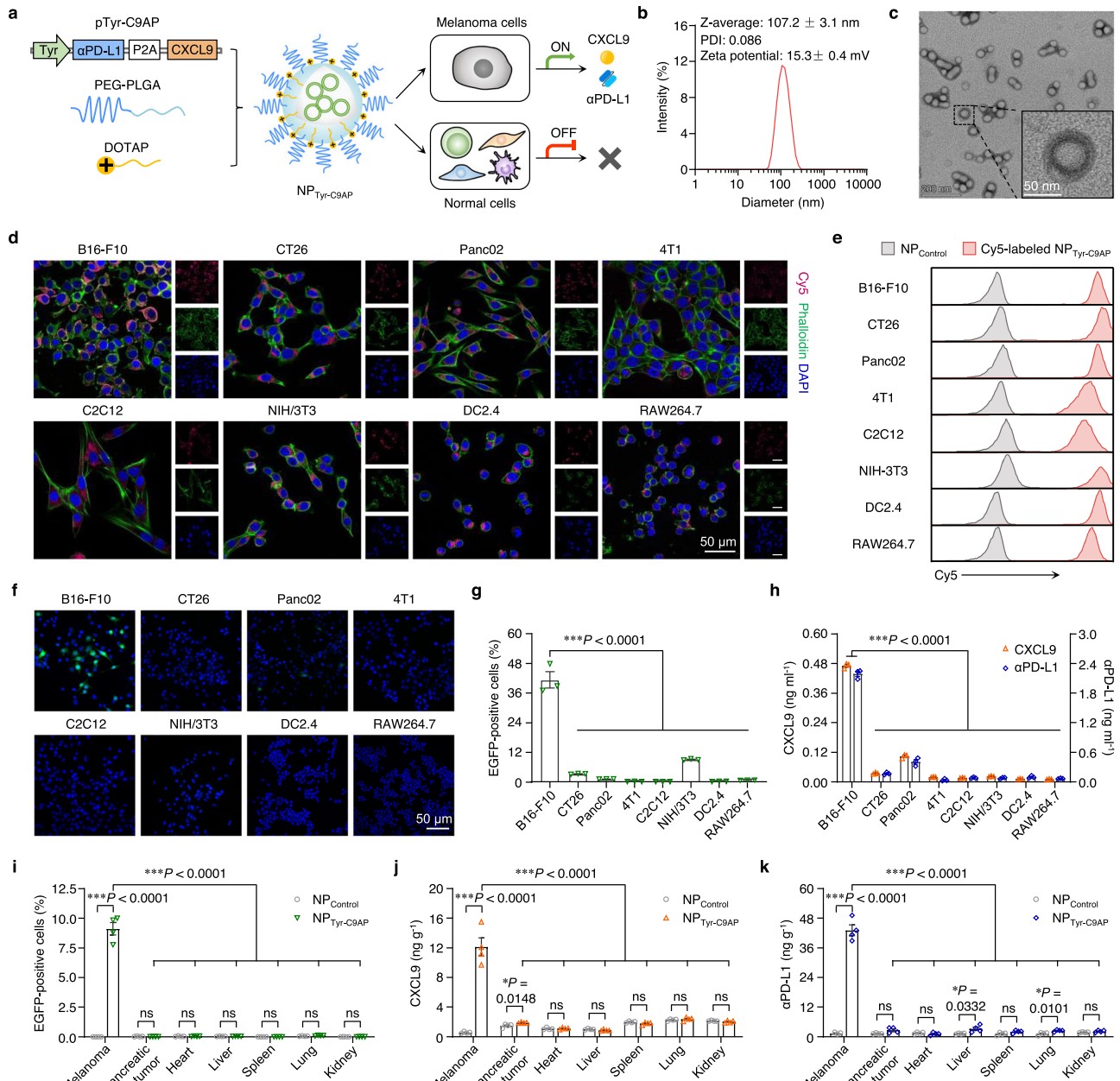

**Fig. 2 | Engineering NP$_{Tyr-C9AP}$ for melanoma-specific coexpression of CXCL9 and αPD-L1. a** Scheme of NP$_{Tyr-C9AP}$ construction using PEG-PLGA and DOTAP to encapsulate pTyr-C9AP. NP$_{Tyr-C9AP}$ can deliver pTyr-C9AP into melanoma cells and other normal cells, but the Tyr promoter of pTyr-C9AP will only turn on in melanoma cells to drive the coexpression of CXCL9 and αPD-L1. **b** Hydrodynamic diameter, PDI and zeta potential of NP$_{Tyr-C9AP}$. **c** Morphology of NP$_{Tyr-C9AP}$. Scale bar, 50 nm. **d, e** CLSM imaging (**d**) and flow cytometry analysis (**e**) of Cy5 fluorescence in B16-F10, CT26, Panc02, 4T1, C2C12, NIH/3T3, DC2.4 and RAW264.7 cells after the transfection of Cy5-labeled NP$_{Tyr-C9AP}$ (magenta). F-actin and nuclei were labeled with Alexa Fluor 488 phalloidin (green) and 4′,6-diamidino-2-phenylindole (DAPI, blue). Scale bar, 50 μm. **f, g** Fluorescence imaging of EGFP expression (**f**) and percentages of EGFP-positive cells (**g**) in B16-F10 cells and the other 7 types of cells after NP$_{Tyr-C9AP}$ transfection. Scale bar, 50 μm. **h** Concentrations of CXCL9 and αPD-L1 proteins in the culture supernatants of B16-F10 cells and the other 7 types of cells after NP$_{Tyr-C9AP}$ transfection. **i–k** Percentages of EGFP-positive cells (**i**), concentrations of CXCL9 (**j**) and αPD-L1 (**k**) in melanoma, pancreatic tumor and normal organs of the NP$_{Tyr-C9AP}$-treated bilateral tumor mice (n = 4 mice per group). The data are shown as the means ± SEM of n = 3 biologically independent samples (**g**, **h**), n = 4 biologically independent mice (**i–k**) or are representative of three independent experiments (**b**, **c–f**). Statistical data were analyzed by one-way ANOVA with Tukey's multiple comparison test (**g–k**), two-sided Student's t-test (**i–k**). ***P < 0.001; *P < 0.05; ns indicates no significant difference. Source data are provided as a Source Data file.

NP$_{Tyr-C9AP}$ by transfecting B16-F10 melanoma cells and 7 different cell lines: CT26, Panc02, 4T1, C2C12, NIH/3T3, DC2.4 and RAW264.7. The transfection efficiency was confirmed by incubating cyanine 5 (Cy5)-labeled NP$_{Tyr-C9AP}$, which encapsulated Cy5-labeled pTyr-C9AP, with these 8 types of cells. As visualized by confocal laser scanning microscopy (CLSM), Cy5-labeled NP$_{Tyr-C9AP}$ was internalized by 8 different cells (Fig. 2d). Although Cy5 fluorescence was stronger in B16-F10,

CT26, Panc02, NIH/3T3 and DC2.4 cells, flow cytometric histograms showed that all 8 types of cells were almost 100% Cy5-positive after the transfection of Cy5-labeled NP$_{Tyr-C9AP}$ (Fig. 2e), suggesting that NP$_{Tyr-C9AP}$ efficiently delivered pTyr-C9AP into different cells. Because the pTyr-C9AP used here carried an enhanced green fluorescent protein (EGFP) reporter gene behind the Tyr promoter (Supplementary Fig. 1b), the expression of EGFP was monitored to indicate the cell

specificity of $NP_{Tyr-C9AP}$. As observed by fluorescence microscopy, EGFP was only highly expressed in B16-F10 cells after $NP_{Tyr-C9AP}$ transfection (Fig. 2f). The percentage of EGFP-positive B16-F10 cells reached 41.2%, which was significantly higher than that of NIH/3T3 cells (9.0%), CT26 cells (3.2%) and other cells (less than 1%), as detected by flow cytometry (Fig. 2g). Thus, the Tyr promoter drove EGFP expression specifically in B16-F10 melanoma cells, although $NP_{Tyr-C9AP}$ indiscriminately delivered pTyr-C9AP into different cells, demonstrating the potential melanoma specificity of $NP_{Tyr-C9AP}$ for expressing the target genes (CXCL9 and αPD-L1).

The expression of CXCL9 and αPD-L1 was further quantified by quantitative reverse transcription PCR (qRT–PCR) and enzyme-linked immunosorbent assay (ELISA) after incubating $NP_{Tyr-C9AP}$ with B16-F10 cells or the other 7 types of cells. As shown in Supplementary Fig. 3a, b, the mRNA expression of CXCL9 and αPD-L1 in the $NP_{Tyr-C9AP}$-transfected B16-F10 cells increased to 586.4- and 1554.0-fold of that in the B16-F10 cells transfected with $NP_{Control}$ encapsulating a control pUC57 plasmid. In contrast, $NP_{Tyr-C9AP}$ transfection only showed weak or even no CXCL9 and αPD-L1 mRNA expression in the other 7 types of cells. Correspondingly, the concentrations of CXCL9 and αPD-L1 proteins in the culture supernatants of $NP_{Tyr-C9AP}$-transfected B16-F10 cells reached 0.47 and 2.19 ng ml$^{-1}$, which were much higher than that of other cells due to the melanoma specificity of $NP_{Tyr-C9AP}$ (Fig. 2h).

Next, we investigated the in vivo melanoma specificity of $NP_{Tyr-C9AP}$ in bilateral tumor models by injecting $NP_{Tyr-C9AP}$ into mice bearing melanoma and pancreatic tumor. As displayed in Supplementary Fig. 4, Cy5 fluorescence was observed in the melanoma, pancreatic tumor, liver, spleen, lung and kidney after the *i.v.* injection of Cy5-labeled $NP_{Tyr-C9AP}$, indicating that $NP_{Tyr-C9AP}$ accumulated in both two types of tumors and these normal organs. Significantly, after $NP_{Tyr-C9AP}$ treatment, the percentages of EGFP-positive cells in melanoma reached 9.11% of the total cells, while few cells were EGFP-positive in pancreatic tumor, heart, liver, spleen, lung and kidney (Fig. 2i). In addition, as shown in Fig. 2j, k, the concentrations of CXCL9 and αPD-L1 in melanoma of $NP_{Tyr-C9AP}$-treated mice were significantly increased to 12.12 and 43.12 ng per gram (ng g$^{-1}$) tumor tissue; as expected, the concentrations of CXCL9 and αPD-L1 in the pancreatic tumor, heart, liver, spleen, lung or kidney were not increased when compared to that of $NP_{Contorl}$ group. Collectively, these results demonstrated the ability of $NP_{Tyr-C9AP}$ to specifically express CXCL9 and αPD-L1 in melanoma, laying the foundation for the recruitment and activation of T cells for enhanced antimelanoma immunotherapy.

### $NP_{Tyr-C9AP}$ enhances T-cell recruitment and activation in vitro

Next, we investigated the efficiency of $NP_{Tyr-C9AP}$ for enhancing the recruitment and activation of T cells via melanoma cell-specific expression of CXCL9 and αPD-L1. A Tyr promoter-driven CXCL9 expression plasmid (pTyr-CXCL9, Supplementary Fig. 1c), expressing only CXCL9 specifically in melanoma cells, was constructed and encapsulated into $NP_{Tyr-CXCL9}$, which was used as a control for independently evaluating the efficiency of $NP_{Tyr-C9AP}$ on T-cell recruitment. The expression of CXCL9 in B16-F10 cells transfected with phosphate buffered saline (PBS), $NP_{Control}$, $NP_{Tyr-CXCL9}$ or $NP_{Tyr-C9AP}$ was validated prior to the study of T-cell recruitment. As shown in Fig. 3a and Supplementary Fig. 5a, the B16-F10 cells transfected with either $NP_{Tyr-CXCL9}$ or $NP_{Tyr-C9AP}$ expressed fairly high levels of CXCL9. The mRNA expression of CXCL9 in the $NP_{Tyr-CXCL9}$- or $NP_{Tyr-C9AP}$-transfected B16-F10 cells was 3570.4- or 3319.5-fold that in the PBS-treated B16-F10 cells (Supplementary Fig. 5a). The concentrations of CXCL9 protein in the culture supernatants of B16-F10 cells reached 0.44 or 0.42 ng ml$^{-1}$ after $NP_{Tyr-CXCL9}$ or $NP_{Tyr-C9AP}$ transfection; in contrast, nearly no CXCL9 secretion (0.02 ng ml$^{-1}$) was detected in the PBS and $NP_{Control}$ groups (Fig. 3a).

After verifying the high concentration of CXCL9 secreted by $NP_{Tyr-CXCL9}$- or $NP_{Tyr-C9AP}$-transfected B16-F10 cells, we evaluated the

potency of T-cell recruitment by tracking T-cell chemotaxis in a transwell migration assay. DiI-labeled CD8$^+$ T cells (red) were cultured in the upper chamber of transwell plates, and the supernatants of the PBS-, $NP_{Control}$-, $NP_{Tyr-CXCL9}$- or $NP_{Tyr-C9AP}$-transfected B16-F10 cells were placed in the lower chamber. As displayed in Fig. 3b, in comparison with those of the PBS and $NP_{Control}$ groups, the supernatants of either the $NP_{Tyr-CXCL9}$- or $NP_{Tyr-C9AP}$-transfected B16-F10 cells that contained a high concentration of CXCL9 recruited many more DiI-labeled CD8$^+$ T cells (red) at 30 min after being placed in the lower chamber, and the numbers of DiI-labeled CD8$^+$ T cells that recruited into the lower chamber rapidly increased within 180 min. Additionally, the supernatants of the $NP_{Tyr-CXCL9}$- and $NP_{Tyr-C9AP}$-transfected B16-F10 cells recruited a similar quantity of DiI-labeled CD8$^+$ T cells ($1.26 \times 10^5$ cells in the $NP_{Tyr-CXCL9}$ group and $1.14 \times 10^5$ cells in the $NP_{Tyr-C9AP}$ group at 180 min time point) due to their approximate levels of CXCL9, as quantified by flow cytometry-based cell counting (Fig. 3c and Supplementary Fig. 6). These results demonstrated that $NP_{Tyr-C9AP}$ was efficient in mobilizing B16-F10 cells to secrete CXCL9, and the secreted CXCL9 could establish a concentration gradient for recruiting T cells.

Similarly, a Tyr promoter-driven αPD-L1 expression plasmid (pTyr-αPD-L1, Supplementary Fig. 1c) expressing only αPD-L1 specifically in melanoma cells was constructed to prepare the control $NP_{Tyr-αPD-L1}$ to independently evaluate the efficiency of $NP_{Tyr-C9AP}$ in enhancing T-cell activation. qRT–PCR and western blot analysis showed that the $NP_{Tyr-αPD-L1}$- or $NP_{Tyr-C9AP}$-transfected B16-F10 cells expressed high levels of αPD-L1, while the B16-F10 cells transfected with PBS or $NP_{Control}$ had no αPD-L1 expression (Supplementary Fig. 5b, Supplementary Fig. 7). As quantified by ELISA, the concentrations of αPD-L1 protein in the culture supernatants of $NP_{Tyr-αPD-L1}$- or $NP_{Tyr-C9AP}$-transfected B16-F10 cells reached 1.75 or 1.74 ng ml$^{-1}$, respectively (Fig. 3d).

Then, the culture supernatants of the B16-F10 cells transfected with PBS, $NP_{Control}$, $NP_{Tyr-αPD-L1}$ or $NP_{Tyr-C9AP}$ were incubated with interferon-gamma (IFN-γ)-stimulated B16-F10 cells, which had high PD-L1 expression (Supplementary Fig. 8), to analyze the PD-L1 binding capacity of the secreted αPD-L1. Because αPD-L1 was fused with a hexahistidine (6×His) tag, αPD-L1 binding on the IFN-γ-stimulated B16-F10 cells was labeled with Alexa Fluor 647 anti-6×His tag antibody before flow cytometry analysis. As shown in Fig. 3e, in comparison with that of the PBS and $NP_{Control}$ groups, abundant binding of αPD-L1 was detected on the surface of the IFN-γ-stimulated B16-F10 cells after incubation with the culture supernatants of the $NP_{Tyr-αPD-L1}$- or $NP_{Tyr-C9AP}$-transfected B16-F10 cells. ELISA analysis also demonstrated that αPD-L1 secreted in the culture supernatants of the $NP_{Tyr-αPD-L1}$- or $NP_{Tyr-C9AP}$-transfected B16-F10 cells strongly bound to the PD-L1 molecule (Supplementary Fig. 9). These results suggested that $NP_{Tyr-C9AP}$ could mobilize B16-F10 cells to secrete αPD-L1 to block PD-L1.

Subsequently, we evaluated the efficiency of PD-L1 blockade on increasing the cytotoxicity of CD8$^+$ T cells. Ovalbumin (OVA) and EGFP-expressing B16-F10 (B16-F10-OVA-EGFP) cells were stimulated with IFN-γ and then cocultured with OVA-specific CD8$^+$ T cells in the culture supernatants of the B16-F10 cells transfected with PBS, $NP_{Control}$, $NP_{Tyr-αPD-L1}$ or $NP_{Tyr-C9AP}$. The viability of the IFN-γ-stimulated B16-F10-OVA-EGFP cells was analyzed by propidium iodide (PI) staining and Cell Counting Kit-8 (CCK-8) assays. As shown in Fig. 3f, in the culture supernatants of the PBS- or $NP_{Control}$-transfected B16-F10 cells, only 4.9% of the IFN-γ-stimulated B16-F10-OVA-EGFP cells were dead (PI-positive), indicating that the highly expressed PD-L1 of the IFN-γ-stimulated B16-F10-OVA-EGFP cells inhibited the activation and cytotoxicity of OVA-specific CD8$^+$ T cells. In contrast, the culture supernatants of the $NP_{Tyr-αPD-L1}$- or $NP_{Tyr-C9AP}$-transfected B16-F10 cells increased the PI-positive percentages of IFN-γ-stimulated B16-F10-OVA-EGFP cells to a similarly high level (45.4% and 42.8%) by blocking PD-L1. The CCK-8 assay also demonstrated that PD-L1 blockade mediated by αPD-L1 in the culture supernatants of the $NP_{Tyr-αPD-L1}$- or

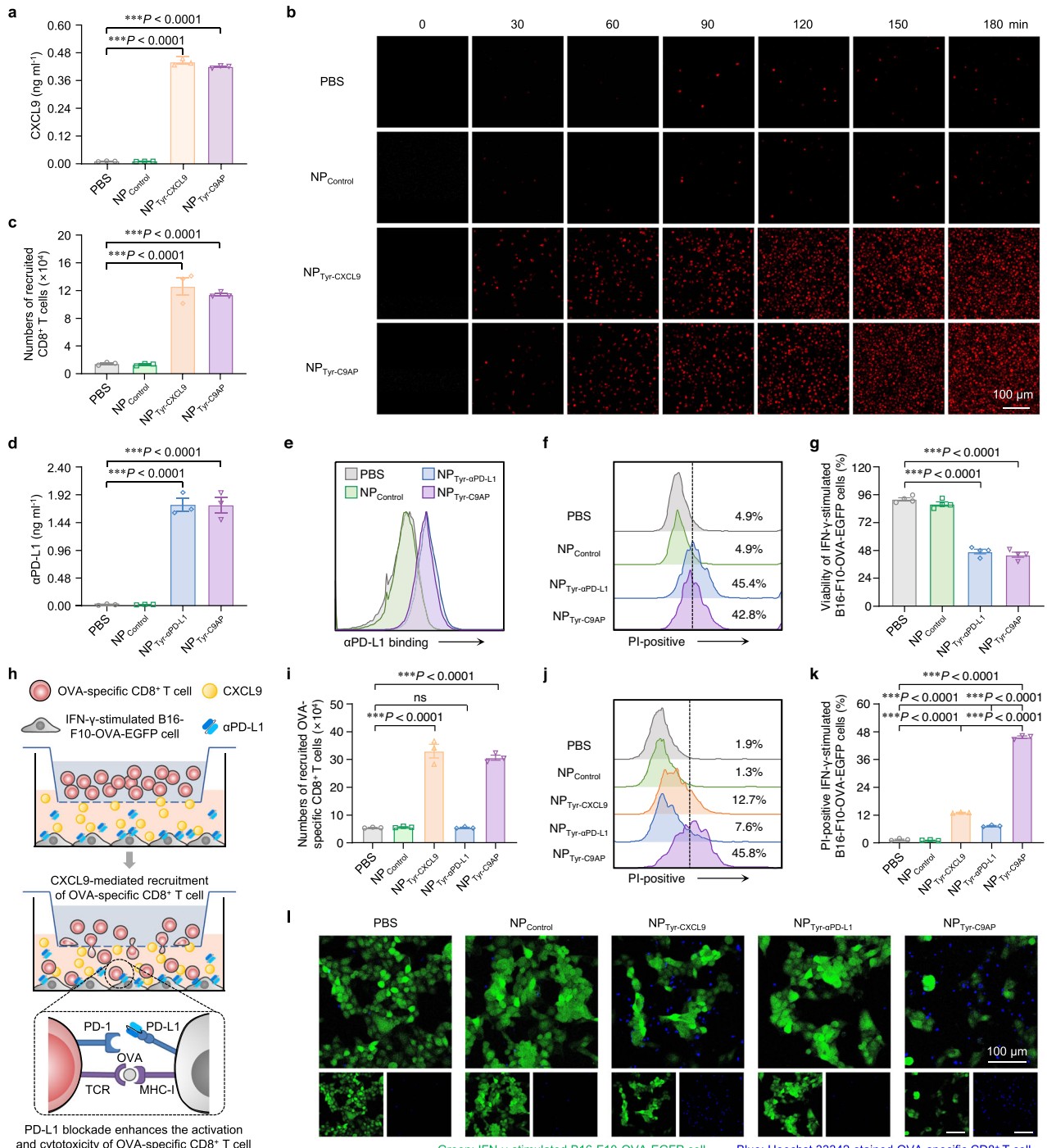

**Fig. 3 | NP_{Tyr-C9AP}-mediated coexpression of CXCL9 and αPD-L1 enhances T-cell recruitment and activation in vitro. a** Concentrations of CXCL9 secreted in the culture supernatants of B16-F10 cells after the transfection of NP_{Tyr-C9AP} or other controls. **b, c** CLSM imaging (**b**) and flow cytometry-based cell counting (**c**) of CD8⁺ T cells (stained with DiI dye, red) recruited into the lower chamber of the transwell plates after addition of the culture supernatants of the B16-F10 cells transfected with NP_{Tyr-C9AP} or other controls. **d** Concentrations of αPD-L1 secreted in the culture supernatants of the B16-F10 cells transfected with NP_{Tyr-C9AP} or other controls. **e** Flow cytometry analysis of the PD-L1 binding capacity of αPD-L1 secreted by the NP_{Tyr-C9AP}-transfected B16-F10 cells. **f, g** Flow cytometry analysis of PI-positive (dead) percentages (**f**) and CCK-8 assays of cell viability (**g**) of the IFN-γ-stimulated B16-F10-OVA-EGFP cells after coincubation with OVA-specific CD8⁺ T cells in the culture supernatants of the B16-F10 cells transfected with NP_{Tyr-C9AP} or other

controls. **h** Scheme of the transwell migration and cytotoxicity assay for detecting the synergistic effects of NP_{Tyr-C9AP} on recruiting and activating T cells to kill melanoma cells. **i** Numbers of OVA-specific CD8⁺ T cells recruited into the lower chamber. **j, k** Flow cytometric histograms (**j**) and statistics (**k**) of the PI-positive (dead) percentages of IFN-γ-stimulated B16-F10-OVA-EGFP cells. **l** CLSM imaging of the recruitment of OVA-specific CD8⁺ T cells (stained with Hoechst 33342, blue) and the killing of the IFN-γ-stimulated B16-F10-OVA-EGFP cells (green). Scale bar, 100 μm. The data are shown as the means ± SEM of $n = 3$ biologically independent samples (**a, c, d, i, k**) or 4 biologically independent samples (**g**) or are representative of three independent experiments (**b, e, f, j, l**). Statistical data were analyzed by one-way ANOVA with Tukey's multiple comparison test. ***$P < 0.001$. Source data are provided as a Source Data file.

$NP_{Tyr-C9AP}$-transfected B16-F10 cells significantly increased the cytotoxicity of OVA-specific CD8+ T cells, thereby reducing the viability of the IFN-γ-stimulated B16-F10-OVA-EGFP cells to 46.9% ($NP_{Tyr-αPD-L1}$ group) or 44.0% ($NP_{Tyr-C9AP}$ group), respectively (Fig. 3g). These results indicated that $NP_{Tyr-C9AP}$ could efficiently enhance the activation and cytotoxicity of T cells by mobilizing B16-F10 cells to secrete αPD-L1 for PD-L1 blockade.

After separately proving the ability of $NP_{Tyr-C9AP}$ to enhance T-cell recruitment and activation by expressing CXCL9 and αPD-L1, we designed a transwell migration and cytotoxicity assay to detect the synergistic effects of $NP_{Tyr-C9AP}$ on recruiting and activating T cells to kill melanoma cells. As illustrated in Fig. 3h, IFN-γ-stimulated B16-F10-OVA-EGFP cells were seeded in the lower chamber of transwell plates, while OVA-specific CD8+ T cells were added to the upper chamber. The recruitment and cytotoxicity of OVA-specific CD8+ T cells were monitored after replacing the original medium of the lower chamber with the culture supernatants of the B16-F10 cells transfected with PBS, $NP_{Control}$, $NP_{Tyr-CXCL9}$, $NP_{Tyr-αPD-L1}$ or $NP_{Tyr-C9AP}$. At 12 h after replacing the culture supernatants, in the $NP_{Tyr-C9AP}$ group, $3.03 \times 10^5$ OVA-specific CD8+ T cells were recruited into the lower chamber by CXCL9 (Fig. 3i), and the recruited OVA-specific CD8+ T cells efficiently killed the IFN-γ-stimulated B16-F10-OVA-EGFP cells due to αPD-L1-mediated PD-L1 blockade, resulting in 45.8% PI-positive IFN-γ-stimulated B16-F10-OVA-EGFP cells (Fig. 3j, k). In contrast, although comparable OVA-specific CD8+ T cells ($3.27 \times 10^5$) were recruited into the lower chamber by CXCL9 in the $NP_{Tyr-CXCL9}$ group, fewer (12.7%) PI-positive IFN-γ-stimulated B16-F10-OVA-EGFP cells were detected owing to PD-L1-mediated T-cell inhibition. Because $NP_{Tyr-αPD-L1}$ transfection only resulted in αPD-L1 expression for PD-L1 blockade, less than $0.6 \times 10^5$ OVA-specific CD8+ T cells migrated into the lower chamber and caused only 7.6% PI-positive IFN-γ-stimulated B16-F10-OVA-EGFP cells. In addition, the minimum cytotoxicity of OVA-specific CD8+ T cells to the IFN-γ-stimulated B16-F10-OVA-EGFP cells was detected in the PBS (1.9% PI-positive) and $NP_{Control}$ (1.3% PI-positive) groups because CXCL9 and αPD-L1 were not expressed.

Moreover, CLSM images verified that $NP_{Tyr-C9AP}$ transfection recruited abundant OVA-specific CD8+ T cells (stained with Hoechst 33342, blue) into the lower chamber, whose cytotoxicity dramatically reduced the numbers of IFN-γ-stimulated B16-F10-OVA-EGFP cells (green) (Fig. 3l). Both the $NP_{Tyr-CXCL9}$ and $NP_{Tyr-αPD-L1}$ groups showed slight killing of the IFN-γ-stimulated B16-F10-OVA-EGFP cells because they only expressed CXCL9 and αPD-L1, respectively. Taken together, these results demonstrated that $NP_{Tyr-C9AP}$ could mobilize melanoma cells to coexpress CXCL9 and αPD-L1 to enhance both the recruitment and activation of T cells, thereby resulting in synergistic tumor-killing effects.

## $NP_{Tyr-C9AP}$ enhances the antimelanoma effects of T cells in vivo

Given the strong ability of $NP_{Tyr-C9AP}$ to enhance T-cell recruitment and activation in vitro, we further investigated whether $NP_{Tyr-C9AP}$ could induce enhanced therapeutic effects against melanoma. The efficiency of $NP_{Tyr-C9AP}$ for coexpressing CXCL9 and αPD-L1 in melanoma in vivo was first assessed. As shown in Supplementary Fig. 10a, b, qRT–PCR analysis verified that both CXCL9 and αPD-L1 mRNA were highly expressed in melanoma tissues after intravenously injecting $NP_{Tyr-C9AP}$ into mice bearing B16-F10 melanoma. The mRNA expression of CXCL9 and αPD-L1 in $NP_{Tyr-C9AP}$-treated melanoma reached 459.5- or 584.1-fold that of the PBS group. In addition, ELISA analysis indicated that the CXCL9 protein concentration in melanoma was increased to 14.07 ng g$^{-1}$ by $NP_{Tyr-C9AP}$ treatment (Fig. 4a), and the serum CXCL9 concentration was correspondingly elevated to 0.26 ng ml$^{-1}$ due to the diffusion of CXCL9 from the melanoma to the periphery (Supplementary Fig. 11). Thus, T cells were expected to be recruited into melanoma by the CXCL9 gradient between melanoma and peripheral lymphoid organs. As shown in Fig. 4b and Supplementary Fig. 12, ELISA

and western blot analysis showed that $NP_{Tyr-C9AP}$ treatment resulted in high expression of αPD-L1 protein in melanoma (57.82 ng g$^{-1}$) enabling efficient PD-L1 blockade to enhance T-cell activation in the tumor microenvironment.

Then, the mice bearing B16-F10 melanoma were intravenously injected with $NP_{Tyr-C9AP}$ every other day for five injections to investigate its therapeutic efficacy (Fig. 4c). The mice injected with $NP_{Tyr-CXCL9}$ or $NP_{Tyr-αPD-L1}$ that expressed comparable levels of either CXCL9 or αPD-L1 in melanoma (Fig. 4a, b) were used as controls to evaluate the independent efficacy of CXCL9 or αPD-L1. As shown in Fig. 4d, e, in comparison with the negative control PBS or $NP_{Control}$, $NP_{Tyr-CXCL9}$ or $NP_{Tyr-αPD-L1}$ that expressed CXCL9 or αPD-L1 alone in melanoma significantly inhibited the growth of melanoma, achieving 59.0% or 69.5% tumor growth inhibition rates. Notably, $NP_{Tyr-C9AP}$ that coexpressed CXCL9 and αPD-L1 exhibited synergistic antimelanoma effects, and the tumor growth inhibition rate reached 84.3%. Moreover, the tumor weights of the $NP_{Tyr-C9AP}$ group measured at the treatment endpoint were only 16.7% that of the PBS group, also demonstrating the enhanced therapeutic efficacy of $NP_{Tyr-C9AP}$ compared to that of the $NP_{Tyr-CXCL9}$ (36.9% that of the PBS group) or $NP_{Tyr-αPD-L1}$ (29.5% that of the PBS group) groups (Fig. 4f). Moreover, we validated the enhanced antimelanoma effects of $NP_{Tyr-C9AP}$ by monitoring the survival of the melanoma-bearing mice. As shown in Fig. 4g, $NP_{Tyr-CXCL9}$ or $NP_{Tyr-αPD-L1}$ treatment significantly prolonged overall survival, and the $NP_{Tyr-C9AP}$-treated mice showed longer overall survival. The median survival time (MST) of the $NP_{Tyr-C9AP}$-treated mice was increased to 34 days, which was higher than that of the $NP_{Tyr-CXCL9}$ (31 days) and $NP_{Tyr-αPD-L1}$ (30 days) groups.

Additionally, $NP_{Tyr-C9AP}$ was proven to be well tolerated and safe in mice because no weight loss was observed during the treatment course (Supplementary Fig. 13), no damage to the major organs was detected by histopathology (Supplementary Fig. 14). And, no potential liver toxicity was detected by biochemical analysis of liver injury markers after $NP_{Tyr-C9AP}$ treatment (Supplementary Fig. 15). Furthermore, after the i.v. injection of $NP_{Tyr-C9AP}$ into the bilateral tumor model, mice bearing B16-F10 melanoma and Panc02 pancreatic tumor, only the growth of melanoma were significantly inhibited by $NP_{Tyr-C9AP}$ treatment while that of pancreatic tumor was not affected, indicating that $NP_{Tyr-C9AP}$ could induce melanoma-specific therapeutic effects (Supplementary Fig. 16).

To determine whether the enhanced therapeutic efficacy of $NP_{Tyr-C9AP}$ resulted from enhancing the intratumoral infiltration and activation of T cells, we analyzed the number and phenotype of T cells in melanoma at the end of the treatments (the gating strategies of flow cytometry analysis are in Supplementary Figure 17). As shown in Fig. 5a and Supplementary Fig. 18, $NP_{Tyr-CXCL9}$ increased the tumor-infiltrating CD3+ T cells to 10.4% of CD45+ lymphocytes by CXCL9-mediated T-cell recruitment, while $NP_{Tyr-αPD-L1}$ increased the tumor-infiltrating CD3+ T cells to 8.9% of CD45+ lymphocytes by αPD-L1-enhanced T-cell activation. $NP_{Tyr-C9AP}$, which coexpressed CXCL9 and αPD-L1, significantly increased the percentage of CD3+ T cells to 21.4% of CD45+ lymphocytes by the cooperative recruitment and activation of T cells. Accordingly, in comparison with $NP_{Tyr-CXCL9}$ and $NP_{Tyr-αPD-L1}$ groups, the number of CD3+ T cells in $NP_{Tyr-C9AP}$-treated melanoma were significantly increased to $1.87 \times 10^6$ cells per gram tumor, verifying the synergistic effect of CXCL9 and αPD-L1 (Fig. 5b). The number of tumor-infiltrating CD8+ T cells was correspondingly increased to $0.94 \times 10^6$ cells per gram tumor in the $NP_{Tyr-C9AP}$ group, which was higher than that in the other groups (Fig. 5c, Supplementary Fig. 19a). In addition, the expression of CD69, a marker of T-cell activation, was analyzed. As shown in Fig. 5d and Supplementary Fig. 19b, the percentages of CD69-positive CD8+ T cells in the $NP_{Tyr-αPD-L1}$ and $NP_{Tyr-C9AP}$ groups were 3.3- and 3.7-fold that in the PBS group, confirming that αPD-L1-mediated PD-L1 blockade enhanced the activation of CD8+ T cells. Owing to the synergy of CXCL9 and αPD-L1, the number of CD69-positive CD8+

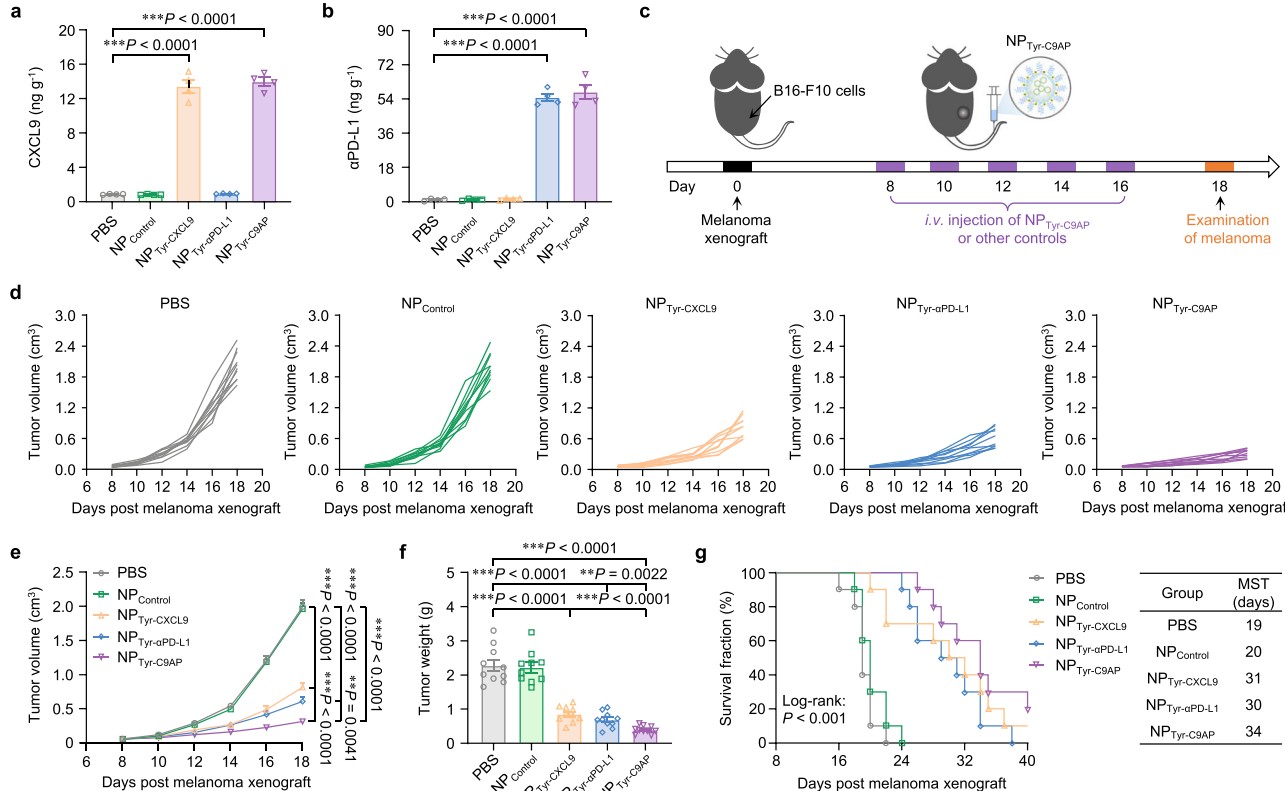

**Fig. 4 | NP_Tyr-C9AP induces enhanced therapeutic efficacy against B16-F10 melanoma. a, b** Concentrations of CXCL9 (**a**) and αPD-L1 (**b**) in B16-F10 melanoma after the *i.v.* injection of NP_Tyr-C9AP or other controls (*n* = 4 mice per group). **c** Therapeutic scheme. Mice bearing B16-F10 melanoma were treated by five *i.v.* injections of NP_Tyr-C9AP or other controls. **d, e** Individual (**d**) and average (**e**) tumor growth curves of the B16-F10 melanoma-bearing mice treated with NP_Tyr-C9AP or other controls (*n* = 10 mice per group). **f** Final tumor weights of the B16-F10 melanoma-bearing mice after the treatments (*n* = 10 mice per group). **g** Survival curves

of the B16-F10 melanoma-bearing mice treated with NP_Tyr-C9AP or other controls (*n* = 10 mice per group). The median survival time (MST) of each group is provided. The data are shown as the means ± SEM of *n* = 4 biologically independent mice (**a, b**), 10 biologically independent mice (**e, f**). Statistical data were analyzed by one-way ANOVA with Tukey's multiple comparison test (**a, b, f**), two-way ANOVA with the Greenhouse–Geisser correction (**e**) or log-rank (Mantel–Cox) test (**g**). *$P < 0.05$; **$P < 0.01$; ***$P < 0.001$. Source data are provided as a Source Data file.

T cells in NP_Tyr-C9AP-treated melanoma reached $2.33 \times 10^5$ cells per gram tumor, which was 5.1- and 2.1-fold that in the NP_Tyr-CXCL9 and NP_Tyr-αPD-L1 groups, respectively (Fig. 5e). The ratio of tumor-infiltrating CD8$^+$ T cells/regulatory T cells (Tregs) was shown to be elevated to 13.37 after NP_Tyr-C9AP treatment (Fig. 5f). Collectively, these results demonstrated that NP_Tyr-C9AP-mediated coexpression of CXCL9 and αPD-L1 synergistically enhanced the intratumoral infiltration and activation of T cells, especially CD8$^+$ T cells, which was also validated by immunofluorescence staining (Fig. 5g).

Because the tumor-killing effects of CD8$^+$ T cells are mediated by cytotoxic proteins and cytokines, we further analyzed the expression of granzyme B, perforin and IFN-γ[40,41]. As shown in Fig. 5h–j and Supplementary Fig. 20, NP_Tyr-C9AP treatment significantly increased the proportions of granzyme B-, perforin- and IFN-γ-positive CD8$^+$ T cells, accounting for 68.4%, 53.2% and 31.1% of CD8$^+$ T cells, respectively, indicating that NP_Tyr-C9AP enhanced the antitumor effector functions of CD8$^+$ T cells. Notably, NP_Tyr-C9AP treatment additionally increased the percentages of natural killer (NK) cells and the ratios of M1-like/M2-like macrophages (Fig. 5k and l, Supplementary Fig. 21a–c); and, the percentages of myeloid-derived suppressor cells (MDSCs) and Tregs were reduced after NP_Tyr-C9AP treatment (Fig. 5m, n, Supplementary Fig. 21d, e). Moreover, we found that pro-inflammatory cytokines (IFN-γ, IL-6, GM-CSF) were increased and anti-inflammatory cytokines (IL-4 and IL-10) were decreased by the expression of CXCL9 and αPD-L1, which promoted the reduction of immunosuppressive cell populations (Supplementary Fig. 22). These results suggested that CXCL9 and

αPD-L1 expressed by NP_Tyr-C9AP not only reinforced T-cell responses but also improved the tumor immune microenvironment, which jointly resulted in enhanced immunotherapy against melanoma.

## NP_Tyr-C9AP shows superior antimelanoma effects to anti-PD-L1 antibody

Because anti-PD-L1 therapy is usually conducted by systemic administration of anti-PD-L1 antibody in clinic, we then compared the antimelanoma effects of NP_Tyr-C9AP with anti-PD-L1 antibody. B16-F10 melanoma-bearing mice were intravenously injected with PBS, anti-PD-L1 antibody, NP_Tyr-αPD-L1, NP_Tyr-CXCL9 & anti-PD-L1 antibody or NP_Tyr-C9AP every other day for five injections (Fig. 6a). As shown in Fig. 6b, c, the tumor growth inhibition rate of NP_Tyr-αPD-L1 group (64.2%) was higher than that of anti-PD-L1 group (58.0%), and the final tumor weights of NP_Tyr-αPD-L1 group were lower than that of anti-PD-L1 group, indicating that specifically expressing αPD-L1 in the tumor microenvironment using our nanomedicine induced superior efficacy as opposed to systemic injection of anti-PD-L1 antibody. Significantly, NP_Tyr-C9AP group exhibited the highest tumor growth inhibition rate (78.2%) and the lowest final tumor weights due to tumor-specific coexpression of CXCL9 and αPD-L1. Notably, as shown in Supplementary Fig. 23, the systemic injection of anti-PD-L1 significantly decreased the body weights of mice, while either NP_Tyr-αPD-L1 or NP_Tyr-C9AP treatment did not obviously affect the body weights when compared to PBS group, suggesting local expression of αPD-L1 by our nanomedicine could reduce the side effects of anti-PD-L1 antibody. These results demonstrated that the superior antimelanoma efficacy

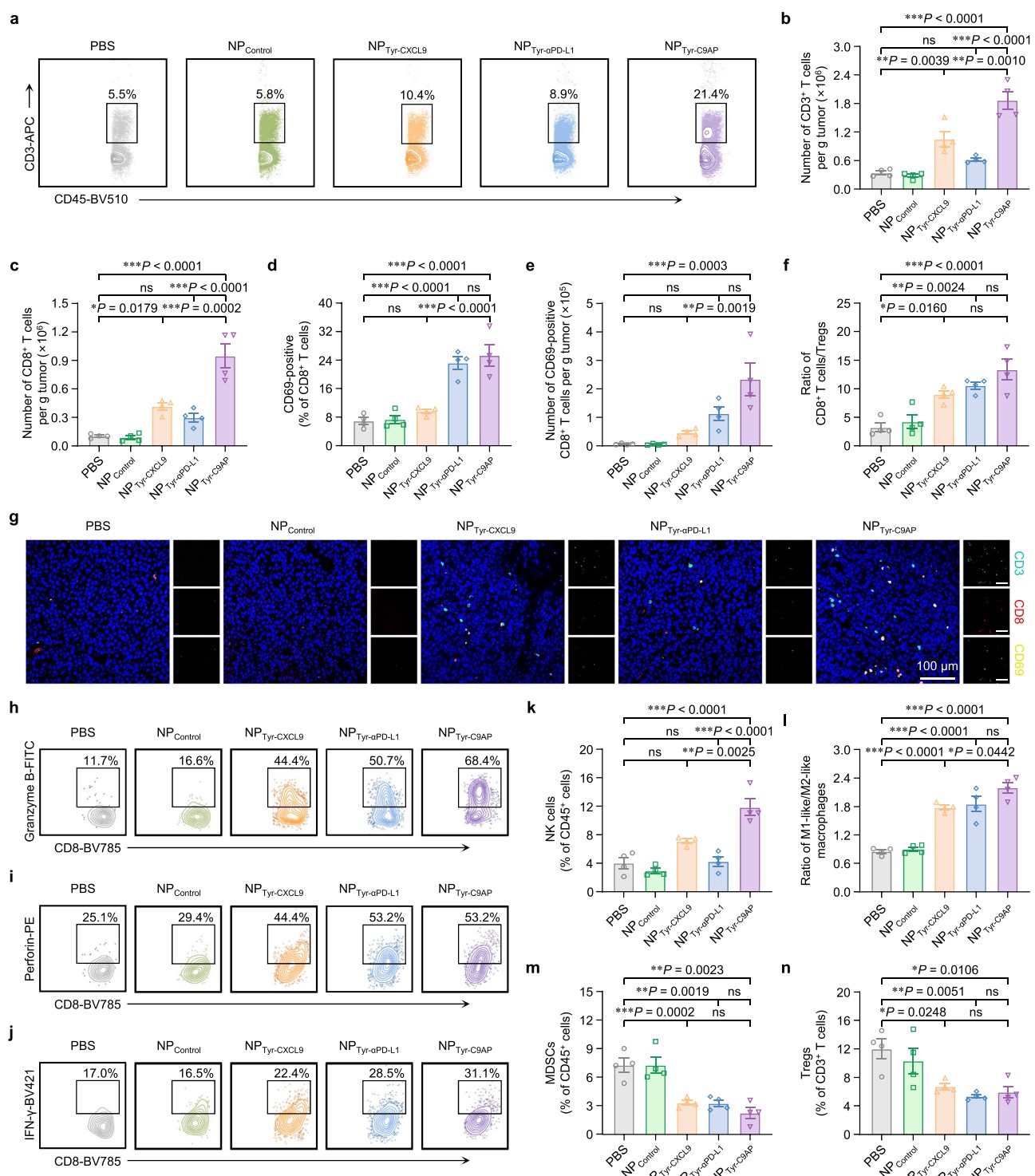

**Fig. 5 | Profiling of the improved intratumoral infiltration and activation of T cells after NP_Tyr-C9AP treatment. a, b** Flow cytometry plots of CD3⁺ T cell percentages in CD45⁺ cells (**a**) or numbers of CD3⁺ T cells (**b**) of the melanoma samples from the mice treated with NP_Tyr-C9AP or other controls ($n = 4$ mice per group). **c** Numbers of CD8⁺ T cells in melanoma. **d, e** Percentages (**d**) and numbers (**e**) of CD69-positive CD8⁺ T cells in melanoma. **f** Ratios of CD8⁺ T cells/Tregs in melanoma. **g** Representative immunofluorescence images of the melanoma. CD3 (aqua green), CD8 (red) and CD69 (yellow) of T cells were labeled by fluorescent antibodies. Scale bar, 100 μm. **h** to **j** Representative flow cytometry plots of granzyme

B- (**h**), perforin- (**i**) and IFN-γ- (**j**) positive CD8⁺ T cells in melanoma. **k–n** Percentages of NK cells in CD45⁺ cells (**k**), ratios of M1-like/M2-like macrophages (**l**), percentages of MDSCs in CD45⁺ cells (**m**) and percentages of Tregs in CD3⁺ T cells (**n**) of melanoma. The data are shown as the means ± SEM of $n = 4$ biologically independent mice (**b–f, k–n**) or are representative of four biologically independent mice (**a, g–j**). Statistical data were analyzed by one-way ANOVA with Tukey's multiple comparison test (**b–f, k–n**). *$P < 0.05$; **$P < 0.01$; ***$P < 0.001$; ns indicates no significant difference. Source data are provided as a Source Data file.

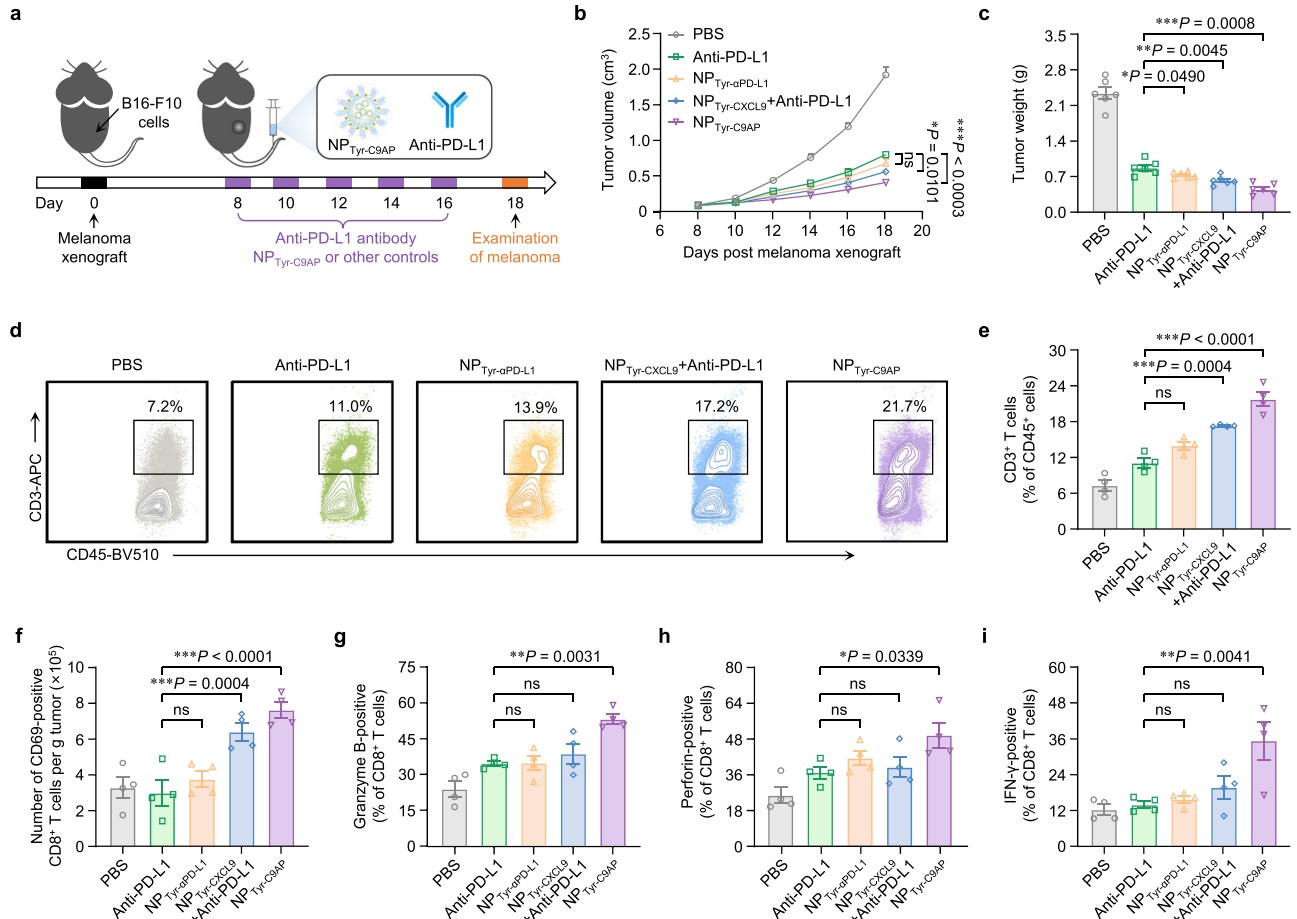

**Fig. 6 | NP$_{Tyr-C9AP}$ exhibits superior efficacy to systemically injected anti-PD-L1 antibody. a** Therapeutic scheme. Mice bearing B16-F10 melanoma were treated by five *i.v.* injections of NP$_{Tyr-C9AP}$, anti-PD-L1 antibody or other controls. **b, c** Tumor growth curves (**b**) and final tumor weights (**c**) of the B16-F10 melanoma-bearing mice after different treatments (*n* = 6 mice per group). **d, e** Flow cytometry plots of CD3$^+$ T cell percentages in CD45$^+$ cells (**d**) or statistical percentages of CD3$^+$ T cells (**e**) in melanoma after different treatments (*n* = 4 mice per group). **f** Numbers of CD69-positive CD8$^+$ T cells in melanoma. **g–i** Granzyme B- (**g**), perforin- (**h**) and IFN-

γ- (**i**) positive CD8$^+$ T cells in melanoma. The data are shown as the means ± SEM of *n* = 4 biologically independent mice (**e–i**), *n* = 6 biologically independent mice (**b, c**) or are representative of four biologically independent mice (**d**). Statistical data were analyzed by two-way ANOVA with the Greenhouse–Geisser correction (**b**), one-way ANOVA with Tukey's multiple comparison test (**c, e–i**). *$P$ < 0.05; **$P$ < 0.01; ***$P$ < 0.001; ns indicates no significant difference. Source data are provided as a Source Data file.

and safety of NP$_{Tyr-C9AP}$ as opposed to the systemically injected anti-PD-L1 antibody.

After the treatments of NP$_{Tyr-C9AP}$ or anti-PD-L1 antibody, the infiltration and activation of T cells in melanoma were characterized by flow cytometry. As shown in Fig. 6d, e, although the percentage of tumor-infiltrating CD3$^+$ T cells was increased to 11.0% of CD45$^+$ lymphocytes after anti-PD-L1 antibody treatment, NP$_{Tyr-αPD-L1}$ group showed higher tumor infiltration of CD3$^+$ T cells (13.9% of CD45$^+$ lymphocytes). Combining anti-PD-L1 antibody with NP$_{Tyr-CXCL9}$ that expressed CXCL9 significantly increased the tumor infiltration of CD3$^+$ T cells (17.2% of CD45$^+$ lymphocytes in NP$_{Tyr-CXCL9}$ & anti-PD-L1 group). Importantly, NP$_{Tyr-C9AP}$ treatment increased the percentages of tumor-infiltrating CD3$^+$ T cells to 21.7% of CD45$^+$ lymphocytes due to the synergistic effects of intratumoral expression of CXCL9 and αPD-L1. In addition, the numbers of CD69-positive CD8$^+$ T cells in melanoma of NP$_{Tyr-αPD-L1}$, NP$_{Tyr-CXCL9}$ & anti-PD-L1 or NP$_{Tyr-C9AP}$ groups respectively reached 3.71, 6.34 or 7.56 × 10$^5$ cells per gram tumor, which was 1.3-, 2.2- and 2.6- fold of that in the anti-PD-L1 group (Fig. 6f, Supplementary Fig. 24a). The percentages of granzyme B-, perforin-, IFN-γ-positive CD8$^+$ T cells in melanoma were also significantly increased to 53.1%, 49.5% and 34.9% of CD8$^+$ T cells by NP$_{Tyr-C9AP}$ treatment when compared to anti-PD-L1 group (Fig. 6g–i, Supplementary Fig. 24b–d). The ratio of CD8$^+$ T cells/Tregs in melanoma also showed similar increase

after NP$_{Tyr-C9AP}$ treatment (Supplementary Fig. 25). Collectively, NP$_{Tyr-C9AP}$ that locally coexpressed CXCL9 and αPD-L1 in melanoma was more potent in enhancing the intratumoral infiltration and activation of T cells than the systemic injection of anti-PD-L1 antibody, thereby inducing superior antimelanoma effects.

## NP$_{Tyr-C9AP}$ induces anti-tumor effects in different melanoma models

To ensure the reproducibility, we next explored the therapeutic efficacy of NP$_{Tyr-C9AP}$ in additional melanoma models. The expression of CXCL9 and αPD-L1 in different melanoma cells, including B16-F10, Clone M-3 and YUMM1.7 cells, were firstly detected after NP$_{Tyr-C9AP}$ treatment. As shown in Supplementary Fig. 26, CXCL9 concentrations in the culture supernatants of NP$_{Tyr-C9AP}$-transfected B16-F10, Clone M-3 and YUMM1.7 cells ranged from 0.25 to 0.53 ng ml$^{-1}$, while αPD-L1 concentrations ranged from 1.00–2.26 ng ml$^{-1}$. After the *i.v.* injection of NP$_{Tyr-C9AP}$ into mice bearing different melanomas, the intratumoral concentrations of CXCL9 and αPD-L1 were 11.11 and 36.82 ng g$^{-1}$ in NP$_{Tyr-C9AP}$-treated Clone M-3 melanoma (Fig. 7a, b), and were 15.06 and 54.79 ng g$^{-1}$ in NP$_{Tyr-C9AP}$-treated YUMM1.7 melanoma (Fig. 7f, g). These results indicated that NP$_{Tyr-C9AP}$ could achieve high expression of CXCL9 and αPD-L1 in different melanomas. Then, NP$_{Tyr-C9AP}$ was intravenously injected into Clone M-3 and YUMM1.7 melanoma-bearing mice

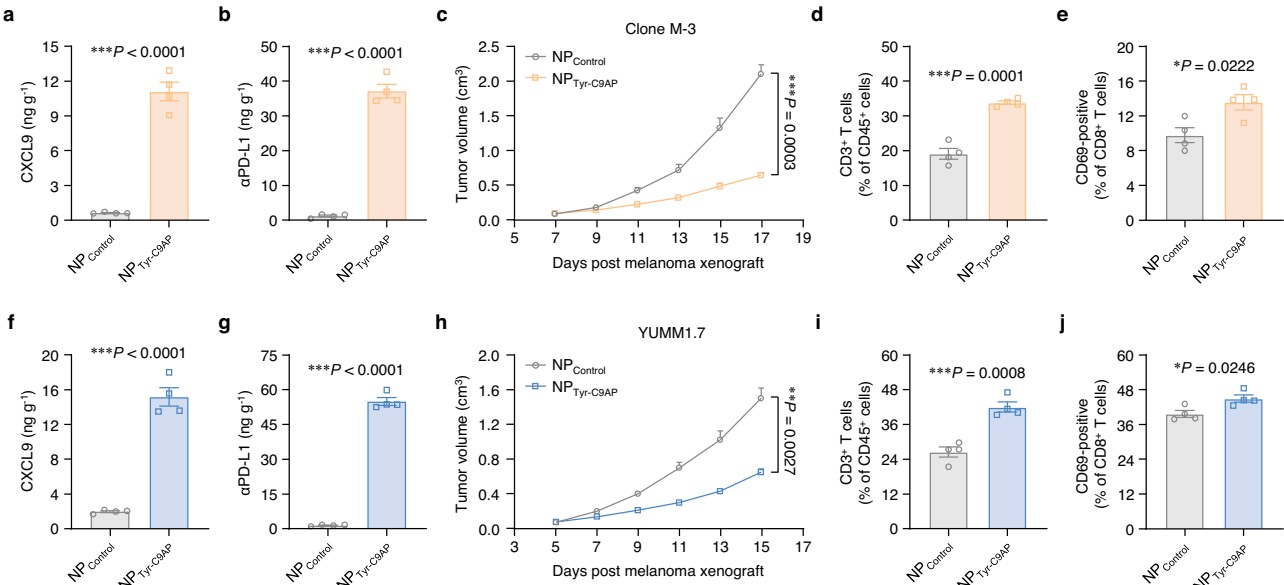

**Fig. 7 | NP_Tyr-C9AP induces enhanced therapeutic effects against different melanomas. a, b** Concentrations of CXCL9 (**a**) and αPD-L1 (**b**) in Clone M-3 melanoma after the *i.v.* injection of NP_Tyr-C9AP or NP_Control (*n* = 4 mice per group). **c** Tumor growth curves of the Clone M-3 melanoma-bearing mice treated with NP_Tyr-C9AP or NP_Control (*n* = 6 mice per group). **d, e** Percentages of CD3+ T cells (**d**) or CD69-positive CD8+ T cells (**e**) in Clone M-3 melanoma after the treatments. **f, g** Concentrations of CXCL9 (**f**) and αPD-L1 (**g**) in YUMM1.7 melanoma after the *i.v.* injection of NP_Tyr-C9AP or NP_Control (*n* = 4 mice per group). **h** Tumor growth curves of the YUMM1.7 melanoma-bearing mice treated with NP_Tyr-C9AP or NP_Control (*n* = 6 mice per group). **i, j** Percentages of CD3+ T cells (**i**) or CD69-positive CD8+ T cells (**j**) in YUMM1.7 melanoma after the treatments. The data are shown as the means ± SEM of *n* = 4 biologically independent mice (**a, b, d–g, i, j**), or 6 biologically independent mice (**c, h**). Statistical data were analyzed by two-sided Student's *t*-test (**a, b, d–g, i, j**), two-way ANOVA with the Greenhouse–Geisser correction (**c, h**). *P < 0.05; **P < 0.01; ***P < 0.001. Source data are provided as a Source Data file.

every other day for five injections to investigate the therapeutic efficacy. As shown in Fig. 7c and h, NP_Tyr-C9AP treatment significantly inhibited the growth of Clone M-3 and YUMM1.7 melanoma, respectively achieving 69.5% and 56.0% tumor growth inhibition rates when compared to that of NP_Control group. The final tumor weights of Clone M-3 and YUMM1.7 melanoma were correspondingly reduced by NP_Tyr-C9AP treatment (Supplementary Fig. 27). Moreover, flow cytometry analysis showed that the percentages of tumor-infiltrating CD3+ T cells and CD69-positive CD8+ T cells were significantly increased after NP_Tyr-C9AP treatment (Fig. 7d, e, i, j, Supplementary Fig. 28). Collectively, these results demonstrated that NP_Tyr-C9AP could induce anti-tumor effects in different melanoma models via coexpressing CXCL9 and αPD-L1 to enhance intratumoral infiltration and activation of T cells.

### Engineering NP_Sur-C9AP to enhance T-cell responses in multiple tumors

Since the Tyr promoter-driven gene nanomedicine NP_Tyr-C9AP has been proven to enhance antitumor T-cell responses by melanoma-specific coexpression of CXCL9 and αPD-L1, we explored engineering a universal gene nanomedicine that enables coexpression of CXCL9 and αPD-L1 specifically in various tumor cells for enhanced immunotherapy against multiple tumors. Survivin (Sur), a member of the inhibitor of apoptosis (IAP) proteins that suppresses apoptosis and regulates cell division, is highly expressed in most cancers but not in most normal tissues[42]. In view of the high tumor specificity of Sur, the promoter of Sur was used as the universal tumor-specific promoter to replace the Tyr promoter of pTyr-C9AP to construct pSur-C9AP (Supplementary Fig. 1d). The resulting pSur-C9AP was encapsulated into the same NPs to prepare the universal gene nanomedicine NP_Sur-C9AP (Fig. 8a). Similar to NP_Tyr-C9Ap, NP_Sur-C9AP was supposed to deliver pSur-C9AP into both tumor cells and other normal cells, but the Sur promoter of pSur-C9AP would only drive the coexpression of CXCL9 and αPD-L1 specifically in tumor cells. To validate the tumor specificity of NP_Sur-C9AP, we transfected four types of tumor cells (B16-F10, CT26, PancO2, 4T1 cells) and

four types of primary normal cells derived from similar tissues (mouse skin, colon, pancreas and breast cells) with NP_Sur-C9AP. As demonstrated in Fig. 8b, NP_Sur-C9AP induced high expression of CXCL9 and αPD-L1 proteins in the four types of tumor cells (B16-F10, CT26, PancO2, and 4T1 cells), but hardly induced the expression of CXCL9 and αPD-L1 proteins in the four types of primary normal cells derived from mouse skin, colon, pancreas, and breast tissues. In addition, after intravenously injecting NP_Sur-C9AP into mice bearing B16-F10 melanoma, the intratumoral concentrations of CXCL9 and αPD-L1 proteins were significantly increased to 11.76 and 55.65 ng g⁻¹; by contrast, the expression of CXCL9 and αPD-L1 proteins was only moderately upregulated in liver and spleen and was not increased in heart, lung and kidney (Fig. 8c, d). Similarly, high expression of CXCL9 and αPD-L1 proteins were detected only in the tumor tissues of mice bearing colorectal (CT26), pancreatic (PancO2) and breast (4T1) tumors after NP_Sur-C9AP treatment (Supplementary Fig. 29). Thus, NP_Sur-C9AP was proven to be a universal gene nanomedicine for coexpressing CXCL9 and αPD-L1 specifically in different tumors, which could be used to enhance T-cell recruitment and activation in multiple tumors.

Then, NP_Sur-C9AP was intravenously injected into B16-F10 melanoma-bearing mice every other day for five injections to investigate its therapeutic efficacy. As shown in Fig. 8e, NP_Sur-C9AP treatment significantly inhibited the growth of B16-F10 melanoma, achieving an 84.4% tumor growth inhibition rate when compared to that of the NP_Control group. The final tumor weights of B16-F10 melanoma also demonstrated the enhanced antimelanoma efficacy of NP_Sur-C9AP (Fig. 8f). In addition, flow cytometry analysis verified that the percentage of tumor-infiltrating CD3+ T cells was increased to 16.8% of CD45+ lymphocytes in melanoma after NP_Sur-C9AP treatment (Fig. 8g, Supplementary Fig. 30a). The number of CD3+ T cells in NP_Sur-C9AP-treated melanoma reached $1.65 \times 10^6$ cells per gram tumor (Supplementary Fig. 30b), indicating that NP_Sur-C9AP effectively recruited T cells into melanoma. Next, the subpopulations and phenotypes of tumor-infiltrating T cells were also analyzed. As shown in Fig. 8h, i and

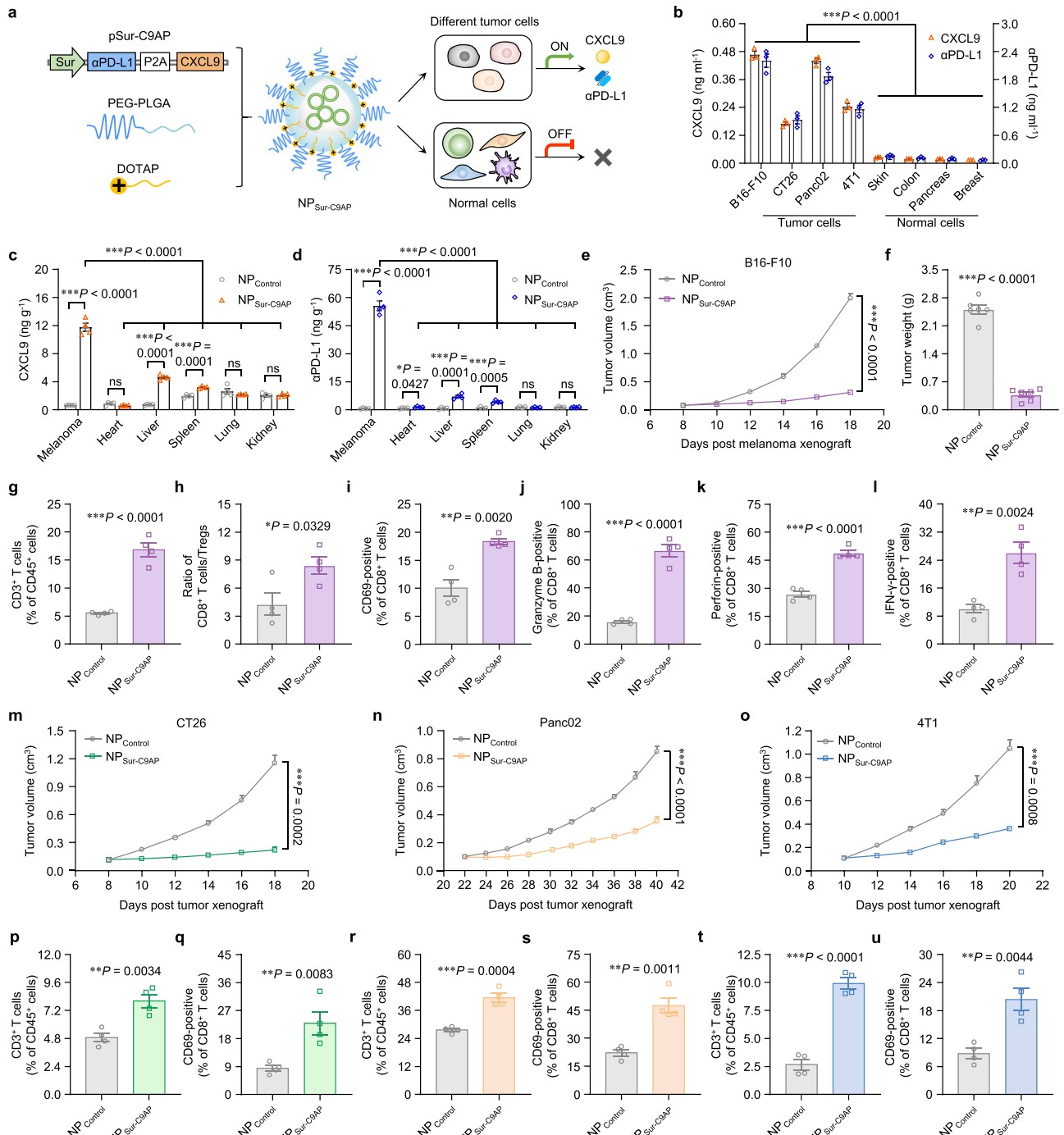

**Fig. 8 | NP_Sur-C9AP enhances T-cell recruitment and activation in multiple tumors for enhanced immunotherapy. a** Scheme of constructing NP_Sur-C9AP by encapsulating pSur-C9AP with PEG-PLGA and DOTAP. NP_Sur-C9AP can deliver pSur-C9AP into different tumor cells and other normal cells, but the Sur promoter of pSur-C9AP will only turn on in tumor cells to drive the coexpression of CXCL9 and αPD-L1. **b** Concentrations of CXCL9 and αPD-L1 proteins in the culture supernatants of four types of tumor cells (B16-F10, CT26, Panc02, 4T1 cells) and four types of primary normal cells (mouse skin, colon, pancreas and breast cells) after NP_Sur-C9AP transfection. **c, d** Concentrations of CXCL9 (**c**) and αPD-L1 (**d**) proteins in melanoma and normal organs of the NP_Sur-C9AP-treated mice ($n = 4$ mice per group). **e, f** Tumor growth curves (**e**) and final tumor weights (**f**) of the B16-F10 melanoma-bearing mice treated with NP_Sur-C9AP or NP_Control ($n = 6$ mice per group). **g, h** Percentages of CD3+ T cells (**g**) or the ratio of CD8+ T cells/Tregs (**h**) in B16-F10 melanoma after the treatments ($n = 4$ mice per group). **i–l** Percentages of CD69- (**i**), granzyme B- (**j**), perforin- (**k**) and IFN-γ- (**l**) positive CD8+ T cells in B16-F10 melanoma ($n = 4$ mice per group). **m–o** Tumor growth curves of CT26 (**m**), Panc02 (**n**) and 4T1 (**o**) tumor-bearing mice treated with NP_Sur-C9AP or NP_Control ($n = 6$ mice per group). **p–u** Percentages of CD3+ T cells or CD69-positive CD8+ T cells of CT26 (**p, q**), Panc02 (**r, s**) and 4T1 (**t, u**) tumors after the treatments. The data are shown as the means ± SEM of $n = 3$ biologically independent samples (**b**), 4 biologically independent mice (**c, d, g–l, p–u**), or 6 biologically independent mice (**e, f, m–o**). Statistical data were analyzed by one-way ANOVA with Tukey's multiple comparison test (**b–d**), two-way ANOVA with the Greenhouse–Geisser correction (**e, m–o**), two-sided Student's $t$-test (**c, d, f–l, p–u**). *$P < 0.05$; **$P < 0.01$; ***$P < 0.001$; ns indicates no significant difference. Source data are provided as a Source Data file.

Supplementary Fig. 31a–c, the ratio of CD8$^+$ T cells/Tregs was increased to 8.42 and the percentage of CD69-positive CD8$^+$ T cells was increased to 18.1% in NP$_{Sur-C9AP}$-treated melanoma, demonstrating that NP$_{Sur-C9AP}$ efficiently enhanced the tumor infiltration and activation of CD8$^+$ T cells. Correspondingly, the percentages of granzyme B-, perforin- and IFN-γ-positive CD8$^+$ T cells were increased to 66.9%, 49.0% and 26.0%, respectively (Fig. 8j–l and Supplementary Fig. 31d–f). These results indicated that NP$_{Sur-C9AP}$ could effectively enhance T-cell infiltration and activation for enhanced melanoma immunotherapy.

The therapeutic efficacy of NP$_{Sur-C9AP}$ was further investigated in CT26, Panc02 and 4T1 tumors after being intravenously injected into these tumor mouse models. As shown in Fig. 8m–o and Supplementary Fig. 32, NP$_{Sur-C9AP}$ treatment markedly inhibited the growth of CT26, Panc02 and 4T1 tumors, whose tumor growth inhibition rates were 81.0%, 56.3% and 66.5%, respectively, exhibiting the superior efficacy of NP$_{Sur-C9AP}$ in different tumors. Moreover, flow cytometry analysis showed that the percentages of CD3$^+$ T cells in CT26, Panc02 and 4T1 tumors achieved 1.65-, 1.50- and 3.67-fold increases after NP$_{Sur-C9AP}$ treatment (Fig. 8p, r, t, Supplementary Fig. 33a–c). Compared to that of the NP$_{Control}$ group, the percentages of CD69-positive CD8$^+$ T cells in the NP$_{Sur-C9AP}$-treated CT26, Panc02 and 4T1 tumors were also significantly increased (Fig. 8q, s, u and Supplementary Fig. 33d–f), indicating that NP$_{Sur-C9AP}$ enhanced the activation of CD8$^+$ T cells in these three tumors. Taken together, these results demonstrated that NP$_{Sur-C9AP}$ could be used as a universal gene nanomedicine to enhance T-cell recruitment and activation in multiple tumors for enhanced immunotherapy via tumor-specific coexpression of CXCL9 and αPD-L1.

## Discussion

αPD-L1 therapy is successful but still faces unsatisfactory therapeutic benefits due to the lack of tumor-infiltrating T cells and the insufficient tumor enrichment of systemically injected αPD-L1 inhibitors[9–11]. In addition, systemic injection of αPD-L1 inhibitors can cause side effects[17,18]. Although some strategies were reported to improve αPD-L1 therapy by increasing either tumor-infiltrating T cells or intratumoral αPD-L1 inhibitors[12–16], there was still no strategy can actively recruit T cells into tumors and meanwhile achieve tumor-specific delivery of αPD-L1 inhibitors to specifically eliminate the inhibition of tumor-infiltrating T cells. Herein, we developed tumor-specific gene nanomedicines that utilize tumor-specific promoters to drive tumor cells to coexpress CXCL9 and αPD-L1 in situ, which effectively recruited T cells into tumors and enhanced the activation of tumor-infiltrating T cells. In addition, our nanomedicines-mediated local expression of αPD-L1 showed superior efficacy and safety when compared to systemic injection of anti-PD-L1 antibody. Overall, we provided a strategy using nanomedicines to engineer diverse tumor cells into the producers of T-cell chemokines and αPD-L1 inhibitors, which skillfully addressed the two hurdles of αPD-L1 therapy.

The concept of locally expressing scFv to block PD-L1/PD-1, such as using CAR-T cells or intratumorally injecting oncolytic virus to express αPD-1/αPD-L1 scFv has been reported[43,44], but these strategies are quite different from ours. Using CAR-T cells to express αPD-L1 scFv can improve the efficacy but cannot really achieve local expression of scFv in tumors; intratumoral injection of oncolytic virus is not applicable for all metastatic tumors. And, these strategies cannot address the insufficient CAR-T or T-cell infiltration. By contrast, our strategy not only addressed the low enrichment and potential toxicities caused by systemic injection of αPD-L1 antibody, but also addressed the insufficient T-cell infiltration. One potential concern is that αPD-L1 scFv lacks antibody-dependent cellular cytotoxicity or phagocytosis (ADCC/ADCP) functions which may reduce its antitumor efficacy in comparison to αPD-L1 antibody. However, αPD-L1 therapy mainly relies on blocking PD-L1 to enhance T-cell activation[45], and the approved atezolizumab that removes the ADCC/ADCP functions has shown good efficacy in treating various tumors[46–49]. In addition, αPD-L1

scFv can more easily penetrate the tumor tissues due to its smaller size than αPD-L1 antibody[50].

Currently, nanoparticles have been proven to be promising delivery systems of nucleic acid drugs, lipid nanoparticles (LNPs)-delivered siRNA or mRNA vaccines, such as Patisiran, Tozinameran and Elasomeran, have been approved[51]. However, nanoparticles are unable to selectively deliver drugs into one specific type of organ or cell, and preferentially accumulate in liver, which hinders drug delivery to treat diseases outside the liver[52,53]. Regulating the properties of nanoparticles by passive, active and endogenous targeting mechanisms has been reported to improve the organ selectivity of drug delivery[54,55]. In addition, cell type-specific promoter-induced gene expression has been reported[56], for example, our group used nanoparticles to deliver CD68 or HSP70 promoter-modified Cas9 plasmid to achieve macrophage-specific or light-responsive gene editing[57,58]. Combining nanoparticles and tumor-specific promoters enables us to drive gene expression, such as the coexpression of CXCL9 and αPD-L1, in tumor cells for enhanced cancer immunotherapy.

In this study, we devised two tumor-specific gene nanomedicines, NP$_{Tyr-C9AP}$ and NP$_{Sur-C9AP}$. They were 100 nm with a PEG shell that was suitable for prolonged blood circulation and tumor accumulation[38,39]. In addition, they can be easily internalized by tumor cells due to their positive charge[59]. After systematic administration, NP$_{Tyr-C9AP}$ with Tyr promoter were demonstrated to induce specific coexpression of CXCL9 and αPD-L1 in multiple melanoma models, while NP$_{Sur-C9AP}$ with Sur promoter specifically coexpressed CXCL9 and αPD-L1 in different tumors. Both of them significantly enhanced the anti-tumor effects of T cells. Furthermore, because our nanomedicines were fabricated with PEG-PLGA and DOTAP which are FDA-approved pharmaceutic adjuvants, no body weight loss, liver toxicities and injuries to major organs were observed after the treatments, suggesting that our tumor-specific gene nanomedicines were safe. But the safety risks of plasmids, including immunogenicity and potential gene insertion[60,61], need to be investigated before clinical use.

In summary, we have developed tumor-specific gene nanomedicines to specifically drive different tumor cells to coexpress CXCL9 and αPD-L1, which provide a solution for addressing both the lack of tumor-infiltrating T cells and the insufficient tumor enrichment of αPD-L1 inhibitors in ICB therapy. Because the lack of tumor antigen presentation is not a major hurdle of αPD-L1 therapy, and αPD-L1 therapy can potentially improve the tumor antigen presentation of dendritic cells (DCs)[62,63], the tumor antigen release and DCs priming were not considered when designing our nanomedicines. Some strategies targeting tumor antigen presentation can be combined to further enhance the efficacy of our strategy.

## Methods
### Ethical statement
This research complies with all relevant ethical regulations. All the experiments in this research were approved by the Animal Care and Use Committee at South China University of Technology (SCUT) (official approval number: 2019012). According the guidelines of the Institutional Animal Care and Use Committee (IACUC) and the local authorities (the Animal Care and Use Committee at SCUT), the tumors in mice are allowed to grow to 2 cm$^3$ as long as the mouse remains otherwise healthy. In some cases, this limit has been exceeded the last day of measurement and the mice were immediately euthanized.

### Materials and reagents
PEG-PLGA for the preparation of gene nanomedicines was methoxy PEG$_{5K}$-PLGA$_{11K}$, which was synthesized as previously reported. DOTAP was purchased from Corden Pharma (Liestal, Switzerland). Mouse CXCL9 ELISA kit was from Abcam (MA, USA). ELISA reagents, including

mouse recombinant PD-L1 protein and HRP-conjugated anti-6×His tag IgG, for αPD-L1 quantification were from Sino Biological (Beijing, China). αPD-L1 protein standard was produced by transfecting HEK-293F cells with pcDNA3.1-αPD-L1 scFv and purified using Ni Sepharose 6 Fast Flow (Cytiva, MA, USA). Transwell plates (24-well) were from Corning (NY, USA). Precision Count Beads were from BioLegend (CA, USA). Antibodies for western blots are as follows: anti-6×His tag (ab18184) was from Abcam, anti-β-actin (BL005B), HRP conjugated-goat anti-mouse IgG (BL001A) and HRP conjugated-goat anti-rabbit IgG (BL003A) were from Biosharp (Anhui, China). Antibodies for flow cytometry are as follows: Alexa Fluor 647 anti-6×His tag (clone: J099B12), Brilliant Violet 510 anti-mouse CD45 (clone: 30-F11), APC anti-mouse CD3 (clone:145-2C11), APC/Cyanine7 anti-mouse CD4 (clone: GK1.5), Brilliant Violet 785 anti-mouse CD8a (clone: 53-6.7), FITC anti-mouse CD8a (clone: 53-6.7), Brilliant Violet 605 anti-mouse CD69 (clone:H1.2F3), Alexa Fluor 488 anti-mouse NK1.1 (clone: PK136), FITC anti-mouse NKp46 (clone: 29A1.4), FITC anti-mouse CD11b (clone: M1/70), APC/Cyanine7 anti-mouse Gr-1 (clone: RB6-8C5), Brilliant Violet 650 anti-mouse F4/80 (clone: BM8), PE/Dazzle594 anti-mouse CD80 (clone: 16-10A1), PE anti-mouse CD206 (clone: C068C2), FITC anti-mouse CD25 (clone: 3C7), Brilliant Violet 421 anti-mouse IFN-γ (clone: XMG1.2), FITC anti-human/mouse Granzyme B (clone: QA16A02), PE anti-mouse Perforin (clone: S16009B) and PE anti-mouse Foxp3 (clone: MF-14) were from BioLegend.

## Cells and animals

The mouse cell lines, including B16-F10, CT26, Panc02, 4T1, C2C12, NIH/3T3, DC2.4, RAW264.7, Clone M-3 (Cloudman S91), and YUMM1.7 cells were from the American Type Culture Collection (ATCC). The B16-F10-OVA-EGFP cell line was constructed by infecting B16-F10 cells with a lentivirus encoding OVA and EGFP. Primary CD8+ T cells or OVA-specific CD8+ T cells were sorted from the spleens of C57BL/6 or OT-1 transgenic mice using a CD8a (Ly2) Microbeads Kit (Miltenyi Biotec., CA, USA). Mouse primary skin, colon, pancreas and breast cells were isolated by digesting the skin, colon, pancreas and breast tissue using 0.2% type-IV collagenase (Sangon, Shanghai, China) and 0.05% dispase (Sangon). The human cell line HEK-293F was obtained from the National Collection of Authenticated Cell Cultures (Shanghai, China). All T cells were cultured in Roswell Park Memorial Institute (RPMI) 1640 medium with 10% FBS, 1 mM sodium pyruvate, 10 mM HEPES, 55 μM β-mercaptoethanol and 10 ng ml$^{-1}$ recombinant murine IL-2 (Pepro-tech) and were activated by soluble anti-mouse CD3 antibody (clone: 145-2C11, BioLegend) and anti-mouse CD28 antibody (clone: 37.51, BioLegend) according to the manufacturer's instructions before use. HEK-293F cells were cultured in the SIMM 293-TII medium from Sino Biological. The other cells were cultured in Dulbecco's modified Eagle's medium (DMEM) with 10% FBS. All cells were maintained in a humidified atmosphere containing 5% $CO_2$ at 37 °C and were regularly assessed for mycoplasma contamination.

Female C57BL/6 (6−10 weeks) and BALB/c (6−10 weeks) mice were purchased from Silaike Jingda Laboratory Animal Co., Ltd. (Hunan, China). OT-1 transgenic mice (C57BL/6-Tg (TcraTcrb)1100Mjb/J) were gifted from Prof. Tian-Meng Sun of Jilin University. All mice were maintained at the specific pathogen-free (SPF) facility of the SCUT and received care in compliance with the *Guide for the Care and Use of Laboratory Animals*. All mice were housed in temperatures 20−25 °C, humidity 30−70% and a 12 h light/12 h dark cycle.

## Plasmid construction

The *αPD-L1* sequence (795 bps) was obtained from the disclosed patent US20100169296A1, and the *CXCL9* sequence (378 bps) was obtained from the CCDS database (CCDS39152.1). The sequences of the Tyr promoter (546 bps) and Sur promoter (260 bps) were modified from previous works[32,33]. All DNA sequences are provided in Supplementary

Table 1. The DNA fragments of the Tyr promoter, Sur promoter and αPD-L1-P2A-CXCL9 were first synthesized by Sangon. Then, the αPD-L1-P2A-CXCL9 fragment was inserted into the NcoI and SalI sites of the pUC57 backbone plasmid (Sangon), and the Tyr promoter was inserted between the EcoRI and NcoI sites (upstream of αPD-L1-P2A-CXCL9) to construct the pTyr-C9AP plasmid (Supplementary Fig. 1a). pTyr-C9AP containing the EGFP reporter gene was constructed by inserting the T2A-EGFP fragment into the SalI site (downstream of CXCL9) (Supplementary Fig. 1b). The pTyr-CXCL9 or pTyr-αPD-L1 plasmid was constructed by replacing the αPD-L1-P2A-CXCL9 fragment of pTyr-C9AP with the CXCL9 or αPD-L1 fragment, respectively (Supplementary Fig. 1c). The pSur-C9AP plasmid was constructed by replacing the Tyr promoter of pTyr-C9AP with the Sur promoter (Supplementary Fig. 1d).

## Preparation and characterization of gene nanomedicines

NP$_{Tyr-C9AP}$ was prepared by encapsulating pTyr-C9AP with PEG$_{5K}$-PLGA$_{11K}$ and DOTAP using a double emulsion method. In brief, pTyr-C9AP (100 μg in 25 μl of DNase/RNase-free water) was emulsified in 500 μl of chloroform containing 1 mg DOTAP and 25 mg PEG$_{5K}$-PLGA$_{11K}$ for 1 min (80 watts, 4 °C) using a VCX130 Sonicator (Sonics & Materials, CS, USA). Then, 5 ml of DNase/RNase-free water was added to the primary emulsion, and the mixture was further emulsified to form a water-oil-water emulsion. Finally, NP$_{Tyr-C9AP}$ was obtained by removing chloroform from the emulsion using an RV10 rotary eva-porator (IKA, Staufen, Germany). NP$_{Control}$, NP$_{Tyr-CXCL9}$, NP$_{Tyr-αPD-L1}$, NP$_{Tyr-C9AP}$ or NP$_{Sur-C9AP}$ was prepared using the same method to encapsulate pUC57, pTyr-CXCL9, pTyr-αPD-L1, pTyr-C9AP or pSur-C9AP, respectively. The diameter and zeta potential of these nano-medicines were characterized by a Zetasizer Nano ZS90 (Malvern, Worcestershire, UK). Nanomedicine morphology was visualized by a Talos L120C TEM (Thermo Fisher, MA, USA).

## Cellular uptake of NP$_{Tyr-C9AP}$ by different cells

pTyr-C9AP was first labeled with Cy5 using a *Label* IT Nucleic Acid Labeling Kit (Mirus Bio., WI, USA) and then encapsulated to prepare Cy5-labeled NP$_{Tyr-C9AP}$. B16-F10, CT26, Panc02, 4T1, C2C12, NIH-3T3, DC2.4 and RAW264.7 cells were seeded in 24-well plates (5 × 10$^4$ cells per well) and then incubated with Cy5-labeled NP$_{Tyr-C9AP}$ at a final concentration of 1 μg ml$^{-1}$ pTyr-C9AP for 4 h. For CLSM imaging, the cells were fixed with 4% paraformaldehyde, and the F-actin and nucleus were labeled with Alexa Fluor 488 Phalloidin (Invitrogen, MA, USA) and DAPI (Biosharp, Anhui, China) before visualization by an LSM880 (Zeiss, Oberkochen, Germany). The confocal data were collected using ZEISS ZEN2 (black edition) software and analyzed with ZEISS ZEN2 (blue edition) software. For flow cytometry analysis, the cells were collected by trypsinization and detected by FACSCelesta (BD Bios-ciences, CA, USA). The flow cytometry data were collected using the BD FACS Diva software v8.0.1.1 and further analyzed by FlowJo soft-ware v10.0.7 (BD Biosciences).

## Biodistribution of NP$_{Tyr-C9AP}$ in vivo

The biodistribution of Cy5-labeled NP$_{Tyr-C9AP}$ was analyzed in a bilat-eral tumor model. Because B16-F10 cells can produce abundant mel-anin that interferes with the in vivo detection of Cy5 signal, the bilateral tumor model was established by subcutaneously inoculating 1.5 × 10$^6$ YUMM1.7 melanoma cells into the right flank of C57BL/6 mice and 5 × 10$^6$ Panc02 pancreatic tumor cells into the left flank. When the melanoma volume reached 400−500 mm$^3$, the mice were intrave-nously injected Cy5-labeled NP$_{Tyr-C9AP}$ at a dose of 1 mg Cy5-labeled pTyr-C9AP per kg body weight. Twelve hours later, the YUMM1.7 melanoma, Panc02 pancreatic tumor and normal organs were col-lected, and were analyzed using IVIS Lumina III Living Image system (PerkinElmer, MA, USA). The data were collected using Living image software v4.4 (PerkinElmer).

## Tumor-specific analysis of gene nanomedicines

For determination of whether $NP_{Tyr-C9AP}$ could drive the coexpression of CXCL9 and αPD-L1 specifically in melanoma cells in vitro, B16-F10, CT26, Panc02, 4T1, C2C12, NIH/3T3, DC2.4 and RAW264.7 cells were seeded in 24-well plates ($5 \times 10^4$ cells per well) and then incubated with $NP_{Tyr-C9AP}$ or $NP_{Control}$ at a final concentration of 1 μg ml$^{-1}$ pTyr-C9AP or pUC57 for 6 h. Forty-eight hours after transfection, the fluorescence images of cells were collected using the NIS-Elements Viewer software v5.21 of Eclipse Ti2-E fluorescence microscope (Nikon, Tokyo, Japan), and the cells were collected to analyze the percentages of EGFP-positive cells by flow cytometry or to analyze CXCL9 and αPD-L1 mRNA expression by qRT–PCR using primers listed in Supplementary Table 2. Seventy-two hours after transfection, the culture supernatants were collected to detect the concentrations of CXCL9 and αPD-L1 by ELISA. The CXCL9 concentration was quantified by mouse CXCL9 ELISA kit. For quantifying αPD-L1 concentration, mouse recombinant PD-L1 (5 μg ml$^{-1}$) was coated on ELISA plates followed by blocking with no protein blocking solution (Sangon). Samples and αPD-L1 protein standards were then incubated in the coated plates, and HRP-conjugated anti-6×His tag IgG (1 μg ml$^{-1}$) was added after washing. After chemiluminescent color reaction, the 450 nm absorbance was detected.

For analysis of the melanoma specificity of $NP_{Tyr-C9AP}$ in vivo, the bilateral tumor mice were established by subcutaneously inoculating $3 \times 10^5$ B16-F10 cells into the right flank of C57BL/6 mice and $5 \times 10^6$ Panc02 cells into the left flank. The mice were randomly divided into two groups (n = 4 mice per group) when the volumes of B16-F10 melanoma and Panc02 pancreatic tumors reached 50–100 mm$^3$. Then, the mice bearing bilateral tumors were intravenously injected with $NP_{Tyr-C9AP}$ or $NP_{Control}$ every other day for three injections. The injection doses were 1 mg pTyr-C9AP (with EGFP reporter gene) or pUC57 per kg body weight. Seventy-two hours after the last injection, the B16-F10 melanoma, Panc02 pancreatic tumor, liver, lung, spleen, kidney and heart were isolated to analyze the percentages of EGFP-positive cells by flow cytometry, or were lysed to detect CXCL9 and αPD-L1 concentrations in different tissues by ELISA.

For analysis of the tumor specificity of $NP_{Sur-C9AP}$ in vitro, B16-F10, CT26, Panc02, 4T1, primary skin, colon, pancreas and breast cells were seeded in 24-well plates ($5 \times 10^4$ cells per well) and then incubated with $NP_{Sur-C9AP}$ or $NP_{Control}$ at a final concentration of 1 μg ml$^{-1}$ pSur-C9AP or pUC57 for 6 h. Seventy-two hours after transfection, the culture supernatants were collected to detect the concentrations of CXCL9 and αPD-L1 by ELISA.

For analysis of the tumor specificity of $NP_{Sur-C9AP}$ in vivo, multiple tumor models were established. Mice bearing B16-F10 melanoma were established by subcutaneously inoculating $3 \times 10^5$ B16-F10 cells into the right flank of C57BL/6 mice. Mice bearing CT26 colorectal tumors were established by subcutaneously inoculating $1 \times 10^6$ CT26 cells into the right flank of BALB/c mice. Mice bearing Panc02 pancreatic tumors were established by subcutaneously inoculating $3 \times 10^6$ Panc02 cells into the right flank of C57BL/6 mice. Mice bearing 4T1 orthotopic breast tumors were established by injecting $5 \times 10^5$ 4T1 cells into the mammary fat pads of BALB/c mice. The mice bearing different tumors were randomly divided into two groups (n = 4 mice per group) when the tumor volumes reached 100–200 mm$^3$. Then, the mice were intravenously injected with $NP_{Sur-C9AP}$ or $NP_{Control}$ every other day for three injections. The injection doses were 1 mg pSur-C9AP or pUC57 per kg body weight. Seventy-two hours after the last injection, the B16-F10, CT26, Panc02, 4T1 tumors, liver, lung, spleen, kidney and heart were collected and lysed to detect CXCL9 and αPD-L1 concentrations in different tissues by ELISA.

## Analysis of CXCL9 expression and function in vitro

B16-F10 cells were seeded in 24-well plates ($5 \times 10^4$ cells per well) and then incubated with PBS, $NP_{Control}$, $NP_{Tyr-CXCL9}$ or $NP_{Tyr-C9AP}$ for 6 h.

The transfection doses were 1 μg ml$^{-1}$ pUC57, pTyr-CXCL9 or pTyr-C9AP. Forty-eight hours after transfection, the cells were collected to analyze CXCL9 mRNA expression by qRT–PCR. Seventy-two hours after transfection, the culture supernatants were collected to detect CXCL9 protein concentration by ELISA.

The function of CXCL9 was analyzed by tracking T-cell chemotaxis in a transwell migration assay. Primary CD8$^+$ T cells were sorted and activated as mentioned above. Activated CD8$^+$ T cells were labeled with DiI dye (Meilunbio, Liaoning, China) and cultured in the upper chamber of transwell plates at a density of $1 \times 10^6$ cells per well. After the culture supernatants of the PBS-, $NP_{Control}$-, $NP_{Tyr-CXCL9}$- or $NP_{Tyr-C9AP}$-transfected B16-F10 cells were added in the lower chamber, the migration of DiI-labeled CD8$^+$ T cells was observed by an LSM880. Three hours after incubation, CD8$^+$ T cells recruited to the lower chamber were collected and labeled with FITC anti-mouse CD8a antibody. Then, 10 μl of Precision Count Beads was added to each sample before being analyzed by flow cytometry. The numbers of recruited CD8$^+$ T cells were calculated using the following formula: $X = Y \times Z$ ($X$, absolute count of CD8$^+$ T cells; $Y$, ratio of CD8$^+$ T cells/counting beads; $Z$, absolute count of counting beads).

## Analysis of the expression and function of αPD-L1 in vitro

B16-F10 cells were seeded in 24-well plates ($5 \times 10^4$ cells per well) and incubated with PBS, $NP_{Control}$, $NP_{Tyr-\alpha PD-L1}$ or $NP_{Tyr-C9AP}$ for 6 h. The transfection doses were 1 μg ml$^{-1}$ pUC57, pTyr-αPD-L1 or pTyr-C9AP. Forty-eight hours after transfection, the cells were collected to analyze αPD-L1 mRNA expression by qRT–PCR. Seventy-two hours after transfection, the culture supernatants were collected to detect αPD-L1 protein concentration by ELISA. The αPD-L1 expression in the culture supernatants and the reference β-actin expression in the cell lysates were also analyzed by western blot using primary antibodies (anti-6×His tag, 1:10000; anti-β-actin, 1:4000) and secondary antibodies (HRP-conjugated goat anti-mouse or rabbit IgG, 1:5000).

For analysis of the capacity of αPD-L1 to bind PD-L1, B16-F10 cells were stimulated with 20 ng ml$^{-1}$ IFN-γ (Peprotech) for 24 h. Then, the culture supernatants of the B16-F10 cells transfected with PBS, $NP_{Control}$, $NP_{Tyr-\alpha PD-L1}$ or $NP_{Tyr-C9AP}$ were incubated with IFN-γ-stimulated B16-F10 cells for 1 h. After incubation, the IFN-γ-stimulated B16-F10 cells were collected and stained with Alexa Fluor 647 anti-6×His tag antibody to label the 6×His tag of αPD-L1. The quantity of αPD-L1 bound to the cell surface was detected by flow cytometry.

The efficiency of αPD-L1 in increasing the cytotoxicity of CD8$^+$ T cells was evaluated in a killing assay. B16-F10-OVA-EGFP cells were stimulated with IFN-γ, and primary OVA-specific CD8$^+$ T cells were sorted and activated, as mentioned above. IFN-γ-stimulated B16-F10-OVA-EGFP cells and activated OVA-specific CD8$^+$ T cells were coincubated at a cell ratio of 1:5 in the culture supernatants of B16-F10 cells transfected with PBS, $NP_{Control}$, $NP_{Tyr-\alpha PD-L1}$ or $NP_{Tyr-C9AP}$. Twenty-four hours after incubation, the suspension containing OVA-specific CD8$^+$ T cells was removed. For the PI staining assay, B16-F10-OVA-EGFP cells were collected and stained with PI dye (Thermo Fisher) to analyze the percentages of dead (PI-positive) cells by flow cytometry. For the CCK-8 assay, CCK-8 reagent (Biosharp) was added to the B16-F10-OVA-EGFP cells, and the absorbance at 450 nm was measured after 1 h of incubation to calculate the viability of the B16-F10-OVA-EGFP cells.

## Transwell migration and cytotoxicity assays

IFN-γ-stimulated B16-F10-OVA-EGFP cells ($5 \times 10^4$ cells per well) were seeded in the lower chamber of transwell plates, while activated OVA-specific CD8$^+$ T cells ($2 \times 10^6$ cells per well) were cultured in the upper chamber. Then, the original medium of the lower chamber was replaced with the culture supernatants of the B16-F10 cells transfected with PBS, $NP_{Control}$, $NP_{Tyr-CXCL9}$, $NP_{Tyr-\alpha PD-L1}$ or $NP_{Tyr-C9AP}$ (the transfection doses were 1 μg ml$^{-1}$ pUC57, pTyr-CXCL9, pTyr-αPD-L1 or pTyr-C9AP). The cells were further cultured for 12 h. Finally, as mentioned

above, OVA-specific CD8[+] T cells recruited to the lower chamber were counted using Precision Count Beads, and the percentages of PI-positive B16-F10-OVA-EGFP cells were analyzed by PI-staining assays. For CLSM imaging, activated OVA-specific CD8[+] T cells were prestained with Hoechst 33342 (Thermo Fisher), and the other operations were the same as described above. After culture for 12 h, the recruitment and cytotoxicity of OVA-specific CD8[+] T cells was analyzed by visualizing the numbers of Hoechst 33342-stained OVA-specific CD8[+] T cells and B16-F10-OVA-EGFP cells in the lower chamber with an LSM880.

### Analysis of the expression of CXCL9 and αPD-L1 in vivo

Mice bearing B16-F10 melanoma were randomly divided into five groups ($n = 4$ mice per group) when the tumor volumes reached 100 mm³. Then, the mice were intravenously injected with PBS, $NP_{Control}$, $NP_{Tyr-CXCL9}$, $NP_{Tyr-αPD-L1}$ or $NP_{Tyr-C9AP}$ every other day for three injections. The injection doses were 1 mg pUC57, pTyr-CXCL9, pTyr-αPD-L1 or pTyr-C9AP per kg body weight. Seventy-two hours after the last injection, the melanoma and serum were collected. The mRNA expression of CXCL9 and αPD-L1 in the melanoma tissues was analyzed by qRT–PCR. The CXCL9 concentrations in the melanoma tissues and serum were detected by ELISA. The expression of αPD-L1 protein in the melanoma tissues was analyzed by western blot and was quantified by ELISA.

Mice bearing Clone M-3 or YUMM1.7 melanoma were established by subcutaneously inoculating $1.5 \times 10^6$ Clone M-3 or YUMM1.7 cells into the right flank of C57BL/6 mice. The mice bearing different melanomas were randomly divided into two groups ($n = 4$ mice per group) when the tumor volumes reached 100 mm³. Then, the mice were intravenously injected with $NP_{Tyr-C9AP}$ or $NP_{Control}$ every other day for three injections. The injection doses were 1 mg pTyr-C9AP or pUC57 per kg body weight. Seventy-two hours after the last injection, the melanoma tissues were collected, and the intratumoral concentrations of CXCL9 and αPD-L1 were detected by ELISA.

### Treatment of different melanomas by $NP_{Tyr-C9AP}$

Mice bearing B16-F10 melanoma were randomly divided into five groups ($n = 10$ mice per group) when the tumor volumes reached 50 mm³. Then, the mice were intravenously injected with PBS, $NP_{Control}$, $NP_{Tyr-CXCL9}$, $NP_{Tyr-αPD-L1}$ or $NP_{Tyr-C9AP}$ every other day for five injections. The injection doses were 1 mg pUC57, pTyr-CXCL9, pTyr-αPD-L1 or pTyr-C9AP per kg body weight. The tumor length and width and body weights of the mice were monitored every other day, and the tumor volumes were calculated using the formula: $V = L \times W^2 \times 1/2$ ($V$, volume; $L$, length; $W$, width of tumor). The therapeutic experiments were terminated immediately when the largest tumor size reached or exceeded 2 cm³. The tumor tissues and other major organs (including the heart, liver, spleen, lung and kidney) were collected at the end of the treatments (18 days post-tumor xenograft). The final tumor weights were measured. The histopathology of the other major organs was analyzed by hematoxylin & eosin (H&E) staining. For the survival experiment, mice bearing B16-F10 melanoma ($n = 10$ mice per group) were treated with PBS, $NP_{Control}$, $NP_{Tyr-CXCL9}$, $NP_{Tyr-αPD-L1}$ or $NP_{Tyr-C9AP}$ as described above, and the survival time of each mouse was recorded.

Mice bearing Clone M-3 and YUMM1.7 melanoma were randomly divided into two groups ($n = 6$ mice per group) when the tumor volumes reached 50 mm³. Then, the mice were intravenously injected with $NP_{Tyr-C9AP}$ or $NP_{Control}$ every other day for five injections. The injection doses were 1 mg pTyr-C9AP or pUC57 per kg body weight. The tumor volumes were monitored as described above. The tumor tissues were collected at the end of the treatments and the final tumor weights were measured.

### Potential liver toxicity analysis of $NP_{Tyr-C9AP}$

C57BL/6 mice were intravenously injected with PBS, $NP_{Control}$, $NP_{Tyr-CXCL9}$, $NP_{Tyr-αPD-L1}$ or $NP_{Tyr-C9AP}$ every other day for five injections ($n = 4$

mice per group). The injection doses were 1 mg pUC57, pTyr-CXCL9, pTyr-αPD-L1 or pTyr-C9AP per kg body weight. One week after the last injection, the serum samples were collected to detect the levels of alanine aminotransferase (ALT), aspartate aminotransferase (AST) and alkaline phosphatase (ALP) using ELISA kits (Rayto, Guang-dong, China).

### Comparison of antitumor effects of $NP_{Tyr-C9AP}$ and anti-PD-L1 antibody

Mice bearing B16-F10 melanoma were randomly divided into five groups ($n = 6$ mice per group) when the tumor volumes reached 100 mm³. Then, the mice were intravenously injected with PBS, anti-PD-L1 antibody, $NP_{Tyr-αPD-L1}$, $NP_{Tyr-CXCL9}$ & anti-PD-L1 antibody or $NP_{Tyr-C9AP}$. Anti-PD-L1 antibody (InVivo MAb anti-mouse PD-L1(B7-H1), clone: 10 F.9G2) was purchased from BioXCell (NH, USA), and was injected every other day for five injections at an injection dose of 2.5 mg per kg body weight. $NP_{Tyr-CXCL9}$, $NP_{Tyr-αPD-L1}$ and $NP_{Tyr-C9AP}$ were injected every other day for five injections and the injection doses were 1 mg pTyr-CXCL9, pTyr-αPD-L1 or pTyr-C9AP per kg body weight. The tumor volumes and body weights of the mice were monitored every other day. The tumor tissues were collected at the end of the treatments and the final tumor weights were measured.

### Treatment of different tumors by $NP_{Sur-C9AP}$

Mice bearing B16-F10, CT26, Panc02 and 4T1 tumors were established as described above. When the tumor volumes reached 50 or 100 mm³, the mice bearing different tumors were randomly divided into two groups ($n = 6$ mice per group). For the treatment of mice bearing B16-F10, CT26 or 4T1 tumors, $NP_{Control}$ or $NP_{Sur-C9AP}$ was intravenously injected every other day for five injections. For the treatment of mice bearing Panc02 tumors, $NP_{Control}$ or $NP_{Sur-C9AP}$ was intravenously injected every four days for five injections. The injection doses were 1 mg pUC57 or pSur-C9AP per kg body weight. The tumor volumes were monitored as described above. The tumor tissues were collected at the end of the treatments and the final tumor weights were measured.

### Immune profiling of tumors by flow cytometry

The tumor tissues were dissociated into single cells by enzymatic digestion in RPMI 1640 medium containing type-IV collagenase (1 mg ml⁻¹, Sangon), deoxyribonuclease I (100 µg ml⁻¹, Sangon) and hyaluronidase (100 µg ml⁻¹, Sangon) at 37 °C for 25 min. Before labeling the markers on the cell surface, Fc receptors were firstly blocked using anti-mouse CD16/32 antibody. For analysis of T and NK cells, the single cells were stained with Brilliant Violet 510 anti-mouse CD45, APC anti-mouse CD3, APC/Cyanine7 anti-mouse CD4, Brilliant Violet 785 anti-mouse CD8a, Brilliant Violet 605 anti-mouse CD69, Alexa Fluor 488 anti-mouse NK-1.1 (for C57BL/6 mice) or FITC anti-mouse NKp46 (for BALB/c mice). For analysis of myeloid cells, the single cells were stained with Brilliant Violet 510 anti-mouse CD45, FITC anti-mouse CD11b, APC/Cyanine7 anti-mouse Gr-1, Brilliant Violet 650 anti-mouse F4/80, PE/Dazzle594 anti-mouse CD80, and PE anti-mouse CD206. For analysis of granzyme B, perforin and IFN-γ expression in CD8[+] T cells, single cells were stimulated in RPMI 1640 medium with 10% FBS and Cell Stimulation Cocktail (plus protein transport inhibitors) (eBioscience, CA, USA) according to the manufacturer's instructions. After stimulation, the cells were stained with Brilliant Violet 510 anti-mouse CD45, APC anti-mouse CD3, APC/Cyanine7 anti-mouse CD4 and Brilliant Violet 785 anti-mouse CD8a. Then, the cells were permeabilized with a Cytofix/Cytoperm Fixation/Permeabilization kit (BD Biosciences) and stained with FITC anti-human/mouse Granzyme B, PE anti-mouse Perforin and Brilliant Violet 421 anti-mouse IFN-γ. For analysis of Tregs, the single cells were stained with Brilliant Violet 510 anti-mouse CD45, APC anti-mouse CD3, APC/Cyanine7 anti-mouse CD4, Brilliant Violet 785 anti-mouse CD8a and FITC anti-mouse CD25.

Then, the cells were permeabilized with True-Nuclear Transcription Factor Buffer Set (BioLegend) and stained with PE anti-mouse Foxp3. All surface antibodies were used at 1:100 dilution; all intracellular antibodies were used at 1:50 dilution. All samples were analyzed with FACSCelesta. The data were collected using the BD FACS Diva software v8.0.1.1 and analyzed using FlowJo software v10.0.7.

## Multiple immunofluorescence staining of tumors

The tumor tissues were fixed with 4% paraformaldehyde, embedded in paraffin and cut into 4 μm sections with HistoCore Autocut (Leica Biosystems, MA, USA). Then, the deparaffinized and rehydrated sections were stained with the primary antibodies anti-CD3ε rabbit mAb (Cell Signaling Technology, MA, USA) at 1:200 dilution, anti-CD8α rabbit mAb at 1:200 dilution (Cell Signaling Technology), and Brilliant Violet 605 anti-mouse CD69 at 1:100 dilution (BioLegend) and labeled with a TSA-Rab multiplex immunofluorescence staining kit (Panovue, Beijing, China) according to the manufacturer's instructions. The cell nuclei of the sections were stained with DAPI before being visualized by an LSM880.

## Statistical analysis

All values are presented as the means ± SEM. Statistical analysis was performed using a two-sided t-test to compare two groups, one-way ANOVA with Tukey's multiple comparison test to compare multiple groups, two-way ANOVA with Greenhouse–Geisser correction to compare the tumor growth curves, and the log-rank (Mantel–Cox) test to compare the survival curves. $P < 0.05$ was considered statistically significant. Statistical analysis was conducted using GraphPad Prism v8.0 (GraphPad Software, Inc.).

## Reporting summary

Further information on research design is available in the Nature Portfolio Reporting Summary linked to this article.

## Data availability

The sequence of *CXCL9* cDNA was obtained from the CCDS database with ID number CCDS39152.1. Source data are provided with this paper. The data that support this study are available within the Article, Supplementary Information or Source data files. Source data are provided with this paper.

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

## Acknowledgements

This work was supported by the National Key R&D Program of China (2022YFC3401400 to J.W.), the National Natural Science Foundation of China (52130301 to J. W., 82072048 to C.-F.X., 81901875 to Z.-D.L., and 32271442 to C.-F.X.), the Guangdong Basic and Applied Basic Research Foundation (2022B1515020025 to C.-F.X.), the Science and Technology Program of Guangzhou, China (202103030004 to J.W.), and the Fundamental Research Funds for the Central Universities (2022ZYGXZR102 to C.-F.X.).

## Author contributions

C.-F.X. and J.W. conceived the project, designed the experiments, analyzed the data, wrote and revised the manuscript. Yue Wang and S.-K.Z. performed the experiments, analyzed the data, wrote and revised the manuscript. Yan Wang, Z.-D.L., Y.Z. assisted in the experimental design and data analysis. All the authors discussed the results and assisted in the preparation of the manuscript.

## Competing interests

The authors declare no competing interests.
