## [Peer Review File · Nature Communications]

Engineering tumor-specific gene nanomedicine to recruit and activate T cells for enhanced immunotherapyREVIEWER COMMENTS

Reviewer #1 (Remarks to the Author): with expertise in nanomedicine

In this manuscript, Wang et al. reported a nanomedicine to specifically express aPD-L1 scfv and CXCL9 in tumor sites for immunotherapy of melanoma and other tumors. Extensive data from in vitro and in vivo testing were presented to show that anticancer immunity was enhanced by this nanomedicine, however, major concerns arise, including the lack of novelty and rationale of the nanomedicine design, incomplete characterization of target protein expression, and insufficient mechanism studies. Overall, this inducible aPD-L1 (scFV) and CXCL9 nanotherapeutic may not overcome the main hurdle of anti-PD-L1 therapy encountered in clinics because of the lack of ADCC/ADCP functions of the scFV and the lack of the tumor-specific antigen presenting by their strategies.

Major points:

1. The novelty of this study is moderate. The concept of a local expression of scFV that interrupts PD-L1/PD-1 pathway was reported previously including the combination with CAR-T (Nat Biotechnol. 2018 Oct; 36(9): 847–856., Front. Cell Dev. Biol., 2020 August; 8:803) and oncolytic virus (Cell Mol Immunol. 2019 Sep; 16(9): 780–782). These designs seemed more rational for producing synergistic anticancer activity. In addition, the approach of cell type-specific promoter-induced expression of specific genes has been reported by several studies from this group.
2. It is still unclear how the anti-tumor response was promoted without priming the TME in this study. How was tumor killing generated through T-cell activation but without DC priming? Tumor antigen release and antigen-presenting cell (APC) priming should be considered when designing the delivery system. Neither anti-PD-L1 scFV nor CXCL9 could enhance the exposure of tumor antigen initially and promote the tumor antigen processing and presentation in APCs. Moreover, the migration and maturation of APCs in tumor-draining lymph nodes were not likely to be promoted by this nanoparticle either. This step is crucial to induce tumor antigen-specific T-cell response but was totally ignored in this study.
3. The rationale and advantages of this nano-system should be discussed in more detail. Firstly, ADCC and ADCP effects of PD-L1 antibody play an important role in PD-L1 antibody-mediated immunotherapy of cancers. Using locally induced and secreted scFV of PD-L1 in tumors may dampen the therapeutic potential of anti-PD-L1 antibody. Secondly, as mentioned above, the idea of locally induced scFV to interrupt PD-L1/PD-1 pathway has been extensively studied in more rational design by other groups. In those published studies, inducible scFV to target PD1/PD-L1 axis was combined with CAR-T or oncolytic virus. CAR-T and oncolytic virus can kill tumor cells, more likely leading to the release of tumor antigen and thereafter tumor antigen-specific immune response. The rationale of these combinations is more convincing to me. However, in this study, NK or macrophages-mediated ADCC or ADCP was missing, and other strategies were used for tumor cell killing. Therefore, tumor antigen release may not occur after the treatment with the nanoparticle, which could greatly limit the anti-tumor effect of aPD-L1 therapy.
4. The authors claimed this Nano-system could successfully produce aPD-L1 scFV and CXCL9 in tumor cells, however, they did not provide sufficient characterization for both protein expressions. For example, what are the protein production yields for both proteins in different tumor types? And what is the final working concentration they can achieve in the culture medium for in vitro experiments (CXCL9 was provided but scFV was missing)? And how about the tumor site? And what is the percentile of secreted proteins/total produced proteins? This information is important because it could help us to determine the working concentration of scFV for in vivo study. Since the data for protein characterization such as SDS-page or LS-MS of the produced proteins were not provided, it is suggested to provide the entire picture of the western blot with protein markers in the supplementary data.

5. The biodistribution of these nanoparticles should be carefully studied as it was missing in the current study. Preferably, the tumor-specific induction of gene expression (use Luciferase or RFP as reporter protein) and therapeutic effect (aPD-L1 and CXCL9) should be studied in vivo with bilateral tumor models (for example, B16-F10 tumor on the right back and other types of tumors on the left side).

6. More data are needed to reveal the underlying mechanisms of aPD-L1-induced reduction of immunosuppressive cell populations. Also, why could NP-CXCL9 reduce the M1/M2 ratio and MDSC populations in tumors by enhanced infiltration of T cells? More evidence should be provided about the transformed immune microenvironment such as cytokines/chemokines changes.

Minor concerns:

1. It seems that the TEM image in Figure 2c indicated that the nanoparticles were aggregated.

2. Fluorescence images with clearer background and DAPI staining should be presented in Figure 2f.

3. The quantitative dose of aPD-L1 induced in various tumors in vivo should be measured. And the PD-L1 expression level in the TME should also be studied. Will there be any response differences if the TME shows a different PD-L1 level?

4. CD8/CD4 ratio is not a good parameter for immune activation in tumors. The CD8/Treg ratio provides a better indication of immune activation. In addition, why did CXCL9 recruit more CD8 cells than CD4 T cells?

5. In vitro and vivo expression of CXCL9 and aPD-L1 should be determined on the protein level as well (quantitatively, related to figure 2 i to l, figure 6b).

6. Representative flow cytometry images related to Figs 5 and 6 should be provided as supplementary files.

7. Did the authors observe systemic antigen-specific immune response after this nanoparticle treatment? And what about the generation of memory T cells and long-term immune protection from these nanoparticles?

8. To show the gating strategy of FACS, pseudocolor plots may provide more information.

Reviewer #2 (Remarks to the Author): with expertise in cancer immunology/immunotherapy

The manuscript entitled "Engineering tumor-specific gene nanomedicine to recruit and activate T cells for enhanced immunotherapy" describes a novel nanoparticle agent delivering an expression plasmid for producing anti-PD-L1 blocking scFv and a chemokine CXCL9 in melanoma cells. This is achieved by using a melanocyte-specific promoter. The authors also describe a similar nanoparticle that can be used for treating various cancers more broadly where the survivin promoter is used to drive anti-PD-L1 scFv and CXCL9 expression. The idea of modifying tumor cells using a nanomedicine approach in a way that would promote T cell infiltration and activity is interesting. Furthermore, the effect of described nanomedicines on T cell recruitment and activity is evident in vitro and in vivo, and the anti-tumor effect in a mouse model is impressive. One potential concern is that only one in vivo model was used to test melanoma-specific nanomedicine. There are several specific points for improvement indicated below:

1. The NP^{Tyr}-C9AP was tested in B16 melanoma cells and tumors. It would be good to include 1-2 additional mouse melanoma models, such as YUMM models or other available models to ensure reproducibility. It would also make this study very compelling if a similar human-specific nanomedicine could be generated and tested in a human model, such as co-culture of Patient-Derived tumor cells

and tumor infiltrating leukocyte, for example.

2. Additional characterization of potential toxicities could further improve the study. For instance, serum levels of liver enzymes AST and ALT could be assessed. In addition, if feasible, Cy5-labeled NPTyr-C9AP could be administered in vivo to see in which organs nanoparticles would accumulate.
3. Discussion mostly states the results. It is important to include a proper discussion. For example, describe literature on nanomedicines/nanoparticles with a similar design, clinical experience, etc.

Reviewer #3 (Remarks to the Author): with expertise in cancer immunology/immunotherapy

This manuscript describes an interesting approach to simultaneously enhance T cell recruitment and activity against solid tumors through a nanomedicine approach designed to engineer tumor cells to secrete CXCL9 and anti-PD-L1. The major strength of the work is the novelty of the approach; whilst other approaches have been developed to engineer tumor cells to express CXCR3 ligands, this is the first to my knowledge to specifically utilize tumor promoters to achieve this. One question that emerges from this work is the relevance of using this system to target anti-PD-L1 specifically to the tumor microenvironment. To improve the therapeutic relevance of this study further experiments are suggested to confirm the utility of the Survivin promoter in human tumor cells relative to relevant control cells derived from similar tissues and to benchmark therapeutic efficacy to anti-PD-L1.

Major points

- 1) Can the authors demonstrate superior efficacy / safety of targeting PD-L1 through this approach as opposed to systemic administration of anti-PD-L1
- 2) The extent of the immune infiltration observed post therapy by IF (Figure 5g) looks rather modest. How do the authors reconcile this with profound therapeutic effects?
- 3) Some of the observations appear under powered and it is not apparent how many repeats have been performed e.g. Figure 2l, 4a n = 3 mice per group, others e.g. figure 4d, 4f are n = 10 mice per group but not stated how many repeats
- 4) Figure 5h the data on CD3+ T cells should be stratified at least into CD4+ and CD8+, otherwise it could be that changes in expression of effector molecules merely reflect changes in the proportion of CD8+ T cells.
- 5) IN Figure 6b the selectivity of the Survivin promoter is demonstrated but this is relative to other cell lines that are not relevant to the tissue from which the tumors originated. Matched samples should be used, preferably using human-derived tumor cells.

Minor points

- 1) It is suggested to add information regarding how many bps of the relevant promoters were incorporated into the construct within the main text, as this is critical information with regards to specificity.
- 2) In Figure 2 what is the concentration of CXCL9 protein that is achieved following the use of NPTyr-C9AP?
- 3) What are the kinetics of the killing assays in Figure 3? The reviewer asks as generally it is difficult to model the impact of PD-1:PD-L1 interactions in vitro.
- 4) The authors state that the recruitment of NK cells contributed to the therapeutic efficacy. Whilst this is possible, it was not experimentally determined and so this comment should be softened.

We greatly appreciate the valuable comments of the three reviewers, helping us improve the quality of the manuscript. For the purpose of clarity, the reviewer's comments are in black while our point-by-point responses are marked in blue.

Reviewers' comments:

Reviewer #1:

General Comments: In this manuscript, Wang et al. reported a nanomedicine to specifically express aPD-L1 scfv and CXCL9 in tumor sites for immunotherapy of melanoma and other tumors. Extensive data from in vitro and in vivo testing were presented to show that anticancer immunity was enhanced by this nanomedicine, however, major concerns arise, including the lack of novelty and rationale of the nanomedicine design, incomplete characterization of target protein expression, and insufficient mechanism studies. Overall, this inducible aPD-L1 (scFV) and CXCL9 nanotherapeutic may not overcome the main hurdle of anti-PD-L1 therapy encountered in clinics because of the lack of ADCC/ADCP functions of the scFV and the lack of the tumor-specific antigen presenting by their strategies.

Response: Thanks for the comments. We appreciate the concerns about the novelty and rationale of our nanomedicine design. As we have discussed in the Introduction section of the manuscript, the unsatisfactory therapeutic benefits of α PD-L1 therapy in clinical practices are mainly due to the lack of tumor-infiltrating T cells and the insufficient tumor accumulation of α PD-L1 inhibitors after systemic injection. In addition, PD-L1 is not only expressed in tumor cells but also dendritic cells (DCs), macrophages, B cells, endothelial cells, epithelial cells, fibroblasts, and so on (Nat Med. 2003 May;9(5):562–7, Sci Transl Med. 2016 Mar 2;8(328):328rv4). Systemic injection of α PD-L1 inhibitors can induce T-cell cytotoxicity against normal tissues (Ann Oncol. 2015 Dec;26(12):2375–91). Although some reported strategies are effective in improving α PD-L1 therapy by increasing either tumor-infiltrating T cells or intratumoral α PD-L1 inhibitors, **no strategy can actively recruit T cells into tumors and meanwhile**

achieve tumor-specific delivery of α PD-L1 inhibitors to specifically eliminate the immune inhibition of tumor-infiltrating T cells. Our nanomedicines employ biodegradable nanoparticles to overcome the delivery barriers of plasmids, and innovatively utilize tumor-specific promoters to drive tumor cells to express CXCL9 and α PD-L1 scFv in situ, which are demonstrated to simultaneously build sharp CXCL9 gradients for effectively recruiting T cells into tumors and achieve high intratumoral α PD-L1 concentrations for blocking PD-L1 to enhance T-cell activation. As a result, our nanomedicines have induced enhanced anti-tumor effects. Moreover, we have constructed two versions of tumor-specific nanomedicines driven by tyrosinase promoter or survivin promoter for satisfying the treatments of melanoma and different other tumors. Overall, our design provides a novel strategy using nanomedicines to engineer diverse tumor cells into the producers of T-cell chemokines and α PD-L1 inhibitors, which skillfully address the hurdles of immune checkpoint blockade (ICB) therapy. More discussion about the novelty and rationale of our nanomedicine design has been added in the revised manuscript (line 583 of page 35 to line 604 of page 36; line 611 to 623 of page 37).

As to the concern about the lack of ADCC/ADCP functions of the scFv, α PD-L1 therapy mainly relies on blocking PD-L1 to eliminate the inhibition signals against T cells to enhance T-cell activation and cytotoxicity (Cancer Discov. 2018 Sep;8(9):1069–86). Although the ADCC/ADCP functions of fragment crystallizable region (Fc) can induce NK cells and macrophages to contribute anti-tumor responses, the ADCC/ADCP effects induced by Fc/Fc receptor interaction are not crucial for α PD-L1 therapy (Int J Cancer. 2019 Jan 15;144(2):345–54). In addition, α PD-L1 scFv that lacks Fc has been proved to be enough for inducing anti-tumor responses by enhancing T-cell activation (Nat Biotechnol. 2018 Oct;36(9):847–56, Cancer Immunol Res. 2020 May;8(5):632–47). And, blocking PD-L1 by α PD-L1 scFv can enhance the activation of both T cells and NK cells due to the PD-1 inhibitory receptor is expressed on both T and NK cells (Front Immunol. 2019 Jun 4; 10:1242). Moreover, atezolizumab contains a modified Fc receptor binding site that removes ADCC and ADCP functions (Front

Immunol. 2020 May 29; 11:1088) has shown good efficacy in treating various tumors (N Engl J Med. 2020 May 14;382(20):1894–905, N Engl J Med. 2020 Oct 1;383(14):1328–39, N Engl J Med. 2018 Nov 29;379(22):2108–21), and atezolizumab is approved for the treatments of multiple cancer types, including melanoma, small cell lung cancer (SCLC), NSCLC, RCC, head and neck squamous cell carcinomas (HNSCC), classical Hodgkin lymphomas (cHL), and Merkel cell carcinoma (Mol Cell. 2019 Nov 7;76(3):359–70). The α PD-L1 scFv without Fc can not only easily penetrate the tumor to block PD-L1 of tumor cells, but also avoid ADCC/ADCP-mediated potential toxicities to normal tissues. Therefore, α PD-L1 scFv without Fc is an excellent anti-PD-L1 drug. More discussion about the rationale of α PD-L1 scFv therapy has been added in the revised manuscript (line 604 of page 36 to line 610 of page 37).

In regard to the lack of the tumor-specific antigen presentation in our strategy, our goal is to address the poor therapeutic benefits of ICB therapy. The lack of the tumor-specific antigen presentation is not a major hurdle of ICB therapy. During tumor growth and the ICB treatments, both alive and injured tumor cells will continuously expose and release tumor antigens, and antigen-presenting cells (APCs) can capture these antigens and infiltrate into lymph nodes to activate tumor antigen-specific T cells (Cancer Lett. 2016 Jan 1;370(1):85-90). In addition, although APCs in the tumor microenvironment are suppressed (Nat Med. 2003 May;9(5):562–7, Sci Transl Med. 2016 Mar 2;8(328):328rv4), α PD-L1 therapy-mediated blockade of PD-L1/PD-1 pathway can potentially improve the tumor antigen processing and presentation of APCs for activating T cells (J Clin Invest. 2018 Feb 1;128(2):580–8, J Clin Invest. 2018 Feb 1;128(2):805–15, Mol Cell. 2019 Nov 7;76(3):359–70). Thus, our strategy utilizing nanomedicines to coexpress CXCL9 and α PD-L1 scFv in tumors has shown enhanced anti-tumor effects via CXCL9-mediated T-cell recruitment and α PD-L1 scFv-enhanced T-cell activation. More discussion about the rationale of our nanomedicine design has been added in the revised manuscript (line 641 of page 38 to line 645 of page 39).

Furthermore, we have used ELISA to reanalyze the protein expression levels of CXCL9 and α PD-L1 in different tumor cells and multiple tumor tissues after NP_{Tyr-C9AP}

or NP_{Sur-C9AP} treatments. The quantitative results have been added in the revised manuscript and supplementary information (Fig. 2h, j and k, Fig. 3a and d, Fig. 4a and b, Supplementary Fig. 26, Fig. 7a, b, f and g, Fig. 8b to d, Supplementary Fig. 29; line 145 of page 8 to line 148 of page 9, line 156 to 159 of page 9, line 194 to 197 of page 12, line 220 to 222 of page 13, line 312 to 319 of page 19, line 468 to 474 of page 28, line 516 to 526 of page 31).

In addition, in order to further elucidate the immunological mechanisms of our nanomedicine-mediated anti-tumor effects, we have analyzed the populations of granzyme B-, perforin- and IFN- γ -positive CD8⁺ effector T cells and the ratios of CD8⁺ T cells/Tregs in tumor tissues by flow cytometry after the treatments of NP_{Tyr-C9AP} or NP_{Sur-C9AP}. The results of flow cytometry results have been added in the revised manuscript and supplementary information (Fig. 5f, h to j, Supplementary Fig. 19 to 21, Fig. 8h, j to l, Supplementary Fig. 31; line 380 and 381, 386 to 390 of page 23, line 539 to 544 of page 32). The expression of cytokines, including IFN- γ , IL-6, GM-CSF, IL-4 and IL-10, have also been detected by qRT-PCR analysis after the treatments of NP_{Tyr-C9AP} to indicate the improvements of tumor immune microenvironments. The results of qRT-PCR have been added in the revised manuscript and supplementary information (line 394 to 397 of page 23; Supplementary Fig. 22).

Major comments:

Comment 1: The novelty of this study is moderate. The concept of a local expression of scFV that interrupts PD-L1/PD-1 pathway was reported previously including the combination with CAR-T (Nat Biotechnol. 2018 Oct; 36(9): 847–856., Front. Cell Dev. Biol., 2020 August; 8:803) and oncolytic virus (Cell Mol Immunol. 2019 Sep; 16(9): 780–782). These designs seemed more rational for producing synergistic anticancer activity. In addition, the approach of cell type-specific promoter-induced expression of specific genes has been reported by several studies from this group.

Response: Thanks for the comments. Although the concept of a local expression scFv that blocks the PD-L1/PD-1 pathway has been reported in other works (Nat Biotechnol.

2018 Oct; 36(9): 847–56, Cell Mol Immunol. 2019 Sep; 16(9): 780–2). These strategies are quite different from ours. Using CAR-T cells to express α PD-L1 scFv improved the efficacy of CAR-T cells and reduced toxicities associated with systemic injection of α PD-L1 inhibitors (Nat Biotechnol. 2018 Oct; 36(9): 847–56). But CAR-T cells were distributed in the whole body after systemic injection, which could not really achieve local expression of scFv in tumor tissues; and this strategy did not address the insufficient infiltration of CAR-T cells in the treatments of solid tumors. Although intratumoral injection of oncolytic virus engineered with α PD-L1 scFv achieved local expression in tumor tissues (Cell Mol Immunol. 2019 Sep; 16(9): 780–2), the intratumoral injection strategy is not applicable for all metastatic tumors, and this study also did not address the insufficient T-cell infiltration. By contrast, our tumor-specific nanomedicines have overcome the two crucial hurdles of ICB therapy via specifically engineering tumor cells to locally coexpress CXCL9 and α PD-L1 for recruiting T cells and enhancing T-cell activation, resulting in enhanced anti-tumor effects. Our strategy has not only addressed the problems of low enrichment and potential toxicities caused by systemic injection of α PD-L1 inhibitors, but also addressed the problem of insufficient T-cell infiltration in tumors. More discussion about the novelty of our nanomedicine design has been added in the revised manuscript (line 583 of page 35 to line 604 of page 36).

In addition, although our group has reported the approach of cell type-specific promoter-induced expression of Cas9, the study was using CD68 promoter to achieve macrophage-specific gene editing (ACS Nano. 2018 Feb 27;12(2): 994–1005), which was unrelated to cancer immunotherapy and did not affect the novelty of this work. As mentioned above, there is no study using tumor cell type-specific promoters to specifically drive tumor cells to coexpress CXCL9 and α PD-L1 for enhancing T-cell infiltration and activation in tumors. The key novelty and significance of this work is that our tumor-specific gene nanomedicines can simultaneously address the two crucial hurdles of ICB therapy. More discussion has been added in the revised manuscript (line 617 to 623 of page 37).

Comment 2: It is still unclear how the anti-tumor response was promoted without priming the TME in this study. How was tumor killing generated through T-cell activation but without DC priming? Tumor antigen release and antigen-presenting cell (APC) priming should be considered when designing the delivery system. Neither anti-PD-L1 scFV nor CXCL9 could enhance the exposure of tumor antigen initially and promote the tumor antigen processing and presentation in APCs. Moreover, the migration and maturation of APCs in tumor-draining lymph nodes were not likely to be promoted by this nanoparticle either. This step is crucial to induce tumor antigen-specific T-cell response but was totally ignored in this study.

Response: Thanks for the comments. We apologize that Reviewer #1 might misunderstand our study. Our strategy has promoted anti-tumor responses by recruiting activated T cells into tumors and eliminating PD-L1 inhibitory signal to enhance T-cell activation simultaneously, that is our study enhances the tumor infiltration and anti-tumor effector functions of activated T cells. These T cells have already been activated by innate antigen presentation of APCs or DCs that have captured tumor antigens during tumor growth and the treatments of our nanomedicines. As mentioned above, the lack of DCs priming or presenting tumor antigens to activate T cells is not a major hurdle of ICB therapy. During tumor growth and the ICB treatments, both alive and injured tumor cells will continuously expose and release tumor antigens, and tumor cells killed by the immune system will also release tumor antigens. DCs can capture these antigens and infiltrate into lymph nodes to activate tumor antigen-specific T cells (Cancer Lett. 2016 Jan 1;370(1): 85–90). Thus, tumor antigen release and APCs priming were not considered when designing our nanomedicines. More discussion has been added in the revised manuscript (line 641 of page 38 to line 645 of page 39).

We agree with Reviewer #1 that our nanomedicines cannot enhance the initial exposure of tumor antigens. But it's reported that PD-L1 is upregulated on APCs of tumor microenvironment and tumor-draining lymph nodes, α PD-L1 therapy-mediated blockade of PD-L1/PD-1 pathway can potentially improve the tumor antigen

processing and presentation of APCs for activating T cells (J Clin Invest. 2018 Feb 1;128(2):580–8, J Clin Invest. 2018 Feb 1;128(2):805–15). Downregulating PD-L1 expression on APCs can enhance the response rates of ICB therapy (Mol Cell. 2019 Nov 7;76(3):359–70). Therefore, our nanomedicines-mediated α PD-L1 expression may also have promoted APCs-mediated antigen presentation to T cells.

In addition, CXCR3, the receptor of CXCL9, is expressed not only on T cells but also on NK cells and macrophages (Cancer Treat Rev. 2018 Feb;63:40-47). Our results show that our NP_{Tyr-C9AP} have increased the proportions of activated NK cells (Fig. 5k and Fig. R1) and M1-like macrophages in tumors (Fig. 5l), which also contributes to the exposure, processing and presentation to activate tumor antigen-specific T cells.

Fig. R1 Percentages of CD69-positive activated NK cells in B16-F10 melanoma after different treatments. B16-F10 melanoma-bearing mice were injected with PBS, NP_{Control}, NP_{Tyr-CXCL9}, NP_{Tyr- α PD-L1} or NP_{Tyr-C9AP} every other day for five injections ($n = 4$ mice per group). The injection doses were 1 mg pUC57, pTyr-CXCL9, pTyr- α PD-L1 or pTyr-C9AP per kg body weight. The percentages of CD69-positive activated NK cells were detected by flow cytometry after the treatments. The data are shown as the means \pm SEM of $n = 4$. One-way ANOVA with Tukey's multiple comparison test. * $P < 0.05$; *** $P < 0.001$; ns indicates no significant difference.

Fig. 5k, l Percentages of NK cells in CD45⁺ cells (**k**), ratios of M1-like/M2-like macrophages (**l**) of B16-F10 melanoma after different treatments. The data are shown

as the means \pm SEM of $n = 4$. One-way ANOVA with Tukey's multiple comparison test. * $P < 0.05$; ** $P < 0.01$; *** $P < 0.001$; ns indicates no significant difference.

Comment 3: The rationale and advantages of this nano-system should be discussed in more detail. Firstly, ADCC and ADCP effects of PD-L1 antibody play an important role in PD-L1 antibody-mediated immunotherapy of cancers. Using locally induced and secreted scFv of PD-L1 in tumors may dampen the therapeutic potential of anti-PD-L1 antibody. Secondly, as mentioned above, the idea of locally induced scFv to interrupt PD-L1/PD-1 pathway has been extensively studied in more rational design by other groups. In those published studies, inducible scFv to target PD1/PD-L1 axis was combined with CAR-T or oncolytic virus. CAR-T and oncolytic virus can kill tumor cells, more likely leading to the release of tumor antigen and thereafter tumor antigen-specific immune response. The rationale of these combinations is more convincing to me. However, in this study, NK or macrophages-mediated ADCC or ADCP was missing, and other strategies were used for tumor cell killing. Therefore, tumor antigen release may not occur after the treatment with the nanoparticle, which could greatly limit the anti-tumor effect of aPD-L1 therapy.

Response: Thanks for the comments. As mentioned above, α PD-L1 therapy mainly relies on blocking PD-L1 to eliminate the inhibition signals against T cells to enhance T-cell activation and cytotoxicity (Cancer Discov. 2018 Sep;8(9):1069–86). Although the ADCC/ADCP functions of fragment crystallizable region (Fc) can induce NK cells and macrophages to contribute anti-tumor responses, the ADCC/ADCP effects induced by Fc/Fc receptor interaction are not crucial for α PD-L1 therapy (Int J Cancer. 2019 Jan 15;144(2):345–54). In addition, α PD-L1 scFv that lacks Fc has been proved to be enough for inducing anti-tumor responses by enhancing T-cell activation (Nat Biotechnol. 2018 Oct;36(9):847–56, Cancer Immunol Res. 2020 May;8(5):632–47). And, blocking PD-L1 by α PD-L1 scFv can enhance the activation of both T cells and NK cells due to the PD-1 inhibitory receptor is expressed on both T and NK cells (Front Immunol. 2019 Jun 4;10:1242). Moreover, atezolizumab contains a modified Fc receptor binding site that removes ADCC and ADCP functions (Front Immunol. 2020 May 29; 11:1088) has shown good efficacy in treating various tumors (N Engl J Med.

2020 May 14;382(20):1894–905, N Engl J Med. 2020 Oct 1;383(14):1328–39, N Engl J Med. 2018 Nov 29;379(22):2108–21), and atezolizumab is approved for the treatments of multiple cancer types, including melanoma, small cell lung cancer (SCLC), NSCLC, RCC, head and neck squamous cell carcinomas (HNSCC), classical Hodgkin lymphomas (cHL), and Merkel cell carcinoma (Mol Cell. 2019 Nov 7;76(3):359–70). The α PD-L1 scFv without Fc can not only easily penetrate the tumor to block PD-L1 of tumor cells, but also avoid ADCC/ADCP-mediated potential toxicities to normal tissues. Therefore, α PD-L1 scFv without Fc is an excellent anti-PD-L1 drug even though its ADCC and ADCP functions are missing. More discussion about the rationale of α PD-L1 scFv therapy has been added in the revised manuscript (line 604 of page 36 to line 610 of page 37).

In regard to those published studies that combining scFv with CAR-T or oncolytic virus, their strategies are quite different from ours, and they also did not fully address the local expression of scFv and insufficient infiltration of CAR-T cells or T cells into tumors (line 597 to 604 of page 36 of the revised manuscript). We have discussed the novelty of our study in the Response to Major Comment 1.

We agree with Reviewer #1 that CAR-T cells and oncolytic virus can kill tumor cells to promote the release of tumor antigens for activating tumor antigen-specific immune responses. But tumor antigen release and presentation have also occurred both during and after the treatments of our nanomedicines. As mentioned in the Response to Major Comment 2, during tumor growth and the treatments, both alive and injured tumor cells will continuously expose and release tumor antigens, and tumor cells killed by the immune system (including T cells, NK cells, macrophages and so on that reside or infiltrate in the tumor microenvironment) will also release tumor antigens. DCs can capture these antigens and infiltrate into lymph nodes to activate tumor antigen-specific T cells (Cancer Lett. 2016 Jan 1;370(1):85-90). In addition, our results show that our NP_{Tyr-C9AP} have increased the proportions of activated NK cells (Fig. 5k and Fig. R1) and M1-like macrophages in tumors (Fig. 5l), which also contributes to the exposure, processing and presentation to activate tumor antigen-specific T cells. Thus, our

nanomedicines have shown induced anti-tumor effects even though tumor antigen release and APCs priming were not considered when designing our nanomedicines. Of course, we believe combining some strategies targeting tumor antigen release and APCs priming can further enhance the efficacy of our nanomedicines. More discussion has been added in the revised manuscript (line 641 of page 38 to line 645 of page 39).

Comment 4: The authors claimed this Nano-system could successfully produce α PD-L1 scFV and CXCL9 in tumor cells, however, they did not provide sufficient characterization for both protein expressions. For example, what are the protein production yields for both proteins in different tumor types? And what is the final working concentration they can achieve in the culture medium for in vitro experiments (CXCL9 was provided but scFV was missing)? And how about the tumor site? And what is the percentile of secreted proteins/total produced proteins? This information is important because it could help us to determine the working concentration of scFV for in vivo study. Since the data for protein characterization such as SDS-page or LS-MS of the produced proteins were not provided, it is suggested to provide the entire picture of the western blot with protein markers in the supplementary data.

Response: Thanks for the kind suggestions. We have added more characterizations of both CXCL9 and α PD-L1 protein expression. To detect the protein production yields and the percentiles of secreted/total CXCL9 and α PD-L1 in different cells, B16-F10 cells and other 7 types of cells were seeded in 24-well plates and transfected with NP_{Tyr-C9AP}. Seventy-two hours later, the culture supernatants and cell lysates were collected to quantify the production yields of CXCL9 and α PD-L1. As shown in Fig. 2h, after NP_{Tyr-C9AP} transfection, the concentrations of CXCL9 and α PD-L1 in the culture supernatants of B16-F10 cells reached 0.47 ng ml⁻¹ (CXCL9) and 2.19 ng ml⁻¹ (α PD-L1), which were much higher than that of other cells due to the melanoma specificity of tyrosinase promoter (line 145 of page 8 to line 148 of page 9 of the revised manuscript). After calculation, every million NP_{Tyr-C9AP}-transfected B16-F10 cells secreted 1.18 ng CXCL9 and 5.47 ng α PD-L1 proteins in the culture supernatants (Fig.

R2a and b). The protein quantity of CXCL9 and α PD-L1 in the cell lysates of NP_{Tyr-C9AP}-transfected B16-F10 cells were also detected. As shown in Fig. R2c and d, every million NP_{Tyr-C9AP}-transfected B16-F10 cells contained 0.42 ng CXCL9 and 1.16 ng α PD-L1 proteins inside the cells. Thus, the total proteins of CXCL9 and α PD-L1 produced by NP_{Tyr-C9AP}-transfected B16-F10 cells were 1.60 ng and 6.63 ng per million cells (Fig. R2e and f). Furthermore, the percentiles of secreted/total proteins were calculated using the following formula: $A / (A + B) \times 100\%$, where A is the quantity of protein in culture supernatants, and B is the quantity of protein in cell lysates. The percentiles of secreted/total CXCL9 and α PD-L1 proteins ranged from 64.4% to 99.2% after NP_{Tyr-C9AP} transfection (Fig. R2g and h).

Fig. 2h Concentrations of CXCL9 and α PD-L1 proteins in the culture supernatants of B16-F10 cells and other 7 types of cells after NP_{Tyr-C9AP} transfection. The data are shown as the means \pm SEM of $n = 3$. One-way ANOVA with Tukey's multiple comparison test. *** $P < 0.001$.

Fig. R2 Protein production yields of CXCL9 and α PD-L1, and ratios of secreted/total CXCL9 or α PD-L1 in different cells after NP_{Tyr-C9AP} transfection. **a, b** Protein quantity

of CXCL9 (a) and α PD-L1 (b) secreted in the culture supernatants of B16-F10 cells and other 7 types of cells after NP_{Tyr-C9AP} transfection. c, d Protein quantity of CXCL9 (c) and α PD-L1 (d) in the lysates of B16-F10 cells and other 7 types of cells after NP_{Tyr-C9AP} transfection. e, f Total proteins of CXCL9 (e) and α PD-L1 (f) produced by B16-F10 cells and other 7 types of cells after NP_{Tyr-C9AP} transfection. g, h Ratios of secreted/total CXCL9 (g) or α PD-L1 (h) protein produced by B16-F10 cells and other 7 types of cells after NP_{Tyr-C9AP} transfection. The data are shown as the means \pm SEM of $n = 3$. One-way ANOVA with Tukey's multiple comparison test. *** $P < 0.001$.

Similarly, we detected the protein production yields of CXCL9 and α PD-L1 in different cells after NP_{Sur-C9AP} transfection. B16-F10, CT26, Panc02, 4T1 tumor cells and 4 types of corresponding tissue-derived primary cells were transfected with NP_{Sur-C9AP}, and the concentrations and quantity of CXCL9 and α PD-L1 proteins in the culture supernatants and cell lysates were quantified as mentioned above. As shown in Fig. 8b, after NP_{Sur-C9AP} transfection, the concentrations of CXCL9 and α PD-L1 in the culture supernatants of 4 types of tumor cells were significantly higher than that of tissue-derived primary cells due to the tumor specificity of survivin promoter (line 516 to 520 of page 31 of the revised manuscript). The concentrations of CXCL9 and α PD-L1 in the culture supernatants of NP_{Sur-C9AP}-transfected B16-F10 cells reached 0.46 ng ml⁻¹ (CXCL9) and 2.20 ng ml⁻¹ (α PD-L1). Every million NP_{Sur-C9AP}-transfected B16-F10 cells secreted 1.16 ng CXCL9 and 5.49 ng α PD-L1 proteins in the culture supernatants (Fig. R3a and b). And, every million NP_{Sur-C9AP}-transfected B16-F10 cells contained 0.27 ng CXCL9 and 1.79 ng α PD-L1 proteins inside the cells (Fig. R3c and d). Thus, the total proteins of CXCL9 and α PD-L1 produced by NP_{Sur-C9AP}-transfected B16-F10 cells were 1.43 ng and 7.28 ng per million cells (Fig. R3e and f). The percentiles of secreted/total CXCL9 and α PD-L1 proteins ranged from 70.6% to 90.7% after NP_{Sur-C9AP} transfection (Fig. R3g and h).

Fig. 8b Concentrations of CXCL9 and α PD-L1 proteins in the culture supernatants of four types of tumor cells (B16-F10, CT26, Panc02, 4T1 cells) and four types of primary normal cells (mouse skin, colon, pancreas and breast cells) after NP_{Sur-C9AP} transfection. The data are shown as the means \pm SEM of $n = 3$. One-way ANOVA with Tukey's multiple comparison test. *** $P < 0.001$.

Fig. R3 Protein production yields of CXCL9 and α PD-L1, and the ratios of secreted/total CXCL9 or α PD-L1 in different cells after NP_{Sur-C9AP} transfection. **a, b** Protein quantity of CXCL9 (**a**) and α PD-L1 (**b**) secreted in the culture supernatants of 4 types of tumor cells and other 4 types of primary normal cells after NP_{Sur-C9AP} transfection. **c, d** Protein quantity of CXCL9 (**c**) and α PD-L1 (**d**) proteins in the lysates of 4 types of tumor cells and other 4 types of primary normal cells after NP_{Sur-C9AP} transfection. **e, f** Total proteins of CXCL9 (**e**) and α PD-L1 (**f**) produced by 4 types of tumor cells and other 4 types of primary normal cells after NP_{Sur-C9AP} transfection. **g, h** Ratios of secreted/total CXCL9 (**g**) or α PD-L1 (**h**) protein produced by 4 types of tumor cells and other 4 types of primary normal cells after NP_{Sur-C9AP} transfection. The data are shown as the means \pm SEM of $n = 3$. One-way ANOVA with Tukey's multiple comparison test. *** $P < 0.001$.

We are sorry that we only provided the final concentrations of CXCL9 in Fig. 3a while α PD-L1 expression was analyzed by western blot (Fig. 3d of the original manuscript). We have redetected the final concentrations of α PD-L1 using ELISA, and the western blot image of α PD-L1 expression was replaced by the new ELISA results. As shown in Fig. 3a and d, the final working concentrations of CXCL9 and α PD-L1 were 0.42 or 1.74 ng ml⁻¹ in the culture supernatants of NP_{Tyr-C9AP}-transfected B16-F10 cells (line 194 to 197 of page 12, line 220 to 222 of page 13 of the revised manuscript).

Fig. 3a, d Concentrations of CXCL9 (a) and α PD-L1(d) secreted in the culture supernatants of B16-F10 cells after the transfection of NP_{Tyr-C9AP} or other controls. The data are shown as the means \pm SEM of $n = 3$. One-way ANOVA with Tukey's multiple comparison test. *** $P < 0.001$.

To detect the concentrations of CXCL9 and α PD-L1 in melanoma tissues after NP_{Tyr-C9AP} treatment, C57BL/6 mice bearing B16-F10 melanoma were randomly divided into five groups ($n = 4$ mice per group) when the tumor volumes reached 100 mm³. Then, the mice were intravenously injected with PBS, NP_{Control}, NP_{Tyr-CXCL9}, NP_{Tyr- α PD-L1} or NP_{Tyr-C9AP} every other day for three injections. The injection doses were 1 mg pUC57, pTyr-CXCL9, pTyr- α PD-L1 or pTyr-C9AP per kg body weight. Seventy-two hours after the last injection, the melanoma tissues were collected and lysed to detect the intratumoral concentrations of CXCL9 and α PD-L1. ELISA results showed that the intratumoral concentrations of CXCL9 in NP_{Tyr-CXCL9} and NP_{Tyr-C9AP} groups were 13.52 and 14.07 ng per gram tumor tissue (Fig. 4a), and the intratumoral concentrations of α PD-L1 in NP_{Tyr- α PD-L1} and NP_{Tyr-C9AP} group were 54.88 or 57.82 ng per gram tumor tissue (Fig. 4b). The results have been described in the revised manuscript (line 312 and 313, 317 to 319 of page 19)

Fig. 4a, b Concentrations of CXCL9 (a) and α PD-L1 (b) in B16-F10 melanoma after the *i.v.* injection of NP_{Tyr-C9AP} or other controls ($n = 4$ mice per group). The data are

shown as the means \pm SEM of $n = 4$. One-way ANOVA with Tukey's multiple comparison test. *** $P < 0.001$.

Similarly, we detected the protein production yields of CXCL9 and α PD-L1 in tumor tissues after NP_{Sur-C9AP} treatment. Mice bearing B16-F10 melanoma, CT26 colorectal tumors, Panc02 pancreatic tumors, or 4T1 orthotopic breast tumors were randomly divided into two groups ($n = 4$ mice per group) when the tumor volumes reached 100–200 mm³. Then, the mice were intravenously injected with NP_{Sur-C9AP} or NP_{Control} every other day for three injections. The injection doses were 1 mg pSur-C9AP or pUC57 per kg body weight. Seventy-two hours later, the B16-F10, CT26, Panc02 and 4T1 tumor tissues were collected and lysed to detect the intratumoral concentrations of CXCL9 and α PD-L1 using ELISA. As shown in Fig. 8c and Supplementary Fig. 29a to c, the intratumoral concentrations of CXCL9 in B16-F10, CT26, Panc02 and 4T1 tumor tissues were 11.76, 7.18, 16.94 or 14.58 ng per gram tumor tissue. As shown in Fig. 8d and Supplementary Fig. 29d to f, the intratumoral concentrations of α PD-L1 in B16-F10, CT26, Panc02 and 4T1 tumor tissues were 55.65, 24.42, 63.83 or 48.26 ng per gram tumor tissue. The results have been described in the revised manuscript (line 520 to 526 of page 31).

Fig. 8c, d Concentrations of CXCL9 (**c**) and α PD-L1 (**d**) proteins in melanoma and normal organs of the NP_{Sur-C9AP}-treated mice ($n = 4$ mice per group). The concentrations of CXCL9 and α PD-L1 were indicated as ng protein per gram tissue (ng g⁻¹). The data are shown as the means \pm SEM of $n = 4$. One-way ANOVA with Tukey's multiple comparison test, Student's *t*-test. * $P < 0.05$; *** $P < 0.001$; ns indicates no significant difference.

Supplementary Fig. 29 Concentrations of CXCL9 and α PD-L1 proteins in different tumors and normal organs after NP_{Sur-C9AP} treatments. **a to c** Concentrations of CXCL9 protein in different tumors and normal organs of mice bearing CT26 (**a**), Panc02 (**b**) or 4T1 (**c**) tumors after NP_{Sur-C9AP} treatments. **d to f** Concentrations of α PD-L1 protein in different tumors and normal organs of mice bearing CT26 (**d**), Panc02 (**e**) or 4T1 (**f**) tumors after NP_{Sur-C9AP} treatments. Mice bearing CT26, Panc02 or 4T1 tumors were intravenously injected with NP_{Control} or NP_{Sur-C9AP} every other day for three injections ($n = 4$ mice per group). The injection doses were 1 mg pUC57 or pSur-C9AP per kg body weight. Seventy-two hours after the last injection, the tumor tissue, liver, lung, spleen, kidney and heart were isolated to detect the concentrations of CXCL9 and α PD-L1 using ELISA. The concentrations of CXCL9 and α PD-L1 were indicated as ng protein per gram tissue (ng g⁻¹). The data are shown as the means \pm SEM of $n = 4$. One-way ANOVA with Tukey's multiple comparison test, Student's t -test. * $P < 0.05$; ** $P < 0.01$; *** $P < 0.001$; ns indicates no significant difference.

As suggested, the entire uncropped western blot images with protein markers were provided in the Supplementary information. Please refer to the revised Supplementary Fig. 7 and Supplementary Fig. 12.

Supplementary Fig. 7 Western blot analysis of α PD-L1 expression in NP_{Tyr-C9AP}-transfected B16-F10 cells. B16-F10 cells were incubated with PBS, NP_{Control}, NP_{Tyr- α PD-L1} or NP_{Tyr-C9AP} for 6 h, and the culture supernatants and cell lysates were collected to detect α PD-L1 expression by western blot at 72 h after different transfections. The expression of β -actin in the cell lysates was detected as the reference control. The data are representative of three independent experiments.

Supplementary Fig. 12 Western blot analysis of α PD-L1 expression in melanoma after different treatments. Mice bearing B16-F10 melanoma were intravenously injected with PBS, NP_{Control}, NP_{Tyr-CXCL9}, NP_{Tyr- α PD-L1} or NP_{Tyr-C9AP} ($n = 4$ mice per group). The injection doses were 1 mg pUC57, pTyr-CXCL9, pTyr- α PD-L1 or pTyr-C9AP per kg body weight. Seventy-two hours after injection, the α PD-L1 expression in melanoma tissues were detected by western blot. The data are representative of three independent experiments.

Comment 5: The biodistribution of these nanoparticles should be carefully studied as it was missing in the current study. Preferably, the tumor-specific induction of gene expression (use Luciferase or RFP as reporter protein) and therapeutic effect (α PD-L1 and CXCL9) should be studied in vivo with bilateral tumor models (for example, B16-

F10 tumor on the right back and other types of tumors on the left side).

Response: Thanks for the good suggestions. We have analyzed the biodistribution of our nanomedicines in a bilateral tumor model. Because B16-F10 melanoma cells can produce a large amount of melanin that will seriously interfere with the detection of Cy5 fluorescence signal using IVIS instrument, we established the bilateral tumor model by subcutaneously inoculating 1.5×10^6 YUMM1.7 melanoma cells into the right flank of C57BL/6 mice and 5×10^6 Panc02 pancreatic tumor cells into the left flank. When the melanoma volume reached 400–500 mm³, the mice were intravenously injected Cy5-labeled NP_{Tyr-C9AP} at a dose of 1 mg Cy5-labeled pTyr-C9AP per kg body weight. Twelve hours later, the YUMM1.7 melanoma, Panc02 pancreatic tumors and normal organs were collected, and were analyzed using IVIS Lumina III Living Image system (PerkinElmer, MA, USA). As displayed in Supplementary Fig. 4, Cy5-labeled NP_{Tyr-C9AP} were accumulated in both YUMM1.7 and Panc02 tumor tissues, meanwhile, other normal organs including liver, spleen, lung and kidney also showed Cy5-labeled NP_{Tyr-C9AP} accumulation (line 150 to 153 of page 9, line 737 of page 43 to line 746 of page 44 of the revised manuscript).

Supplementary Fig. 4 Organ distribution of NP_{Tyr-C9AP} in bilateral tumor model. The bilateral tumor model was established by subcutaneously inoculating 1.5×10^6 YUMM1.7 melanoma cells into the right flank of C57BL/6 mice and 5×10^6 Panc02 pancreatic tumor cells into the left flank. When the melanoma volume reached 400–500 mm³, the mice were intravenously injected Cy5-labeled NP_{Tyr-C9AP} at a dose of 1 mg Cy5-labeled pTyr-C9AP per kg body weight. Twelve hours later, the YUMM1.7 melanoma, Panc02 pancreatic tumors and normal organs were collected, and were analyzed using IVIS Lumina III Living Image system (PerkinElmer, MA, USA).

For analyzing the melanoma-specific gene expression of NP_{Tyr-C9AP} in a bilateral tumor model, we subcutaneously inoculated 3×10^5 B16-F10 cells into the right flank of C57BL/6 mice and 5×10^6 Panc02 cells into the left flank. The mice were randomly divided into two groups ($n = 4$ mice per group) when the volumes of B16-F10 melanoma and Panc02 pancreatic tumors reached 50–100 mm³. Then, the mice bearing bilateral tumors were intravenously injected with NP_{Tyr-C9AP} or NP_{Control} every other day for three injections. The injection doses were 1 mg pTyr-C9AP (with EGFP reporter gene) or pUC57 per kg body weight. Seventy-two hours after the last injection, the B16-F10 melanoma, Panc02 pancreatic tumor, liver, lung, spleen, kidney and heart were isolated to analyze the percentages of EGFP-positive cells using flow cytometry, or were lysed to detect CXCL9 and α PD-L1 concentrations in different tissues using ELISA. As shown in Fig. 2i, after NP_{Tyr-C9AP} treatment, the percentages of EGFP-positive cells in B16-F10 melanoma reached 9.11% of the total cells, while few cells were EGFP-positive in Panc02 pancreatic tumors, heart, liver, spleen, lung and kidney. Furthermore, as shown in Fig. 2j and k, the concentrations of CXCL9 and α PD-L1 in melanoma of NP_{Tyr-C9AP}-treated mice were significantly increased to 12.12 and 43.12 ng per gram tumor tissue; as expected, the concentrations of CXCL9 and α PD-L1 in the pancreatic tumor, heart, liver, spleen, lung or kidney were not increased when compared to that of NP_{Control} group. These results indicated that NP_{Tyr-C9AP} could induce melanoma-specific expression of CXCL9 and α PD-L1 in vivo even though NP_{Tyr-C9AP} also were accumulated in other tumors and normal organs. The results and methods have been described in the revised manuscript (line 153 to 159 of page 9, line 762 of page 44 to line 771 of page 45).

Fig. 2i Percentages of EGFP-positive cells in melanoma, pancreatic tumor and normal organs of the NP_{Tyr-C9AP}-treated bilateral tumor mice ($n = 4$ mice per group). The data are shown as the means \pm SEM of $n = 4$. One-way ANOVA with Tukey's multiple comparison test, Student's t -test. *** $P < 0.001$; ns indicates no significant difference.

Fig. 2j, k Concentrations of CXCL9 (j) and α PD-L1 (k) in melanoma, pancreatic tumor and normal organs of the NP_{Tyr-C9AP}-treated bilateral tumor mice ($n = 4$ mice per group). The data are shown as the means \pm SEM of $n = 4$. One-way ANOVA with Tukey's multiple comparison test, Student's t -test. *** $P < 0.001$; * $P < 0.05$; ns indicates no significant difference.

For analyzing the tumor-specific induction of therapeutic effects in the bilateral tumor model, the mice bearing bilateral tumors were intravenously injected with NP_{Tyr-C9AP} or NP_{Control} every other day for five injections ($n = 6$ mice per group). The injection doses were 1 mg pTyr-C9AP or pUC57 per kg body weight. The length and width of tumors and body weights of the mice were monitored every other day. The tumors were collected at the end of the treatments and the final tumor weights were measured. As shown in Supplementary Fig. 16, the growth of B16-F10 melanoma were significantly inhibited by NP_{Tyr-C9AP} treatment while that of Panc02 pancreatic tumor was not affected, indicating that NP_{Tyr-C9AP} could induce melanoma-specific therapeutic effects (line 341 to 345 of page 20 of the revised manuscript).

Supplementary Fig. 16 NP_{Tyr-C9AP} induces melanoma-specific therapeutic effects. **a** to **d** Growth curves and final weights of B16-F10 melanoma (**a**, **b**) and Panc02 pancreatic tumor (**c**, **d**) in the bilateral tumor-bearing mice treated with NP_{Tyr-C9AP} or NP_{Control} ($n = 6$ mice per group). Mice bearing bilateral tumors were intravenously injected with NP_{Tyr-C9AP} or NP_{Control} every other day for five injections. The injection doses were 1 mg pTyr-C9AP or pUC57 per kg body weight. The length and width of tumors of the mice were monitored every other day. The tumors were collected at the end of the treatments and the final tumor weights were measured. The data are shown as the means \pm SEM of $n = 6$. Two-way ANOVA with the Greenhouse–Geisser correction (**a**, **c**), Student's t -test (**b**, **d**). *** $P < 0.001$; ns indicates no significant difference.

Collectively, these results demonstrated the ability of NP_{Tyr-C9AP} to specifically express CXCL9 and α PD-L1 in melanoma and to induce specific therapeutic effects against melanoma.

Comment 6: More data are needed to reveal the underlying mechanisms of α PD-L1-induced reduction of immunosuppressive cell populations. Also, why could NP-CXCL9 reduce the M1/M2 ratio and MDSC populations in tumors by enhanced infiltration of T cells? More evidence should be provided about the transformed immune microenvironment such as cytokines/chemokines changes.

Response: Thanks for the suggestions. The mechanisms of α PD-L1-induced reduction

of immunosuppression have been studied by many published works. For example, several groups have reported that α PD-L1 antibodies can reduce the number or activity of immunosuppressive cells, such as M2-like macrophages, MDSCs and regulatory T cells (Cancer Res. 2019 Apr 1;79(7):1493–1506, Biomark Res. 2021 Oct 24;9(1):77., Scand J Immunol. 2022 Mar;95(3): e13129). To further reveal the underlying mechanisms of our nanomedicines, the expression of 5 key cytokines related to the transformation of tumor immune microenvironment was detected. We intravenously injected B16-F10 melanoma-bearing mice with NP_{Control}, NP_{Tyr- α PD-L1} or NP_{Tyr-CXCL9} every other day for five injections. At the end of the treatments, the mRNA expression of three pro-inflammatory cytokines (IFN- γ , IL-6, GM-CSF) and two anti-inflammatory cytokines (IL-4 and IL-10) in melanoma tissues were detected by qRT-PCR using the primers in Supplementary Table 3. As shown in Supplementary Fig. 22a and b, in comparison to NP_{Control} group, the mRNA expression of pro-inflammatory IFN- γ , IL-6, and GM-CSF in the melanoma of NP_{Tyr- α PD-L1}- or NP_{Tyr-CXCL9}-treated mice were significantly increased; whereas, the mRNA expression of anti-inflammatory IL-4 and IL-10 were significantly decreased in the melanoma of NP_{Tyr-CXCL9}- or NP_{Tyr- α PD-L1}-treated mice. Thus, both NP_{Tyr- α PD-L1} and NP_{Tyr-CXCL9} increased the mRNA expression of three pro-inflammatory cytokines (IFN- γ , IL-6, GM-CSF), and decreased the mRNA expression of two anti-inflammatory cytokines (IL-4 and IL-10) in melanoma tissues. The increase of IFN- γ , IL-6, GM-CSF and decrease of IL-4 and IL-10 promoted the reduction of immunosuppressive cell populations, including polarization of M1-like/M2-like macrophages (Nat Rev Immunol. 2018 Sep; 18(9): 545–58), the apoptosis of MDSCs and Tregs (Eur J Immunol. 2014 Aug; 44(8): 2457–67, Cell Death Dis. 2021 May 18;12(6):501). The results have been described in the revised manuscript (line 394 to 397 of page 23).

Supplementary Table 3 qRT-PCR primer sequences for analyzing the expression of 5 key cytokines.

IFN- γ	Forward: 5'-ATGAACGCTACACACTGCATC-3' Reverse: 5'-CCATCCTTTTGCCAGTTCCTC-3'
IL-6	Forward: 5'-TAGTCCTTCTACCCCAATTTCC-3' Reverse: 5'-TTGGTCCTTAGCCACTCCTTC-3'
GM-CSF	Forward: 5'-GGCCTTGGAAGCATGTAGAGG-3' Reverse: 5'-GGAGAACTCGTTAGAGACGACTT-3'
IL-4	Forward: 5'-GGTCTCAACCCCCAGCTAGT-3' Reverse: 5'-GCCGATGATCTCTCTCAAGTGAT-3'
IL-10	Forward: 5'-GCTCTTACTGACTGGCATGAG-3' Reverse: 5'-CGCAGCTCTAGGAGCATGTG-3'

Supplementary Fig. 22 qRT-PCR analysis of the cytokine changes in tumor microenvironment after different treatments. **a** mRNA expression of IFN- γ , IL-6 and GM-CSF in melanoma after different treatments. **b** mRNA expression of IL-4 and IL-10 in melanoma after different treatments. B16-F10 melanoma-bearing mice were injected with NP_{Control}, NP_{Tyr- α PD-L1} or NP_{Tyr-CXCL9} every other day for five injections ($n = 3$ mice per group). The injection doses were 1 mg pUC57, pTyr- α PD-L1 or pTyr-CXCL9 per kg body weight. The mRNA expression of 5 key cytokines related to the transformation of immune microenvironment in melanoma were detected by qRT-PCR at the end of treatments using primers listed in Supplementary Table 3. The data are shown as the means \pm SEM of $n = 3$. One-way ANOVA with Tukey's multiple comparison test. * $P < 0.05$; ** $P < 0.01$; ns indicates no significant difference.

Minor comments:

Comment 1: It seems that the TEM image in Figure 2c indicated that the nanoparticles were aggregated.

Response: Thanks for the comment. In Fig. 2c, the NP_{Tyr-C9AP} samples for TEM imaging were prepared by dripping the aqueous solution of NP_{Tyr-C9AP} onto the holey carbon film coated Cu TEM grids followed by negative staining of phosphotungstic acid (PTA). Then, the NP_{Tyr-C9AP} samples on the holey carbon film were left to dry naturally, and the morphology of NP_{Tyr-C9AP} was examined using an Talos L120c TEM (Thermo Fisher, MA, USA) at an accelerating voltage of 120 kV. In the drying process, the NP_{Tyr-C9AP} samples may aggregate spontaneously due to the water evaporation-mediated concentration of the samples. However, according to the data of dynamic light scattering (DLS) (Fig. 2b), NP_{Tyr-C9AP} showed a 107.2 nm hydrodynamic diameter and the polydispersity index (PDI) of NP_{Tyr-C9AP} was less than 0.1, indicating that the size of NP_{Tyr-C9AP} was uniform and NP_{Tyr-C9AP} was not aggregated in aqueous solution.

Comment 2: Fluorescence images with clearer background and DAPI staining should be presented in Figure 2f.

Response: Thanks for the suggestion. We have repeated the experiments of Fig. 2f, and the figures have been replaced with fluorescence images with clearer background and DAPI staining. In Fig. 2f, B16-F10, CT26, Panc02, 4T1, C2C12, NIH/3T3, DC2.4 and RAW264.7 cells were seeded in 24-well plates at a density of 5×10^4 cells per well. Then, the cells were incubated with NP_{Tyr-C9AP} or NP_{Control} at a final concentration of 1 $\mu\text{g ml}^{-1}$ pTyr-C9AP (EGFP gene as reporter gene) or pUC57 for 6 h. Forty-eight hours after transfection, the cells were fixed and stained with DAPI, the fluorescence images of different cells were observed by fluorescence microscopy (Nikon, Tokyo, Japan).

Fig. 2f Fluorescence imaging of EGFP expression in B16-F10 cells and other 7 types of cells after NP_{Tyr-C9AP} transfection. Scale bar, 50 μm.

Comment 3: The quantitative dose of αPD-L1 induced in various tumors in vivo should be measured. And the PD-L1 expression level in the TME should also be studied. Will there be any response differences if the TME shows a different PD-L1 level?

Response: Thanks for the suggestions. In the Response to Major Comment 4, the quantitative concentrations of αPD-L1 induced in various tumors in vivo have been detected. As shown in Fig. 4b, after treating B16-F10 melanoma-bearing mice with NP_{Tyr-C9AP} and other controls, the intratumoral concentrations of αPD-L1 in NP_{Tyr-αPD-L1}- and NP_{Tyr-C9AP}-treated melanoma were 54.88 or 57.82 ng per gram tumor tissue (line 317 to 319 of page 19 of the revised manuscript).

Fig. 4b Concentrations of αPD-L1 in B16-F10 melanoma after the *i.v.* injection of NP_{Tyr-C9AP} or other controls ($n = 4$ mice per group). The data are shown as the means \pm SEM of $n = 4$. One-way ANOVA with Tukey's multiple comparison test. *** $P < 0.001$.

In addition, as shown in Fig. 8d and Supplementary Fig. 29d to f, the intratumoral concentrations of αPD-L1 in B16-F10, CT26, Panc02 and 4T1 tumors were 55.65,

24.42, 63.83 or 48.26 ng per gram tumor tissue after the treatments of NP_{Sur-C9AP} (line 520 to 526 of page 31 of the revised manuscript).

Fig. 8d Concentrations of α PD-L1 protein in melanoma and normal organs of the NP_{Sur-C9AP}-treated mice ($n = 4$ mice per group). The data are shown as the means \pm SEM of $n = 4$. One-way ANOVA with Tukey's multiple comparison test, Student's t -test. * $P < 0.05$; *** $P < 0.001$; ns indicates no significant difference.

Supplementary Fig. 29d to f Concentrations of α PD-L1 protein in different tumors and normal organs of mice bearing CT26 (**d**), Panc02 (**e**) or 4T1 (**f**) tumors after NP_{Sur-C9AP} treatments. Mice bearing CT26, Panc02 or 4T1 tumors were intravenously injected with NP_{Control} or NP_{Sur-C9AP} every other day for three injections ($n = 4$ mice per group). The injection doses were 1 mg pUC57 or pSur-C9AP per kg body weight. Seventy-two hours after the last injection, the tumor tissue, liver, lung, spleen, kidney and heart were isolated to detect the concentrations of α PD-L1 using ELISA. The concentrations of α PD-L1 were indicated as ng protein per gram tissue (ng g⁻¹). The data are shown as the means \pm SEM of $n = 4$. One-way ANOVA with Tukey's multiple comparison test, Student's t -test. * $P < 0.05$; ** $P < 0.01$; *** $P < 0.001$; ns indicates no significant difference.

To detect the expression levels of PD-L1 in different tumors, we established the B16-F10, CT26, Panc02 and 4T1 tumor mouse models as described in the Response to Major Comment 4. When the tumor volumes reached about 500–1000 mm³, the B16-F10, CT26, Panc02 or 4T1 tumor tissues were isolated and dissociated into single cells to analyze PD-L1 expression using flow cytometry. As shown in Fig. R4, the PD-L1 expression level in the tumor microenvironments of B16-F10 melanoma was the

highest, which was 2.0-, 12.1- and 5.5-fold of that in CT26, Panc02 and 4T1 tumors, respectively.

Fig. R4 PD-L1 expression in different tumor microenvironments. **a, b** Representative flow cytometry plots (**a**) and mean fluorescence intensity (MFI) (**b**) of PD-L1 expression in B16-F10, CT26, Panc02 and 4T1 tumor tissues. The data are shown as the means \pm SEM of $n = 3$.

The relationship between PD-L1 level of TME and therapeutic responses of α PD-L1 inhibitors has been studied by many published works, and PD-L1 expression is used as a biomarker for predicting the efficacy of α PD-L1 inhibitors (Mol Cancer Ther. 2015 Apr;14(4):847–56, Mol Cancer. 2018 Aug 23;17(1):129, Bioconjug Chem. 2018 Jan 17;29(1):96–103). In addition, our results also showed that the PD-L1 expression level in TME can cause different therapeutic responses. As shown in Fig. 8e and n, after NP_{Sur-C9AP} treatment, the tumor growth inhibition rate of B16-F10 melanoma was the highest (84.5%) and that of Panc02 tumor was the lowest (56.3%), while Fig. R4 showed that the PD-L1 level in B16-F10 melanoma was significantly higher than Panc02 tumor. Considering the intratumoral concentrations of CXCL9 and α PD-L1 proteins in B16-F10 and Panc02 tumors were comparable after NP_{Sur-C9AP} treatment (Fig. 8c, d and Supplementary Fig. 29b, e), these results suggested that high PD-L1 expression in TME can result in high therapeutic response to α PD-L1 inhibitors.

Fig. 8e, n Tumor growth curves of B16-F10 (**e**) and Panc02 (**n**) tumor-bearing mice treated with NP_{Sur-C9AP} or NP_{Control} ($n = 6$ mice per group). Two-way ANOVA with the Greenhouse–Geisser correction. *** $P < 0.001$.

Comment 4: CD8/CD4 ratio is not a good parameter for immune activation in tumors. The CD8/Treg ratio provides a better indication of immune activation. In addition, why did CXCL9 recruit more CD8 cells than CD4 T cells?

Response: Thanks for the suggestion and comment. We have replaced the results of CD8/CD4 ratio with the results of CD8/Treg ratio (line 380 and 381 of page 23, line 539 and 540 of page 32 of the revised manuscript), and the data are shown in Fig. 5f and Fig. 8h of the revised manuscript.

Fig. 5f Ratios of CD8⁺ T cells/Tregs in melanoma from the mice treated with NP_{Tyr-C9AP} or other groups ($n = 4$ mice per group). The data are shown as the means \pm SEM of $n = 4$. One-way ANOVA with Tukey's multiple comparison test. * $P < 0.05$; ** $P < 0.01$; *** $P < 0.001$; ns indicates no significant difference.

Fig. 8h Ratios of CD8⁺ T cells/Tregs in B16-F10 melanoma from the mice treated with NP_{Sur-C9AP} or NP_{Control} ($n = 4$ mice per group). The data are shown as the means \pm SEM of $n = 4$. Student's t -test. * $P < 0.05$.

In regard to the reason why CXCL9 recruited more CD8⁺ than CD4⁺ T cells, it may be because tumor cells express MHC-I/antigen peptide complex that can only be recognized by CD8⁺ T cells not MHC-II/antigen peptide complex. When CD8⁺ T cells are recruited in the tumor tissues, they will proliferate after recognizing MHC-I/antigen peptide complex. In addition, CXCL9 mainly recruits Th1 cells (CD4⁺) and CD8⁺ T cells, and Th1 cells recruited in the tumor site can further enhance the activation and proliferation of CD8⁺ T cells (Nat Rev Immunol. 2017 Sep;17(9):559-572), that maybe another important reason why more CD8⁺ T cells seemed to be recruited in tumors.

Comment 5: In vitro and vivo expression of CXCL9 and α PD-L1 should be determined on the protein level as well (quantitatively, related to figure 2 i to l, figure 6b).

Response: Thanks for the suggestion. In the Response to Major Comment 4, we have quantitatively detected the in vitro and in vivo expression of CXCL9 and α PD-L1 proteins using ELISA, and the results of Fig. 2i to l and Fig. 6b of the original manuscript have been replaced by the quantitative data of CXCL9 and α PD-L1 protein expression in Fig. 2h, j and k and Fig. 8b of the revised manuscript. The results have been described in the revised manuscript (line 145 of page 8 to line 148 of page 9, line 156 to 159 of page 9, line 516 to 520 of page 31).

Comment 6: Representative flow cytometry images related to Figs 5 and 6 should be provided as supplementary files.

Response: Thanks for the suggestion. The representative flow cytometry images related to Fig. 5 and 6 have been provided in the supplementary information. Please refer to the revised Supplementary Fig. 19, 21, 30, 31 and 33.

Comment 7: Did the authors observe systemic antigen-specific immune response after this nanoparticle treatment? And what about the generation of memory T cells and long-term immune protection from these nanoparticles?

Response: Thanks for the comments. Our study aims to address the problem of insufficient T-cell infiltration in tumors and the problems of low enrichment and potential toxicities caused by systemic injection of α PD-L1 inhibitors. These T cells recruited by CXCL9 have already been activated by diverse tumor antigens in tumor-draining lymph nodes and secondary lymphatic organs (Nat Rev Immunol. 2017 Sep;17(9):559-572). Thus, the T cells that kill tumor cells in our study were polyclonal with diverse tumor antigen specificities. So, we did not detect the antigen specificities of T cells to observe systemic antigen-specific immune responses. In addition, our strategy is different tumor vaccination strategy, thus we also did not analyze the generation of memory T cells and long-term immune protection.

Comment 8: To show the gating strategy of FACS, pseudocolor plots may provide more information.

Response: Thanks for the suggestion. The pseudocolor plots of FACS gating strategies have been added in the supplementary information. Please refer to Supplementary Fig. 17.

Reviewer #2:

General Comments: The manuscript entitled “Engineering tumor-specific gene nanomedicine to recruit and activate T cells for enhanced immunotherapy” describes a novel nanoparticle agent delivering an expression plasmid for producing anti-PD-L1 blocking scFV and a chemokine CXCL9 in melanoma cells. This is achieved by using a melanocyte-specific promoter. The authors also describe a similar nanoparticle that can be used for treating various cancers more broadly where the survivin promoter is used to drive anti-PD-L1 scFv and CXCL9 expression. The idea of modifying tumor cells using a nanomedicine approach in a way that would promote T cell infiltration and activity is interesting. Furthermore, the effect of described nanomedicines on T cell recruitment and activity is evident in vitro and in vivo, and the anti-tumor effect in a mouse model is impressive. One potential concern is that only one in vivo model was used to test melanoma-specific nanomedicine.

Response: Thanks for the comments. To address the potential concern, we have added two melanoma models, mice bearing Clone M-3 (Cloudman S91) and YUMM1.7 mouse melanoma, to test our melanoma-specific nanomedicine in vivo. The results show that NP_{Tyr-C9AP} can induce clone M-3 and YUMM1.7 melanoma to efficiently coexpress CXCL9 and α PD-L1, thereby enhancing the infiltration and activation of T cells in tumor tissues to achieve enhanced anti-melanoma effects (Fig. 7 of the revised manuscript, Supplementary Fig. 26 to 28 of the revised supplementary information). The results and methods have been described in the revised manuscript (line 465 of page 28 to line 499 of page 30, line 861 of page 49 to line 867 of page 50, line 883 of page 50 to line 888 of page 51).

In addition, we have constructed human tumor-specific gene nanomedicines (NP_{hTyr-C9AP} and NP_{hSur-C9AP}) that can specifically induce the expression of human CXCL9 and α PD-L1 in human tumor cells. As shown in the Fig R5a to d, NP_{hTyr-C9AP} can induce human melanoma cells to coexpress human CXCL9 and α PD-L1, and NP_{hSur-C9AP} can induce different human tumor cells to coexpress human CXCL9 and α PD-L1. Moreover, the Transwell migration and CCK-8 assay showed that NP_{hTyr-C9AP}

can enhance human T-cell recruitment and cytotoxicity by inducing A375 cells to secrete human CXCL9 and α PD-L1 (Fig. R6a and b). In summary, our gene nanomedicines can achieve antitumor effects in different tumor models. Please refer to the Response to Major Comment 1.

Major comments:

Comment 1: The NP_{Tyr-C9AP} was tested in B16 melanoma cells and tumors. It would be good to include 1-2 additional mouse melanoma models, such as YUMM models or other available models to ensure reproducibility. It would also make this study very compelling if a similar human-specific nanomedicine could be generated and tested in a human model, such as co-culture of Patient-Derived tumor cells and tumor infiltrating leukocyte, for example.

Response: Thanks for the suggestions. We have included two additional mouse melanoma models, Clone M-3 (Cloudman S91) and YUMM1.7 models, to test the efficacy of NP_{Tyr-C9AP} for reproducibility. The expression of CXCL9 and α PD-L1 in different melanoma cells, including B16-F10, Clone M-3 and YUMM1.7 cells, were firstly detected after NP_{Tyr-C9AP} transfection. As shown in Supplementary Fig. 26, CXCL9 concentrations in the culture supernatants of NP_{Tyr-C9AP}-transfected B16-F10, Clone M-3 and YUMM1.7 cells ranged from 0.25 to 0.53 ng ml⁻¹, while α PD-L1 concentrations ranged from 1.00 to 2.26 ng ml⁻¹. Then, the Clone M-3 and YUMM1.7 melanoma mouse models were established by subcutaneously inoculating 1.5×10^6 Clone M-3 cells or 1.5×10^6 YUMM1.7 cells into the right flanks of C57BL/6 mice. The melanoma-bearing mice were randomly divided into two groups ($n = 4$ mice per group) when the tumor volumes reached 100 mm³. Then, the mice were intravenously injected with NP_{Tyr-C9AP} or NP_{Control} every other day for three injections. The injection doses were 1 mg pTyr-C9AP or pUC57 per kg body weight. Seventy-two hours after the last injection, the melanoma tissues were collected and the intratumoral concentrations of CXCL9 and α PD-L1 were detected by ELISA. As shown in Fig. 7a, b, f and g, the intratumoral concentrations of CXCL9 and α PD-L1 were 11.11 and 36.82 ng per gram tumor tissue in NP_{Tyr-C9AP}-treated Clone M-3 melanoma, and were 15.06

and 54.79 ng per gram tumor tissue in NP_{Tyr-C9AP}-treated YUMM1.7 melanoma. These results indicated that NP_{Tyr-C9AP} could induce high expression of CXCL9 and α PD-L1 in different melanomas.

Moreover, the mice bearing Clone M-3 or YUMM1.7 melanoma ($n = 6$ mice per group) were intravenously injected with NP_{Tyr-C9AP} or NP_{Control} every other day for five injections when the tumor volumes reached 50 mm³. The injection doses were 1 mg pTyr-C9AP or pUC57 per kg body weight. The tumor volumes were monitored every other day and the final tumor weights were measured. As shown in Fig. 7c and h, NP_{Tyr-C9AP} treatment significantly inhibited the growth of Clone M-3 and YUMM1.7 melanoma, respectively achieving 69.5% and 56.0% tumor growth inhibition rates when compared to that of NP_{Control} group. The final tumor weights of Clone M-3 and YUMM1.7 melanoma were correspondingly reduced by NP_{Tyr-C9AP} treatment (Supplementary Fig. 27). In addition, the tumor tissues were collected for immune profiling of flow cytometry at the end of the treatments. The results showed that the percentages of tumor-infiltrating CD3⁺ T cells and CD69-positive CD8⁺ T cells were significantly increased after NP_{Tyr-C9AP} treatment (Fig. 7d, e, i and j, Supplementary Fig. 28).

Collectively, these results demonstrated that NP_{Tyr-C9AP} could induced anti-tumor effects in different melanoma models via coexpressing CXCL9 and α PD-L1 to enhance intratumoral infiltration and activation of T cells. These results and methods have been described in the revised manuscript (line 465 of page 28 to line 499 of page 30, line 861 of page 49 to line 867 of page 50, line 883 of page 50 to line 888 of page 51).

Supplementary Fig. 26 Concentrations of CXCL9 (a) and α PD-L1 (b) in the culture supernatants of different melanoma cells after NP_{Tyr-C9AP} transfection. B16-F10, Clone M-3 and YUMM1.7 cells were incubated with NP_{Control} or NP_{Tyr-C9AP} for 6 h. The transfection doses were 1 μ g ml⁻¹ pUC57 or pTyr-C9AP. The concentrations of CXCL9 and α PD-L1 in the culture supernatants were detected by ELISA at 72 h after different transfections. The data are shown as the means \pm SEM of $n = 3$.

Fig. 7 NP_{Tyr-C9AP} induces enhanced therapeutic effects against different melanomas. **a, b** Concentrations of CXCL9 (a) and α PD-L1 (b) in Clone M-3 melanoma after the *i.v.* injection of NP_{Tyr-C9AP} or NP_{Control} ($n = 4$ mice per group). **c** Tumor growth curves of the Clone M-3 melanoma-bearing mice treated with NP_{Tyr-C9AP} or NP_{Control} ($n = 6$ mice per group). **d, e** Percentages of CD3⁺ T cells (d) or CD69-positive CD8⁺ T cells (e) in Clone M-3 melanoma after the treatments. **f, g** Concentrations of CXCL9 (f) and α PD-L1 (g) in YUMM1.7 melanoma after the *i.v.* injection of NP_{Tyr-C9AP} or NP_{Control} ($n = 4$ mice per group). **h** Tumor growth curves of the YUMM1.7 melanoma-bearing mice treated with NP_{Tyr-C9AP} or NP_{Control} ($n = 6$ mice per group). **i, j** Percentages of CD3⁺ T cells (i) or CD69-positive CD8⁺ T cells (j) in YUMM1.7 melanoma after the treatments. The data are shown as the means \pm SEMs of $n = 4$ (a, b, d to g, i, j), or 6 (c, h). Student's *t*-test (a, b, d to g, i, j), two-way ANOVA with the Greenhouse–Geisser correction (c, h). * $P < 0.05$; ** $P < 0.01$; *** $P < 0.001$.

Supplementary Fig. 27 Final tumor weights of Clone M-3 (a) and YUMM1.7 (b)

melanoma-bearing mice after NP_{Tyr-C9AP} treatments. Mice bearing Clone M-3 or YUMM1.7 melanoma were intravenously injected with NP_{Control} or NP_{Tyr-C9AP} every other day for five injections ($n = 6$ mice per group). The injection doses were 1 mg pUC57 or pTyr-C9AP per kg body weight. The final tumor weights of the different melanoma-bearing mice were measured at the end of treatments. The data are shown as the means \pm SEM of $n = 6$. Student's t -test. *** $P < 0.001$.

Supplementary Fig. 28 Representative flow cytometry plots of CD8⁺ T cells and CD69-positive CD8⁺ T cells in different melanomas after NP_{Tyr-C9AP} treatments. **a** to **d** Representative flow cytometry plots of CD3⁺ T cell percentages in CD45⁺ cells (**a**, **b**) and CD69-positive cell percentages in CD8⁺ T cells (**c**, **d**) of Clone M-3 or YUMM1.7 melanoma. The statistical data were correspondingly shown in Fig. 7d, e, i, j.

We have additionally constructed the human tumor-specific gene nanomedicines. The human tyrosinase promoter (hTyr) was modified from a previous work (Cancer Gene Ther. 2005 Nov;12(11):864–72). Survivin promoter (Sur) was unchanged because the Sur promoter sequence used in this work can drive gene expression in both mouse and human tumor cells (Cancer Gene Ther. 2004 Apr;11(4):256–62). The anti-human PD-L1 scFv (α hPD-L1) sequence was obtained from the IMGT mAb database (DB ID 1119) and the human CXCL9 (hCXCL9) cDNA sequence was obtained from the CCDS database (CCDS34014.1). All the sequences have been provided in Table R1 and were synthesized by Sangon. Then, the hTyr- α hPD-L1-P2A-hCXCL9 fragment

or hSur- α hPD-L1-P2A-hCXCL9 fragment were inserted into the KasI and Sall sites of the pUC57 backbone plasmid (Sangon) to construct the phTyr-C9AP or phSur-C9AP plasmid. Finally, phTyr-C9AP or phSur-C9AP were encapsulated into our nanoparticles to prepare NP_{hTyr-C9AP} or NP_{hSur-C9AP} using the same method described in manuscript.

Table R1 DNA sequences of human tyrosinase promoter, survivin promoter, CXCL9 and anti-human PD-L1 scFv for constructing human tumor-specific gene nanomedicines.

Human tyrosinase promoter (650 bps)	aattctgtcttcgagaacatagaaaagaattatgaaatgccacatgtggttacaagtaatgcagac ccaaggctccccagggacaagaagtcttgtttaatctctttgtggctctgaaagaaagagagag agaaaagattaagcctccttggagatcatgtgatgacttctgattccagccagaggcagcaa ttctgtcttcgagaacatagaaaagaattatgaaatgccacatgtggttacaagtaatgcagacc aaggctccccagggacaagaagtcttgtttaatctctttgtggctctgaaagaaagagagagag aaaagattaagcctccttggagatcatgtgatgacttctgattccagccagaggcagcattct aaccataagaattaaactattaatggtgaatagagttttcactttaacataggcctatcccactggt gggatacgagccaattcgaaagaaaagtcagtcagcttttcagaggatgaaagcctaagata aagactaaaagtgttgatgctggaggtgggagtggtattataggtctcagccaagacatgtg ataatcactgtagtagtagctggaaagagaaatctgtgactccaattagccagttcctg
Survivin promoter (260 bps)	ggcagggacgagctggcgcggtcgtggtgcaccgacaccgggagagccacgc ggcgggaggactacaactcccggcacaccccgcgcgccccgcttactcccagaagggc gcgggggggtggaccgctaagaggcgtgctcccagatgccccggcgccgaccattaa ccgccagatttgaatcgcgggacccgttggcagaggtggcgggcgccatgggtgccccg acgttccccctgcctgg
Human CXCL9 (378 bps)	atgaagaaaagtgggttctttcctctgggcatcatcttctggttctgattggagtgaaggaac cccagtagtgagaaaggtcgtgttctctgcatcagcaccaccaagggactatccacctaaa tcttgaagacctaaacaatttgcccaagccttctgcgagaaaattgaaatcattgctacac tgaagaatggagtcaaacatgtctaaaccagattcagcagatgtgaaggaactgattaaaaag tgggagaaacaggtcagccaaaagaaaaagcaaaagaatgggaaaaaacatcaaaaaaga aagttctgaaagttcgaatctcaacgttctcgtcaaaagaagactacataa

Anti- human PD-L1 scFv (798 bps)	atgtacagcatgcagctcgcacatcctgtgtcacattgacactgtgctccttgcaacagcgaattca gctacgtgctgacacagcctccttctgtgtctgtggccctggccagacagccagaattacatgt ggaggaaacaacatcggctctaagtctgtgcactggtatcagcagaagcctggccaggccct gtgctgggtgtacgatgataacgataggccttctggcctgcctgagagatttctggcttaact ctggcaacacagctaccctgaccatcagcagagtggaggccggagatgaggccgattactact gtcaggtgtgggactctagcagcgatcacgtggtgttcggcggcggcacaagctgacagtgc tgggcggcggcggctctggagggcggcgaagcggcggcggcggatctgaggtgcagctgc tggagccaggcggcggcctggtgcagccaggcggcagcctgagactgagctgtgaggcctc cggctccaccttctccacctacgccatgtcttgggtgaggcaggccctggaaaggactgga gtgggtgtctggatttagcggatctggaggctcacctttacgctgattctgtgagaggcagatt acaatcagcagagatagctctaagaacaccctgttctgcagatgagctctctgagagccgagg acaccgccgtgtactactgcgccatcctgccagaggctacaactacggctcttccagcactgg ggccagggcacactggtgaccgtgtctagc
--	---

Then, the efficiency of the human tumor-specific gene nanomedicines, NP_{hTyr-C9AP} or NP_{hSur-C9AP}, for coexpressing hCXCL9 and α hPD-L1 in human tumor cells was detected. A375 (human melanoma cell line), HCT-116 (human colorectal cancer cell line) and MDA-MB-231 (human triple-negative breast cancer cell line) cells were seeded in 24-well plates at a density of 5×10^4 cells per well. Then, the cells were incubated with NP_{hTyr-C9AP} or NP_{hSur-C9AP} at a final concentration of $1 \mu\text{g ml}^{-1}$ pH_{Tyr-C9AP} or pH_{Sur-C9AP} for 6 h. Forty-eight hours after transfection, the cells were collected to analyze hCXCL9 and α hPD-L1 mRNA expression by qRT-PCR. The qRT-PCR primer sequences have been provided in Table R2. Seventy-two hours after transfection, the culture supernatants were collected to detect the concentrations of hCXCL9 and α hPD-L1 by ELISA. As shown in Fig. R5a and b, after NP_{hTyr-C9AP} transfection, the mRNA and protein expression of hCXCL9 and α hPD-L1 in A375 cells were significantly increased; by contrast, HCT-116 and MDA-MB-231 cells only showed weak or even no hCXCL9 and α hPD-L1 expression. In addition, NP_{hSur-C9AP} transfection induced high coexpression of hCXCL9 and α hPD-L1 in A375, HCT-116 and MDA-MB-231 cells (Fig. R5c, d). These results demonstrated that the human tumor-specific gene nanomedicines have been generated, and NP_{hTyr-C9AP} or NP_{hSur-C9AP} can be respectively used for inducing melanoma- or different tumor-specific coexpression of hCXCL9 and α hPD-L1.

Fig. R5 Human tumor-specific gene nanomedicine for coexpressing hCXCL9 and αhPD-L1 in human tumor cells. **a, b** mRNA (**a**) and protein (**b**) expression of hCXCL9 and αhPD-L1 in different human tumor cells after NP_{hTYr-C9AP} transfection. **c, d** mRNA (**c**) and protein (**d**) expression of hCXCL9 and αhPD-L1 in different human tumor cells after NP_{hSur-C9AP} transfection. The data are shown as the means ± SEM of $n = 3$. One-way ANOVA with Tukey's multiple comparison test. ** $P < 0.01$; *** $P < 0.001$.

Table R2 qRT-PCR primer sequences for analyzing the mRNA expression of human CXCL9 and anti-human PD-L1.

Human CXCL9	Forward: 5'-CCAGTAGTGAGAAAGGGTCGC-3' Reverse: 5'-AGGGCTTGGGGCAAATTGTT-3'
Anti-human PD-L1 scFv	Forward: 5'-CGCTGATTCTGTGAGAGGCAG-3' Reverse: 5'-GATGGCGCAGTAGTACACG-3'
Human GAPDH	Forward: 5'-GGAGCGAGATCCCTCCAAAAT-3' Reverse: 5'-GGCTGTTGTCATACTTCTCATGG-3'

Due to the impact of COVID-19 pandemic, we are unable to obtain tumor tissues from cancer patients. Thus, we used A375 (human melanoma cell line) cells and human primary CD8⁺ cytotoxic T cells (PB009-3F-C, OriBiotech, Shanghai, China; ethical approval number: LP202006) for in vitro functional assays. The human T-cell

recruitment was analyzed by tracking human CD8⁺ T-cell chemotaxis in a transwell migration assay. The human CD8⁺ T cells were cultured in RPMI 1640 medium supplemented with 10% FBS, 20 ng ml⁻¹ recombinant human IL-2 (Peprotech), and were activated by soluble anti-human CD3 antibody (clone: OKT3, BioLegend) and anti-human CD28 antibody (clone: CD28.2, BioLegend) according to the manufacturer's instructions before use. Then, activated human CD8⁺ T cells were cultured in the upper chamber of transwell plates at a density of 1 × 10⁶ cells per well, and the culture supernatants of NP_{Control}- or NP_{hTyr-C9AP}-transfected A375 cells were added in the lower chamber. Three hours after incubation, human CD8⁺ T cells recruited to the lower chamber were collected and labeled with FITC anti-human CD8a antibody (clone: SK1, BioLegend). Finally, 10 μl of Precision Count Beads (BioLegend) was added to each sample before flow cytometry analysis. The numbers of recruited human CD8⁺ T cells were calculated using the following formula: Absolute Count of human CD8⁺ T Cells = Ratio of human CD8⁺ T Cells/Counting Beads × Absolute Count of Counting Beads. As shown in Fig. R6a, the culture supernatants of NP_{hTyr-C9AP}-transfected A375 cells that contained a high concentration of hCXCL9 recruited 1.78 × 10⁵ human CD8⁺ T cells, which was significantly higher than that of NP_{Control} group (3.8 × 10⁴). These results demonstrated that NP_{hTyr-C9AP} could be used for recruiting human CD8⁺ T cells via inducing hCXCL9 expression of human melanoma cells.

The efficiency of NP_{hTyr-C9AP} for blocking hPD-L1 to enhance the cytotoxicity of human CD8⁺ T cells was further analyzed. Human myeloid-derived DCs were purchased from OriBiotech (PB-DC002F-C, Shanghai, China; ethical approval number: LP202006) and cultured in RPMI 1640 medium supplemented with 10% FBS, 20 ng ml⁻¹ GM-CSF (Peprotech) and 10 ng ml⁻¹ IL-4 (Peprotech) for two days. Then, human DCs were resuspended in fresh culture medium and incubated with A375 cell lysates containing tumor-associated antigens. Twenty-four hours later, 250 U ml⁻¹ TNF-α (Peprotech) were added to the medium as relevant maturation stimulant. Subsequently, human CD8⁺ T cells were cocultured with the matured DCs at a cell ratio of 10:1. Three days after coculture, the human CD8⁺ T cells were obtained and cocultured with IFN-

γ -stimulated A375 cells at a cell ratio of 10:1 in the culture supernatants of NP_{Control}- or NP_{hTyr-C9AP}-transfected A375 cells. After incubating for 24 hours, the supernatants containing human CD8⁺ T cells were removed, and CCK-8 reagent was added to the A375 cells followed by measuring the absorbance at 450 nm. As shown in Fig. R6b, the CCK-8 assay demonstrated that hPD-L1 blockade mediated by α hPD-L1 in the culture supernatants of NP_{hTyr-C9AP}-transfected A375 cells significantly increased the cytotoxicity of human CD8⁺ T cells, thereby reducing the viability of the IFN- γ -stimulated A375 cells to 43.08%. These results indicated that NP_{hTyr-C9AP} could enhance the activation and cytotoxicity of human CD8⁺ T cells by mobilizing human melanoma cells to secrete α hPD-L1.

Fig. R6 Human tumor-specific gene nanomedicine for enhancing human T-cell recruitment and activation. **a** Counts of human CD8⁺ T cells recruited into the lower chamber of the transwell plates after placing the culture supernatants of NP_{Control}- or NP_{hTyr-C9AP}-transfected A375 cells. **b** Cell viability of IFN- γ -stimulated A375 cells after being cocultured with activated human CD8⁺ T cells in the culture supernatants of NP_{Control}- or NP_{hTyr-C9AP}-transfected A375 cells. The data are shown as the means \pm SEM of $n = 3$. Student's t -test. ** $P < 0.01$; *** $P < 0.001$.

Comment 2: Additional characterization of potential toxicities could further improve the study. For instance, serum levels of liver enzymes AST and ALT could be assessed. In addition, if feasible, Cy5-labeled NP_{Tyr-C9AP} could be administered in vivo to see in which organs nanoparticles would accumulate.

Response: Thanks for the suggestions. For detecting the potential toxicities of our nanomedicines, C57BL/6 mice were intravenously injected with PBS, NP_{Control}, NP_{Tyr-CXCL9}, NP_{Tyr- α PD-L1} or NP_{Tyr-C9AP} every other day for five injections ($n = 4$ mice per group). The injection doses were 1 mg pUC57, pTyr-CXCL9, pTyr- α PD-L1 or pTyr-

C9AP per kg body weight. One week after the last injection, the serum samples were collected to detect the levels of AST, ALT and ALP using ELISA kits (Rayto, Guangdong, China). As shown in Supplementary Fig. 15, the serum levels of AST, ALT and ALP were not significantly increased after the treatments, indicating that NP_{Tyr-C9AP} did not cause potential toxicities to liver and was safe in mice. The results and methods have been described in the revised manuscript (line 339 to 341 of page 20, line 890 to 896 of page 51).

Supplementary Fig. 15 Potential liver toxicity analysis of NP_{Tyr-C9AP}. **a to c** Serum levels of alanine aminotransferase (ALT) (**a**), aspartate aminotransferase (AST) (**b**) and alkaline phosphatase (ALP) (**c**) after different treatments. C57BL/6 mice were intravenously injected with PBS, NP_{Control}, NP_{Tyr-CXCL9}, NP_{Tyr-αPD-L1} or NP_{Tyr-C9AP} every other day for five injections ($n = 4$ mice per group). The injection doses were 1 mg pUC57, pTyr-CXCL9, pTyr-αPD-L1 or pTyr-C9AP per kg body weight. One week after the last injection, the serum samples were collected to detect the concentrations ALT, AST and ALP using ELISA kits (Rayto, Guangdong, China). The data are shown as the means \pm SEM of $n = 4$. One-way ANOVA with Tukey's multiple comparison test. ns indicates no significant difference.

Reviewer #1 also suggested us to detect the organ accumulation of our nanomedicines. As mentioned in the Response to Major Comment 5 of Reviewer #1, we have analyzed the biodistribution of Cy5-labeled NP_{Tyr-C9AP} in a bilateral tumor model. Because B16-F10 melanoma cells can produce a large amount of melanin that will seriously interfere with the detection of Cy5 fluorescence signal using IVIS instrument, we established the bilateral tumor model by subcutaneously inoculating 1.5×10^6 YUMM1.7 melanoma cells into the right flank of C57BL/6 mice and 5×10^6 Panc02 pancreatic tumor cells into the left flank. When the melanoma volume reached 400–500 mm³, the mice were intravenously injected Cy5-labeled NP_{Tyr-C9AP} at a dose

of 1 mg Cy5-labeled pTyr-C9AP per kg body weight. Twelve hours later, the YUMM1.7 melanoma, Panc02 pancreatic tumors and normal organs were collected, and were analyzed using IVIS Lumina III Living Image system (PerkinElmer, MA, USA). As displayed in Supplementary Fig. 4, Cy5-labeled NP_{Tyr-C9AP} were accumulated in both YUMM1.7 and Panc02 tumor tissues, meanwhile, other normal organs including liver, spleen, lung and kidney also showed Cy5-labeled NP_{Tyr-C9AP} accumulation (line 150 to 153 of page 9, line 737 of page 43 to line 746 of page 44 of the revised manuscript).

Supplementary Fig. 4 Organ distribution of NP_{Tyr-C9AP} in bilateral tumor model. The bilateral tumor model was established by subcutaneously inoculating 1.5×10^6 YUMM1.7 melanoma cells into the right flank of C57BL/6 mice and 5×10^6 Panc02 pancreatic tumor cells into the left flank. When the melanoma volume reached 400–500 mm³, the mice were intravenously injected Cy5-labeled NP_{Tyr-C9AP} at a dose of 1 mg Cy5-labeled pTyr-C9AP per kg body weight. Twelve hours later, the YUMM1.7 melanoma, Panc02 pancreatic tumors and normal organs were collected, and were analyzed using IVIS Lumina III Living Image system (PerkinElmer, MA, USA).

Comment 3: Discussion mostly states the results. It is important to include a proper discussion. For example, describe literature on nanomedicines/nanoparticles with a similar design, clinical experience, etc.

Response: Thanks for the suggestions. We have revised the discussion of the revised manuscript (line 582 of page 35 to line 645 of page 39). As suggested, we have added more discussion about literatures on the clinical experiences of nanomedicines, and have discussed the current progress on nanomedicines with similar designs (line 611 to

line 623 of page 37). In addition, we have added more discussion about the novelty and rational of our strategy. The discussion about the results have also been refined.

Reviewer #3:

General Comments: This manuscript describes an interesting approach to simultaneously enhance T cell recruitment and activity against solid tumors through a nanomedicine approach designed to engineer tumor cells to secrete CXCL9 and anti-PD-L1. The major strength of the work is the novelty of the approach; whilst other approaches have been developed to engineer tumor cells to express CXCR3 ligands, this is the first to my knowledge to specifically utilize tumor promoters to achieve this. One question that emerges from this work is the relevance of using this system to target anti-PD-L1 specifically to the tumor microenvironment. To improve the therapeutic relevance of this study further experiments are suggested to confirm the utility of the Survivin promoter in human tumor cells relative to relevant control cells derived from similar tissues and to benchmark therapeutic efficacy to anti-PD-L1.

Response: Thanks for the comments. As we have discussed in the Introduction section of the manuscript and the Response to General Comments of Reviewer #1, insufficient tumor accumulation of α PD-L1 inhibitors after systemic injection is one major cause of the unsatisfactory therapeutic benefits of α PD-L1 therapy, and systemic injection of α PD-L1 inhibitors can induce T-cell cytotoxicity against normal tissues (Ann Oncol. 2015 Dec;26(12):2375–91). Thus, targeting α PD-L1 inhibitors specifically to the tumor microenvironment is important. To further demonstrate the relevance of using our nanomedicines to target α PD-L1 specifically to the tumor microenvironment, we studied the relevance between the α PD-L1 concentration and the cytotoxicity of CD8⁺ T cells using CCK8 assay. As shown in Fig. R7a, after coculturing B16-F10-OVA-EGFP cells and OVA-specific CD8⁺ T cells in the culture supernatants containing different concentrations of α PD-L1 protein, higher concentrations of α PD-L1 protein resulted in lower viability of B16-F10-OVA-EGFP cells, indicating that using our

nanomedicines to locally express high concentration of α PD-L1 in the tumor microenvironment can enhance anti-tumor effects of T cells. Please refer to the Response to Minor Comment 3. Furthermore, we have additionally compared the anti-tumor effects of tumor-specific targeting of α PD-L1 through our nanomedicines and systemic administration of α PD-L1 antibody. The results demonstrated that targeting α PD-L1 specifically to the tumor microenvironment using our nanomedicines showed superior efficacy and safety as opposed to the systemic administration of α PD-L1 antibody (Fig. 6 and Supplementary Fig. 23 to 25; line 416 of page 25 to line 463 of page 28 of the revised manuscript). Please refer to the Response to Major Comment 1.

In regard to the utility of the Survivin (Sur) promoter in human tumor cells relative to relevant control cells derived from similar tissues, we have retested the specificity of NP_{Sur-C9AP} in four types of tumor cells (B16-F10, CT26, Panc02, 4T1 cells) and four types of primary normal cells derived from similar tissues (mouse skin, colon, pancreas and breast cells). As shown in Fig. 8b, NP_{Sur-C9AP} induced high expression of CXCL9 and α PD-L1 proteins in B16-F10, CT26, Panc02, and 4T1 tumor cells, but hardly induced the expression of CXCL9 and α PD-L1 proteins in the primary normal cells derived from mouse skin, colon, pancreas, and breast tissues (line 516 to 520 of page 31 of the revised manuscript). Please refer to the Response to Major Comment 5. Additionally, because the Sur promoter used in this study can specifically drive gene expression in mouse and human tumor cells (Cancer Gene Ther. 2004 Apr;11(4):256–62), we have constructed human tumor-specific gene nanomedicine NP_{hSur-C9AP} for specifically coexpressing hCXCL9 and α hPD-L1 in human tumor cells. As shown in the Fig. R6c and d, NP_{hSur-C9AP} transfection induced high coexpression of hCXCL9 and α hPD-L1 in different human tumor cells, including A375, HCT-116 and MDA-MB-231 cells. These results demonstrated that the utility of the Sur promoter in human tumor cells, and NP_{hSur-C9AP} can be used for inducing coexpression of hCXCL9 and α hPD-L1 in different human tumor cells. More detailed information about the human tumor-specific gene nanomedicines is in the Response to Major Comment 1 of Reviewer #2.

Major comments:

Comment 1: Can the authors demonstrate superior efficacy / safety of targeting PD-L1 through this approach as opposed to systemic administration of anti-PD-L1

Response: Thanks for the suggestions. We have compared the therapeutic effects of tumor-specific targeting of α PD-L1 through our nanomedicines and systemic administration of anti-PD-L1 antibody. As shown in Fig. 6a, mice bearing B16-F10 melanoma were randomly divided into five groups ($n = 6$ mice per group) when the tumor volumes reached 100 mm^3 . Then, the mice were intravenously injected with PBS, anti-PD-L1 antibody, NP_{Tyr- α PD-L1}, NP_{Tyr-CXCL9} & anti-PD-L1 antibody or NP_{Tyr-C9AP}. Anti-PD-L1 antibody (InVivo MAb anti-mouse PD-L1(B7-H1), clone: 10F.9G2) was purchased from BioXCell (NH, USA), and was injected every other day for five injections at an injection dose of 2.5 mg per kg body weight. NP_{Tyr-CXCL9}, NP_{Tyr- α PD-L1} and NP_{Tyr-C9AP} were injected every other day for five injections and the injection doses were 1 mg pTyr-CXCL9, pTyr- α PD-L1 or pTyr-C9AP per kg body weight. The tumor volumes and body weights of the mice were monitored every other day. The tumor tissues were collected at the end of the treatments and the final tumor weights were measured. As shown in Fig. 6b and c, the tumor growth inhibition rate of NP_{Tyr- α PD-L1} group (64.2%) was higher than that of anti-PD-L1 group (58.0%), and the final tumor weights of NP_{Tyr- α PD-L1} group were significantly lower than that of anti-PD-L1 group, indicating that specifically expressing α PD-L1 in the tumor microenvironment using our nanomedicine induced superior efficacy as opposed to the systemic injection of anti-PD-L1 antibody. Significantly, NP_{Tyr-C9AP} group exhibited the highest tumor growth inhibition rate (78.2%) and the lowest final tumor weights due to tumor-specific coexpression of CXCL9 and α PD-L1. In addition, NP_{Tyr-C9AP} treatment significantly enhanced the intratumoral infiltration and activation of T cells in comparison to systemic injection of anti-PD-L1 antibody (Fig. 6d to i, Supplementary Fig. 24 and 25). Collectively, these results demonstrated that the therapeutic effects of NP_{Tyr-C9AP} were superior to that of systemically injected anti-PD-L1 antibody. These results and methods have been described in the revised manuscript (line 417 of page 25 to line 427

of page 26, line 433 of page 26 to line 451 of page 27, line 898 of page 51 to line 908 of page 52).

Fig. 6 NP_{Tyr-C9AP} exhibits superior efficacy to systemically injected anti-PD-L1 antibody. **a** Therapeutic scheme. Mice bearing B16-F10 melanoma were treated by five *i.v.* injections of NP_{Tyr-C9AP}, anti-PD-L1 antibody or other controls. **b, c** Tumor growth curves (**b**) and final tumor weights (**c**) of the B16-F10 melanoma-bearing mice after different treatments ($n = 6$ mice per group). **d, e** Flow cytometry plots of CD3⁺ T cell percentages in CD45⁺ cells (**d**) or statistical percentages of CD3⁺ T cells (**e**) in melanoma after different treatments ($n = 4$ mice per group). **f** Numbers of CD69-positive CD8⁺ T cells in melanoma. **g to i** Granzyme B- (**g**), perforin- (**h**) and IFN- γ - (**i**) positive CD8⁺ T cells in melanoma. The data are shown as the means \pm SEMs of $n = 4$ (**e to i**), $n = 6$ (**b, c**) or are representative of four independent experiments (**d**). Two-way ANOVA with the Greenhouse–Geisser correction (**b**), one-way ANOVA with Tukey’s multiple comparison test (**c, e to i**). * $P < 0.05$; ** $P < 0.01$; *** $P < 0.001$; ns indicates no significant difference.

As mentioned in the Response to Major Comment 2 of Reviewer #2, we have studied the potential toxicities of our nanomedicines. The results demonstrated that either NP_{Tyr- α PD-L1} or NP_{Tyr-C9AP} treatment did not increase the serum levels of AST, ALT and ALP (Supplementary Fig. 15, line 339 to 341 of page 20), indicating that NP_{Tyr- α PD-L1} or NP_{Tyr-C9AP} caused no potential toxicities to liver and was safe in mice. By contrast, systemic injection of anti-PD-L1 antibody has been reported to induce T-

cell cytotoxicity against normal tissues (Ann Oncol. 2015 Dec;26(12):2375–91). In addition, as shown in Supplementary Fig. 23, the body weights of mice were significantly decreased by the systemic injection of anti-PD-L1 antibody, while either NP_{Tyr-αPD-L1} or NP_{Tyr-C9AP} treatment did not obviously affect the body weights when compared to PBS group. These results indicated that specifically expressing αPD-L1 in the tumor microenvironment using our nanomedicines was safer than the systemic injection of anti-PD-L1 antibody (line 427 to 432 of page 26 of the revised manuscript).

Supplementary Fig. 15 Potential liver toxicity analysis of NP_{Tyr-C9AP}. **a to c** Serum levels of alanine aminotransferase (ALT) (**a**), aspartate aminotransferase (AST) (**b**) and alkaline phosphatase (ALP) (**c**) after different treatments. C57BL/6 mice were intravenously injected with PBS, NP_{Control}, NP_{Tyr-CXCL9}, NP_{Tyr-αPD-L1} or NP_{Tyr-C9AP} every other day for five injections ($n = 4$ mice per group). The injection doses were 1 mg pUC57, pTyr-CXCL9, pTyr-αPD-L1 or pTyr-C9AP per kg body weight. One week after the last injection, the serum samples were collected to detect the concentrations ALT, AST and ALP using ELISA kits (Rayto, Guangdong, China). The data are shown as the means \pm SEM of $n = 4$. One-way ANOVA with Tukey's multiple comparison test. ns indicates no significant difference.

Supplementary Fig. 23 Body weights of mice bearing B16-F10 melanoma after different treatments. Mice bearing B16-F10 melanoma were intravenously injected with PBS, anti-PD-L1 antibody, NP_{Tyr-αPD-L1}, NP_{Tyr-CXCL9} & anti-PD-L1 antibody or NP_{Tyr-C9AP} every other day for five injections ($n = 6$ mice per group), as illustrated in Fig. 6a. Anti-PD-L1 antibody was injected every other day for five injections and the injection dose was 2.5 mg per kg body weight. NP_{Tyr-CXCL9}, NP_{Tyr-αPD-L1} and NP_{Tyr-C9AP}

were injected every other day for five injections and the injection doses were 1 mg pTyr-CXCL9, pTyr- α PD-L1 or pTyr-C9AP per kg body weight. The body weights of the mice were monitored every other day since the first injection. The data are shown as the means \pm SEM of $n = 6$. Two-way ANOVA with the Greenhouse–Geisser correction. * $P < 0.05$.

Comment 2: The extent of the immune infiltration observed post therapy by IF (Figure 5g) looks rather modest. How do the authors reconcile this with profound therapeutic effects?

Response: Thanks for the concerns. Tumor tissues contain many kinds of cells, especially tumor cells, stromal cells and macrophages are the most abundant (Curr Biol. 2020 Aug 17; 30(16): R921–R925.) while CD3⁺ or CD8⁺ T cells are the minority. That is why the increased infiltration of CD3⁺ or CD8⁺ T cells in the IF images of Fig. 5g seemed to be modest after NP_{Tyr-C9AP} treatment even though the numbers of CD3⁺ or CD8⁺ T cells in the IF images have been obviously increased when compared to the control groups. The data of flow cytometry analysis can more precisely demonstrate the increased infiltration of CD3⁺ or CD8⁺ T cells. As shown in Fig. 5a and b, NP_{Tyr-C9AP} significantly increased the percentage of CD3⁺ T cells to 21.4% of CD45⁺ lymphocytes and increased the number of tumor-infiltrating CD3⁺ T cells to 1.87×10^6 cells per gram tumor in melanoma (line 363 to 371 of page 22 of the revised manuscript). Because a small number of CD8⁺ T cells could kill many tumor cells (Proc Natl Acad Sci U S A. 2006 Jul 18;103(29):10985–90.), the CD8⁺ T cells recruited by NP_{Tyr-C9AP} treatment exhibited strong anti-melanoma effects (Fig. 4e, line 324 of page 19 to line 328 of page 20 of the revised manuscript).

Fig. 5a, b Flow cytometry plots of CD3⁺ T cell percentages in CD45⁺ cells (a) or numbers of CD3⁺ T cells (b) of the melanoma samples from the mice treated with NP_{Tyr-C9AP} or other groups ($n = 4$ mice per group). The data are shown as the means \pm SEMs of $n = 4$ (b) or are representative of four independent experiments (a). One-way

ANOVA with Tukey's multiple comparison test. ** $P < 0.01$; *** $P < 0.001$; ns indicates no significant difference.

Comment 3: Some of the observations appear under powered and it is not apparent how many repeats have been performed e.g. Figure 2l, 4a $n = 3$ mice per group, others e.g. figure 4d, 4f are $n = 10$ mice per group but not stated how many repeats.

Response: Thanks for the concerns. According to the principles of Replacement Refinement & Reduction of Animals in Research, we endeavor to use fewer animals as long as the data are sufficient enough to answer the research questions. Thus, the in vivo experiments were usually conducted by one preliminary experiment using only 2~3 mice and one formal experiment using the indicated number of mice. So, the number of repetitions was not stated. In addition, for the in vivo experiments, such as Figure 2l, 4a, d and f of the original manuscript, each group contained multiple mice and each mouse was biologically independent, which can represent the reproducibility.

Comment 4: Figure 5h the data on CD3⁺ T cells should be stratified at least into CD4⁺ and CD8⁺, otherwise it could be that changes in expression of effector molecules merely reflect changes in the proportion of CD8⁺ T cells.

Response: Thanks for the suggestions. Because CD8⁺ T cells are the main cytotoxic T cells that secrete effector molecules, including granzyme B, perforin and IFN- γ , we have reanalyzed the flow cytometry data by gating CD8⁺ T cells. The results of the percentages of granzyme B-, perforin- and IFN- γ -positive cells in CD8⁺ T cells have been added in Fig. 5h to j of the revised manuscript and Supplementary Fig. 20 of the supplementary information (line 386 to 390 of page 23 in the revised manuscript).

Fig. 5h to j Representative flow cytometry plots of granzyme B- (**h**), perforin- (**i**) and IFN- γ - (**j**) positive cell percentages in CD8⁺ T cells in B16-F10 melanoma after different treatments.

Supplementary Fig. 20 Percentages of granzyme B- (**a**), perforin- (**b**), IFN- γ - (**c**) positive CD8⁺ T cells in B16-F10 melanoma after different treatments. The representative flow cytometry plots were correspondingly shown in Fig. 5h to j. The data are shown as the means \pm SEM of $n = 4$. One-way ANOVA with Tukey's multiple comparison test. * $P < 0.05$; ** $P < 0.01$; *** $P < 0.001$; ns indicates no significant difference.

Comment 5: IN Figure 6b the selectivity of the Survivin promoter is demonstrated but this is relative to other cell lines that are not relevant to the tissue from which the tumors originated. Matched samples should be used, preferably using human-derived tumor cells.

Response: Thanks for the suggestions. As mentioned above, we have retested the specificity of NP_{Sur-C9AP} in B16-F10, CT26, Panc02, and 4T1 tumor cells relative to primary normal cells derived from similar tissues. As shown in Fig. 8b, NP_{Sur-C9AP} transfection induced high expression of CXCL9 and α PD-L1 in the four types of tumor cells (B16-F10, CT26, Panc02, and 4T1 cells), but hardly induced the expression of CXCL9 and α PD-L1 in the four types of primary normal cells derived from mouse skin, colon, pancreas, and breast tissues. Thus, NP_{Sur-C9AP} was proven to be able to coexpress CXCL9 and α PD-L1 specifically in different tumor cells rather than most normal tissue cells. The results and methods have been modified in the revised manuscript (line 516 to 520 of page 31, line 677 to 679 of page 40, line 772 to 776 of page 45).

Fig. 8b Concentrations of CXCL9 and α PD-L1 proteins in the culture supernatants of four types of tumor cells (B16-F10, CT26, Panc02, 4T1 cells) and four types of primary normal cells (mouse skin, colon, pancreas and breast cells) after NP_{Sur-C9AP} transfection. The data are shown as the means \pm SEM of $n = 3$. One-way ANOVA with Tukey's multiple comparison test. *** $P < 0.001$.

The reason why we did not use human-derived tumor cells to demonstrate the specificity of NP_{Sur-C9AP} was because the Survivin (Sur) promoter used in this study can specifically drive gene expression in both mouse and human tumor cells (Cancer Gene Ther. 2004 Apr;11(4):256–62). Even so, we have used the Sur promoter to construct human tumor-specific gene nanomedicine NP_{hSur-C9AP} for specifically coexpressing hCXCL9 and α hPD-L1 in human tumor cells. As shown in the Fig. R5c and d, NP_{hSur-C9AP} transfection induced high coexpression of hCXCL9 and α hPD-L1 in different human tumor cells, including A375, HCT-116 and MDA-MB-231 cells. These results demonstrated that the utility of the Sur promoter in human tumor cells, and NP_{hSur-C9AP} can be used for inducing coexpression of hCXCL9 and α hPD-L1 in different human

tumor cells. More detailed information about the human tumor-specific gene nanomedicines is in the Response to Major Comment 1 of Reviewer #2.

Fig. R5c, d mRNA (**c**) and protein (**d**) expression of hCXCL9 and αhPD-L1 in different human tumor cells after NP_{hSur-C9AP} transfection. The data are shown as the means ± SEM of $n = 3$.

Minor comments:

Comment 1: It is suggested to add information regarding how many bps of the relevant promoters were incorporated into the construct within the main text, as this is critical information with regards to specificity.

Response: Thanks for the suggestions. The information regarding base pair length and DNA sequences of Tyr promoter and Sur promoter has added in the revised manuscript (line 696 to line 699 of page 41) and supplementary information (Supplementary Table 1).

Supplementary Table 1 DNA sequences of Tyr promoter, Sur promoter, CXCL9 and αPD-L1.

Tyr promoter (546 bps)	tctgcaggtcatagttcctgccagctgactttgtcaagacagtgatgtctgtgtccagcagttg tctgagtatcctttcattatccactgtcctttctttaaattccaccccaacattgtaaatagctt ctttctaaactctgttcaaagaaccagcttgagtgtgtcagctgcttctgctggggtcctggc aaaccactagtgcctttattcataagagatgatgtattcttgatactacttctcattgcaaattcc aattattattaatttcatatcaattagaataatatacttcttcaatttagttacctcactatgggcta tgtacaaactccaagaaaaagttagtcattgtgctttgcagaagataaaagcttagttaaaca ggctgagagatttgatgaagaagggagtggttatataggtcttagccaaaacatgtgata gtcactccaggggttgctggaaaagaagtctgtgacactcattaacctattggtgcagatgtt atgatctaaaggagac
---

Sur promoter (260 bps)	ggcagggacgagctggcgcggcgtcgtgggtgcaccgcgaccacgggcagagccacg cggcgggaggactacaactcccggcacaccccgcgcccccgccttactcccagaag gccgcgggggggtggaccgcctaagagggcgtgcgctcccgacatccccgcggcgcgc cattaaccgccagatttgaatcgcgggacccgttggcagaggtggcggcggcggcatggg tgccccgacgttccccctgcctgg
CXCL9 (378 bps)	atgaagtccgctgttctttctcttggcatcatcttctggagcagtgtggagttcgaggaac cctagtataaggaatgcacgatgctcctgcatcagcaccagccgaggcacgatccactac aaatccctcaaagacctcaaacagtttgccecaagcccaattgcaacaaaactgaaatcatt gtactactgaagaacggagatcaaactgcctagatccggactcggcaaatgtgaagaagc tgatgaaagaatgggaaaagaagatcagccaaaagaaaaagcaaaagagggggaaaaaa catcaaaagaacatgaaaaacagaaaacccaaaacaccccaaaagtcgtcgtcgttcaagga agactaca
α PD-L1 (795 bp)	atggaaaccgataccctgctgctgtgggtgctgctgctgtgggtgccgggcagcaccggcg atgatattcagatgaccagagcccagcagcctgagcgcgagcgtgggcgatcgcgtga ccattacctgccgcgagccaggatgtgagcaccgcggtggcgtggtatcagcagaaac cgggcaaaagcggcgaactgctgattatagcgcgagctttctgtatagcggcgtgccgagc cgctttagcggcagcggcagcggcaccgattttaccctgaccattagcagcctgcagccgg aagatthtgcgacctattattgccagcagatctgtatcatccggcgacctttggccagggcac caaaagtggaaattaacgcggaggcgggtggtccggcggaggagcagcgggtggcggga ggttccgaagtgcagctggtgaaagcggcggcggcctggtgcagccggcggcagcct gcgctgagctgcgcggcagcggctttacccttagcagatagctggattcattgggtgcgcc aggcggcgggcaaaggcctggaatgggtggcgtggattagcccgtatggcggcagcacct attatgcggatagcgtgaaaggcggctttaccattagcgcggataccagcaaaaacaccgc gtatctgcagatgaacagcctgcgcgggaagataaccgcggtgtattattgcgcgcgccgc cattggccggcggcgtttgattattggggccagggcacccctggtgaccgtgagcgcgctga gcagc

Comment 2: In Figure 2 what is the concentration of CXCL9 protein that is achieved following the use of NP_{Tyr-C9AP}?

Response: Thanks for the concerns. As mentioned in the Response to Major Comment 4 of Reviewer #1, we have added more characterizations of both CXCL9 and α PD-L1 protein expression. For detecting the concentrations of CXCL9 protein in the culture supernatants of different cells after NP_{Tyr-C9AP} transfection, B16-F10 cells and other 7 types of cells were seeded in 24-well plates and transfected with NP_{Tyr-C9AP}. Seventy-two hours later, the culture supernatants were collected for quantifying CXCL9 concentrations using ELISA. As shown in Fig. 2h, after NP_{Tyr-C9AP} transfection, the concentrations of CXCL9 in the culture supernatants of B16-F10 cells reached 0.47 ng

ml⁻¹, which were much higher than that of other cells due to the melanoma specificity of tyrosinase promoter (line 145 of page 8 to line 148 of page 9, line 749 to 761 of page 44 of the revised manuscript).

Fig. 2h Concentrations of CXCL9 and α PD-L1 proteins in the culture supernatants of B16-F10 cells and other 7 types of cells after NP_{Tyr-C9AP} transfection. The data are shown as the means \pm SEM of $n = 3$. One-way ANOVA with Tukey's multiple comparison test. *** $P < 0.001$.

Next, as mentioned in the Response to Major Comment 5 of Reviewer #1, the concentrations of CXCL9 protein in the tumor tissues and different organs after NP_{Tyr-C9AP} treatment have also been quantified. A bilateral tumor model was established by subcutaneously inoculating 3×10^5 B16-F10 cells into the right flank of C57BL/6 mice and 5×10^6 Panc02 cells into the left flank. The mice were randomly divided into two groups ($n = 4$ mice per group) when the volumes of B16-F10 melanoma and Panc02 pancreatic tumors reached 50–100 mm³. Then, the mice bearing bilateral tumors were intravenously injected with NP_{Tyr-C9AP} or NP_{Control} every other day for three injections. The injection doses were 1 mg pTyr-C9AP (with EGFP reporter gene) or pUC57 per kg body weight. Seventy-two hours after the last injection, the B16-F10 melanoma, Panc02 pancreatic tumor, liver, lung, spleen, kidney and heart were isolated to detect the concentrations of CXCL9 protein using ELISA. As shown in Fig. 2j, the concentration of CXCL9 in melanoma of NP_{Tyr-C9AP}-treated mice was 12.12 ng per gram tumor tissue. By contrast, the concentrations of CXCL9 in the Panc02 pancreatic tumor, heart, liver, spleen, lung or kidney of NP_{Tyr-C9AP}-treated mice were lower than 1.88 ng per gram tissue, which were close to that of the negative control (NP_{Control})

group. The results and methods have been described in the revised manuscript (line 156 to 159 of page 9, line 762 of page 44 to line 771 of page 45).

Fig. 2j Concentrations of CXCL9 in melanoma, pancreatic tumor and normal organs of the NP_{Tyr-C9AP}-treated bilateral tumor mice ($n = 4$ mice per group). The concentration of CXCL9 was indicated as ng protein per gram tissue (ng g⁻¹). The data are shown as the means \pm SEM of $n = 4$. One-way ANOVA with Tukey's multiple comparison test, Student's t -test. *** $P < 0.001$; ns indicates no significant difference.

Comment 3: What are the kinetics of the killing assays in Figure 3? The reviewer asks as generally it is difficult to model the impact of PD-1: PD-L1 interactions in vitro.

Response: Thanks for the concerns. We agree with the reviewer that it is difficult to model the impact of PD-1: PD-L1 interactions in vitro. Even so, we have conducted experiments to explore the impact of PD-1: PD-L1 interaction on the killing effects of T cells via changing the doses of α PD-L1 protein, the ratios of tumor cells/CD8⁺ T cells, or the coculture time.

For analyzing the dose-dependent relationship of α PD-L1, B16-F10-OVA-EGFP cells were stimulated with 20 ng ml⁻¹ IFN- γ (Peprotech) for 24 h to increase their PD-L1 expression. Primary OVA-specific CD8⁺ T cells were sorted from the spleens of OT-1 transgenic mice using a CD8a (Ly2) Microbeads Kit (Miltenyi Biotec., CA, USA) and were activated by soluble anti-mouse CD3 antibody (clone: 145-2C11, BioLegend) and anti-mouse CD28 antibody (clone: 37.51, BioLegend). Then, IFN- γ -stimulated B16-F10-OVA-EGFP cells and activated OVA-specific CD8⁺ T cells were cocultured at a cell ratio of 1:5 in the culture supernatants containing 1, 2 or 4 ng ml⁻¹ α PD-L1 protein. Twenty-four hours after coculture, the supernatants containing OVA-specific CD8⁺ T cells were removed, and CCK-8 reagent was added to the B16-F10-OVA-

EGFP cells followed by measuring the absorbance at 450 nm. As shown in Fig. R7a, with the concentrations of α PD-L1 protein increased from 1 to 4 $\mu\text{g ml}^{-1}$, the viability of B16-F10-OVA-EGFP cells decreased from 83.64% to 45.67%, indicating increasing α PD-L1 dose to reduce the PD-1: PD-L1 interactions can enhance the anti-tumor cytotoxicity of CD8⁺ T cells.

For analyzing the kinetics of the killing assays, IFN- γ -stimulated B16-F10-OVA-EGFP cells and activated OVA-specific CD8⁺ T cells were cocultured at different ratios (tumor cells/CD8⁺ T cells: 1:2, 1:5 or 1:10) in the culture supernatants of NP_{Control}- or NP_{Tyr- α PD-L1}-transfected B16-F10 cells. After coculturing for 4, 8, 12 or 24 hours, the supernatants containing OVA-specific CD8⁺ T cells were removed, and CCK-8 reagent was added to detect the viability of B16-F10-OVA-EGFP cells. As shown in Fig. R7b, with the increase of tumor cells/CD8⁺ T cells ratio and the extension of coculture time, the viability of B16-F10-OVA-EGFP cells decreased gradually. These results indicated that increasing the T-cell number or killing time can enhance the anti-tumor cytotoxicity of CD8⁺ T cells.

Fig. R7 CCK-8 assays of the kinetics of CD8⁺ T cell killing enhanced by α PD-L1. **a** Cell viability of IFN- γ -stimulated B16-F10-OVA-EGFP cells after being cocultured with OVA-specific CD8⁺ T cells in the culture supernatants containing different α PD-L1 concentrations. **b** Cell viability of IFN- γ -stimulated B16-F10-OVA-EGFP cells after being cocultured with OVA-specific CD8⁺ T cells at different cell ratios and different time points in the culture supernatants of NP_{Control}- or NP_{Tyr- α PD-L1}-transfected B16-F10 cells. The data are shown as the means \pm SEM of $n = 4$.

Comment 4: The authors state that the recruitment of NK cells contributed to the therapeutic efficacy. Whilst this is possible, it was not experimentally determined and so this comment should be softened.

Response: Thanks for the suggestions. We have softened the description about the therapeutic role of the recruitment of NK cells in the revised manuscript (line 390 to 394 of page 23).

REVIEWERS' COMMENTS

Reviewer #1 (Remarks to the Author):

All the previous critiques have been addressed with additional experiments and further discussion of this study and the literature. The technical issues have been largely fixed, and further discussion with the citation of additional literature has alleviated the concern about the novelty of this study. No new critiques are given to the revised manuscript.

Reviewer #2 (Remarks to the Author):

The author addressed my comments.

Reviewer #3 (Remarks to the Author):

The authors have provided some new data to address some concerns I raised with regards to specificity of the promoter and FACS gating strategies. However, there are two remaining points that I feel were in adequately addressed.

1) As demonstrated In Figure 6 there is very minimal benefit compared to systemically administered anti-PD-L1. This is unlikely to yield a significant difference in the clinical situation. It is unusual that anti-PD-L1 induced a loss of body weight as this has not been previously reported with the use of anti-PD-L1 in C57BL/6 mice.

2) This reviewer is not at all convinced that one formal experiment per therapeutic model is sufficient to provide robust conclusions. Further repeats should be performed for these studies.

We greatly appreciate the comments of the referees, helping us improve the quality of the manuscript. For the purpose of clarity, the reviewer's comments are in black while our point-by-point responses are marked in blue.

Reviewers' comments:

Reviewer #1:

General comment: All the previous critiques have been addressed with additional experiments and further discussion of this study and the literature. The technical issues have been largely fixed, and further discussion with the citation of additional literature has alleviated the concern about the novelty of this study. No new critiques are given to the revised manuscript.

Response: Thanks a lot.

Reviewer #2:

General comment: The author addressed my comments.

Response: Thanks a lot.

Reviewer #3:

General comment: The authors have provided some new data to address some concerns I raised with regards to specificity of the promoter and FACS gating strategies. However, there are two remaining points that I feel were in adequately addressed.

Response: Thanks a lot.

Comment 1: As demonstrated In Figure 6 there is very minimal benefit compared to systemically administered anti-PD-L1. This is unlikely to yield a significant difference in the clinical situation. It is unusual that anti-PD-L1 induced a loss of body weight as this has not been previously reported with the use of anti-PD-L1 in C57BL/6 mice.

Response: Thanks for the comments. As shown in Fig. 6b, it's true that NP_{Tyr- α PD-L1}

expressing anti-PD-L1 scFv only induced very minimal benefit in comparison to the systemically administered anti-PD-L1 antibody, but NP_{TyT-C9AP} expressing both anti-PD-L1 scFv and CXCL9 showed significantly better anti-tumor effects than the systemically administered anti-PD-L1 antibody. The tumor growth inhibition rate of NP_{TyT-C9AP} treatment group was 78.2% while that of anti-PD-L1 antibody treatment group was 58.0%. In addition, the tumor immune profiling results of the tumors after different treatments also showed that local expression of CXCL9 and anti-PD-L1 scFv in tumor tissue by NP_{TyT-C9AP} induced significantly higher intratumoral infiltration and activation of T cells. We agree with the reviewer that the therapeutic effects in mouse model can not ensure significant benefits in patients due to the differences between the tumors of human and mouse. Thus, we will investigate the anti-tumor effects of our tumor-specific gene nanomedicines in further preclinical and clinical studies.

In regard to the loss of body weight of C57BL/6 mice after the systemic injection of anti-PD-L1 antibody in our study, we speculate that may result from PD-L1 blockade-induced T-cell cytotoxicity to normal tissues. Because PD-L1 is not only expressed in tumor cells but also in dendritic cells (DCs), macrophages, B cells, endothelial cells, epithelial cells, fibroblasts, and so on (Nat Med. 2003 May;9(5):562–7, Sci Transl Med. 2016 Mar 2;8(328):328rv4). Systemic injection of anti-PD-L1 antibody can block the PD-L1 of these normal cells to induce T-cell cytotoxicity against these normal tissues (Ann Oncol. 2015 Dec;26(12):2375–91), thereby leading to immune-related adverse events (irAEs), such as diarrhea, rash, colitis, hepatitis, hypothyroidism, arthritis, etc. (Curr Oncol Rep. 2020 Mar 21;22(4):1–11, Nat Rev Clin Oncol. 2019 Sep;16(9):563–80). These irAEs have been widely observed in the clinical situation after the treatments of anti-PD-L1 antibodies, including atezolizumab, durvalumab and avelumab, with an incidence of about 20–60% (Nat Commun. 2022 Jan 19;13(1):392, Lancet. 2017 Jan 21;389(10066):255–65, Lancet Oncol. 2018 Apr;19(4):521–36, Lancet Oncol. 2017 May;18(5):599–610). Some works reported that the irAEs caused by systemically administered anti-PD-L1 antibody could lead to the loss of body weight of BALB/c mice (J Control Release. 2019 Jun 28;304:233–41,

Sci Immunol. 2019 Jul 12;4(37):eaau6584). As for C57BL/6 mice, some works reported that anti-PD-L1 antibody treatment induced no body weight loss (Cancer Immunol Immunother. 2021 Dec;70(12):3541–3555), but the results of some other works also showed that anti-PD-L1 antibody treatment could slightly affect the body weight of C57BL/6 mice (Biochem Biophys Res Commun. 2020 Feb 12;522(3):604–11). In our study, the loss of body weight was observed in C57BL/6 mice after the systemic administration of anti-PD-L1 antibody, which maybe because the high frequency of administration (every other day for five injections) aggravated the irAEs of anti-PD-L1 antibody.

Comment 2: This reviewer is not at all convinced that one formal experiment per therapeutic model is sufficient to provide robust conclusions. Further repeats should be performed for these studies.

Response: Thanks for the comment. We agree that repeats of experiments are important. To acquire solid conclusions, all of the therapeutic experiments in this work were carefully designed according to the principle of 3Rs (Replacement, Reduction and Refinement) and ARRIVE guidelines. As we have mentioned in the previous response, the therapeutic experiments were repeated twice, including one experiment using 2–3 mice per group and one experiment using 6–10 mice per group. Each mouse in the experiments was biologically independent, which can also represent the reproducibility. In addition, we have investigated the therapeutic effects of our gene nanomedicines in three melanoma models and three other tumor models to provide robust conclusions. Especially, the therapeutic effects of NP_{TyT-C9AP} on B16-F10 melanoma model were repetitively tested in three independent experiments (Fig. 4e, Fig. 4g and Fig. 6b), which also can be considered as repeats to prove the reliability of our conclusions. Furthermore, the therapeutic experiments of Fig. 5 and Fig. 8 were additionally repeated in the last revision to obtain comprehensive immune profiling of tumor tissues. Thus, the therapeutic experiments have been performed repetitively and the conclusions are solid.